# Comprehensive omic characterization of breast cancer in Mexican-Hispanic women

Sandra L. Romero-Cordoba [1,2,3,9,10✉], Ivan Salido-Guadarrama [4,9], Rosa Rebollar-Vega [2,5,6,9], Veronica Bautista-Piña[7], Carlos Dominguez-Reyes[7], Alberto Tenorio-Torres[7], Felipe Villegas-Carlos[7], Juan C. Fernández-López [8], Laura Uribe-Figueroa[2], Luis Alfaro-Ruiz[2,5] & Alfredo Hidalgo-Miranda [2,10✉]

Breast cancer is a heterogeneous pathology, but the genomic basis of its variability remains poorly understood in populations other than Caucasians. Here, through DNA and RNA portraits we explored the molecular features of breast cancers in a set of Hispanic-Mexican (HM) women and compared them to public multi-ancestry datasets. HM patients present an earlier onset of the disease, particularly in aggressive clinical subtypes, compared to non-Hispanic women. The age-related COSMIC signature 1 was more frequent in HM women than in those from other ancestries. We found the $AKT1^{E17K}$ hotspot mutation in 8% of the HM women and identify the $AKT1/PIK3CA$ axis as a potentially druggable target. Also, HM luminal breast tumors present an enhanced immunogenic phenotype compared to Asiatic and Caucasian tumors. This study is an initial effort to include patients from Hispanic populations in the research of breast cancer etiology and biology to further understand breast cancer disparities.

[1] Departamento de Medicina Genómica y Toxicología Ambiental, Instituto de Investigaciones Biomédicas, Universidad Nacional Autónoma de México, Mexico City, Mexico. [2] Cancer Genomics Laboratory, National Institute of Genomic Medicine, México City, Mexico. [3] Biochemistry Department, Instituto Nacional de Ciencias Médicas y Nutrición Salvador Zubirán, Mexico City, Mexico. [4] Laboratorio de Biología Computacional, Instituto Nacional de Enfermedades Respiratorias Ismael Cosío Villegas, Mexico City, Mexico. [5] Genomics Laboratory, Red de Apoyo a la Investigación, Universidad Nacional Autónoma de México-Instituto Nacional de Ciencias Médicas y Nutrición Salvador Zubirán, México City, México. [6] Programa de Doctorado en Ciencias Biomédicas, Universidad Nacional Autónoma de México (UNAM), México City, México. [7] Instituto de Enfermedades de la Mama FUCAM, Mexico City, México. [8] Computational Genomics Laboratory, National Institute of Genomic Medicine, Mexico City, Mexico. [9] These authors contributed equally: Sandra L. Romero-Cordoba, Ivan Salido-Guadarrama, Rosa Rebollar-Vega. [10] These authors jointly supervised this work: Sandra L. Romero-Cordoba, Alfredo Hidalgo-Miranda. ✉email: sromero@iibiomedicas.unam.mx; ahidalgo@inmegen.gob.mx

Breast cancer (BC) is the most common neoplasia in women around the globe and it represents an increasingly urgent health problem worldwide. Out of 19.7 million cases over the next 10 years, 10.6 million will occur in low-and middle-income countries[1]. In Mexico, breast tumors represent the main cause of cancer in women and epidemiological projections esti-mate the number of new cases and the mortality rates will increase in the next years[2].

BC is a heterogeneous disease, both at clinical and molecular levels, as evidenced by the various therapeutic rate responses and by the identification of intrinsic biological and molecular subtypes that present unique features, which may enhance cancer cell fit-ness and increase clinical risk and therapeutic resistance[3]. In the clinical practice, the inmunohistochemical markers estrogen (ER); progesterone (PR) and human epidermal growth factor receptor 2 (HER2) are used to guide diagnosis and treatment decisions[4]. Molecular classification of breast tumors based on gene expres-sion patterns has also been successfully translated into tests to support clinical decisions. Gene expression profiling, particularly the PAM50 intrinsic subtyping signature, has identified at least five categories of breast tumors: luminal A (LumA), luminal B (LumB), HER2-enriched (HER2), basal-like (basal) and claudin-low tumors[3], each one with distinctive oncogenic features. Fur-ther, increased genomic instability is a relevant feature of breast tumors, both at point mutation and at somatic DNA copy-number alterations (SCNA) levels. These two processes play an important role in activating oncogenes or in inactivating tumor suppressors genes[5], and they can also provide important infor-mation about BC biology.

Latin America represents a particular segment of western countries with emerging economies, which in the last years has shown an important rise in cancer diagnosis, especially BC. Recently, Hispanic/Latino populations has been the largest and fastest growing minority population in the United States, where in 2017 was equivalent to 18.1% of the total population and is expected that this number will rise to 35% by 2050[6]. Here, we refer as ancestry to women's origin background measured through ancestry-informative markers or self-reported informa-tion. Likewise, Hispanic/Latino population refers to BC patients from Mexico, Caribbean, Central and South America that share similar admixtures of Native American (Mexican and Mesoa-merican indigenous population), European, and African ancestries[7]. In particular, Mexico represents the largest source of Hispanic/Latino diversity, therefore, Hispanic-Mexican (HM) patients make up an important proportion of Latino BC cases[8].

Mexico is experiencing a demographic, epidemiological and nutritional transition, favoring the exposure to risk factors for cancer, such as aging, smoking, alcohol consumption, and high prevalence of obesity and diabetes. Indeed, landmark epidemio-logical and clinical studies revealed a significant higher mortality/incidence ratio in low-income and middle-income countries than in high-income countries[9], showing a relevant geographic dis-parities. Poor outcomes in Mexican patients may be explained by late detection and limited access to health care[10]. Nevertheless, other biological factors, specific to the Mexican population, such as tumor biology dictated by genomic alterations and molecular factors might impact cancer development. Thus, these features need to be addressed as an important source of tumor etiology and evolution. However, despite the fact that different large efforts in cancer genomics has been conducted in the past years, the genomic alterations of BC in HM and Latino populations remain poorly characterized, since most of these discoveries and exploratory studies have focused on data obtained almost pre-dominantly from Caucasian populations[11]. Given this scenario, a more detailed description of the biological landscape of breast tumors in HM women is still warranted to gain major

understanding on the genetics and molecular factors operating at the basis of BC, especially, among under-represented populations. In this study, we evaluated somatic mutations, SCNA and gene expression patterns on 204 tumors from HM women and we comparatively describe their genomic context in contrast to patients with African, African-American (AA), Asian and Cau-casian (European descent) ancestry. To the best of our knowl-edge, this dataset represents the largest breast cancer genomics characterization of breast tumors in HM women, and, the results from this work highlights unique molecular features of HM breast cancers, as well as, characteristics common to all BC cases.

## Results

**Overview of the HM-profiled tumors**. A comprehensive mole-cular analysis of HM tumors was performed as following: whole-exome sequencing (WES, $n = 134$) to define the somatic mutation landscape, messenger RNA (mRNA) high-throughput microarray profiling ($n = 109$) to evaluate molecular intrinsic subtypes and gene expression portraits, and finally DNA copy-number profil-ing through genome-wide arrays (SNP6 Affymetrix arrays) ($n = 78$ tumors and matched normal blood sample) (Fig. 1a and Supplementary Fig. 1a). Clinical information of analyzed patients is shown in Supplementary Data 1. Mexican ancestry hetero-geneity is the consequence of the admixture of Native-Mexican (Mexican indigenous population), European and African populations[7]. As expected, among patients with genotyping information in our dataset, Native-Mexican genetic ancestry was the leading component with 60% contribution, while the Eur-opean component contributes 34% and just a small contribution of African and Asian genetic ancestry was determined ranging from 5% to 1%, respectively, (Supplementary Fig. 1b, c and Supplementary Data 1).

**Distribution of intrinsic BC subtypes in HM and Hispanics non-Mexican BC samples**. To explore potential differences related with ancestry in tumor biology, we set out to analyze molecular subtypes in tumors collected in different populations. We interrogated molecular architecture based on immu-nochemistry (IHC) markers information and intrinsic molecular subtypes in order to understand the composition of BC at clini-cally and RNA-based subtypes levels in the HM, Hispanic and non-Hispanic women. In tumor samples for which there is available IHC information we only observed a difference in the enrichment of triple-negative (TN) subtype with respect to AA (38% vs. 14% HM-pooling our profiled samples with those deposited in GSE75678 dataset from Monterrey, Mexico; and 25% in Hispanic non-Mexican, BH $p < 0.05$) and Nigerian (43%, BH $p < 0.05$) patients (Fig. 1b and Supplementary Data 2), as previously described in the literature[12]. Classification of tumors based on gene expression data by the Pam50 molecular subtypes are not conventionally reported for Mexican patients with breast cancer. Thus, we then assessed PAM50 subtypes by defining the intrinsic clusters as described by Perou and Parker[13] for BC samples in our in-house-profiled dataset and other publicly available gene expression data (Fig. 1c). PAM50 classification identified four different tumor subtypes (Supplementary Fig. 2a) and normal-like samples, which were discarded due to possible normal cell majority content. By assessing the concordance between the two methods employed to classify samples, we found high (~70%) overlapping between IHC-defined subtypes (based on St. Gallen guidelines[14]) and RNA-based PAM50 centroids intrinsic subtypes (Supplementary Data 2 and Supplementary Fig. 2b, c).

Of note, when integrating our HM women with independent HM patients from Monterrey, Mexico (GSE75678, $n = 53$) into a single dataset ($N = 162$) we observed that HM series contained

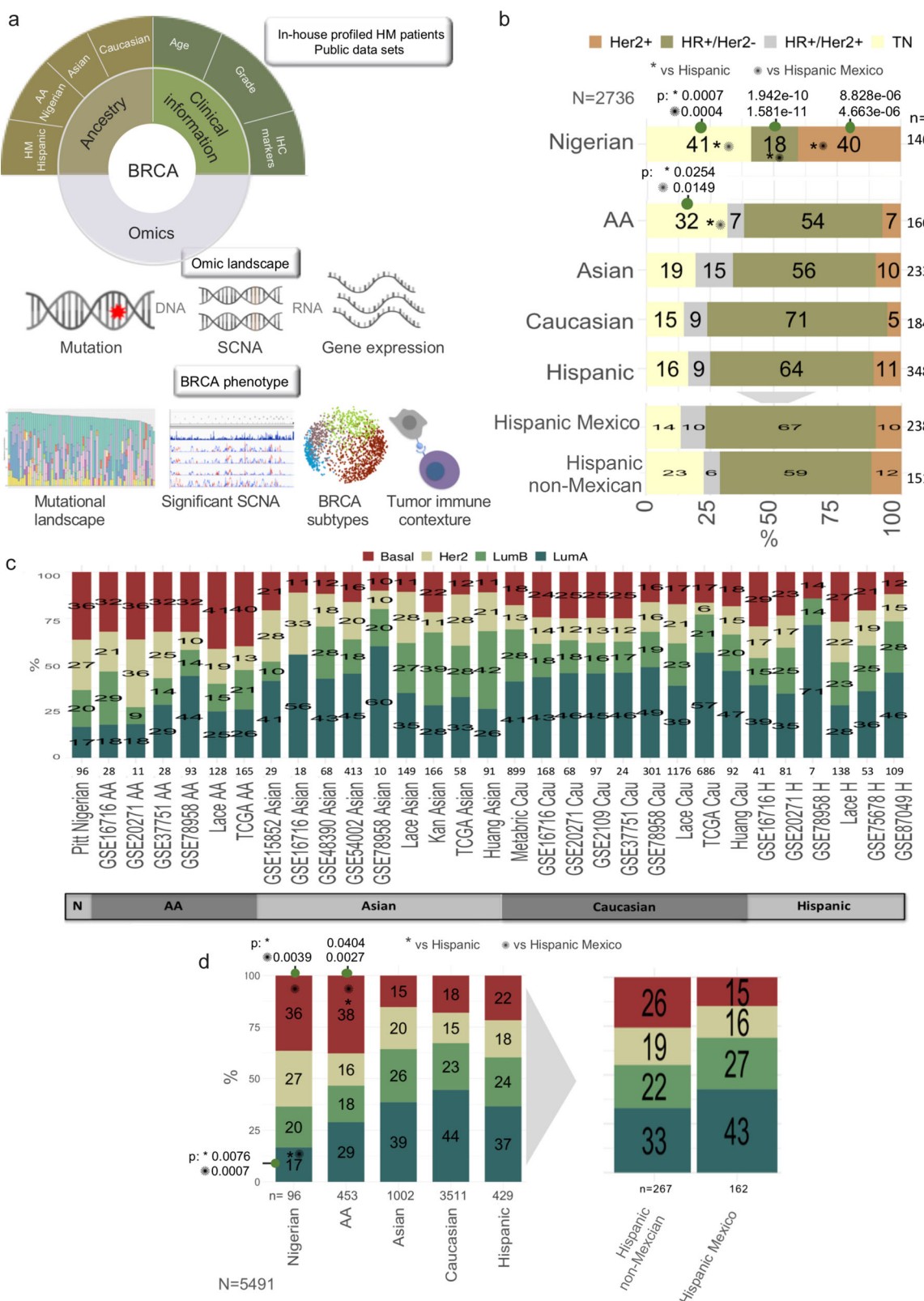

43% LumA tumors, 27% LumB, 16% HER2-enriched and 15% Basal-like (Fig. 1d). Compared to other biospecimen collections including Hispanic patients ($N = 313$) from M.D. Anderson Cancer Center (GSE16716, $N = 41$, Basal-like: 23%), Instituto Nacional de Enfermedades Neoplasicas in Lima, Peru together with the Centro Medico Nacional de Occidente in Guadalajara, Mexico (GSE20271, $N = 81$, Basal-like: 29%), Hospital San Jose

Tec de Monterrey, Mexico (GSE75678, $N = 53$, Basal-like: 22%) and LACE and Pathways cohorts from USA ($N = 138$, Basal-like:27%), our dataset presented a smaller proportion of Basal-like tumors (12% vs. 26% -media proportions of other studies-, Fisher's exact Benjamini–Hochberg, BH, $p = 0.041$) (Fig. 1c and Supplementary Data 2) and a significant reduced odds of having this subtype (OR HM dataset vs. Hispanics: 0.522, 95% CI:

**Fig. 1 "Omic" characterization of multi-ancestry breast cancer molecular profiles. a** Graphical workflow of breast cancer molecular characterization of in-house-profiled HM women and in-silico benchmarked data including Hispanic (Peruvian and US Latina women), African-American, African (Nigerian), Asian and Caucasian breast cancer patients. Briefly, genomic and transcriptomic data were analyzed to get a deep biological landscape describing the mutational and copy-number alterations, as well as gene expression profiling of breast cancer. Each molecular platform was then integrated to get a more robust oncogenic picture of breast tumors and their similarities and differences between ancestries. **b** Frequency of immunohistochemistry subtypes routinely evaluated: hormone receptors (HR-estrogen and progesterone receptor) and HER2 markers across ancestry groups. TN: Triple-negative. **c** Frequency of PAM50 intrinsic molecular subtype in each breast cancer dataset ($N = 5418$), including in-house-profiled Mexican patients (GSE87049 $n = 109$) and public available data. **d** Median frequency of PAM50 intrinsic molecular subtypes among ancestries. HM: our profiled tumors GSE87049 integrated with GSE75678 Mexican tumors. Hispanic non-Mexican: Hispanic tumors from GSE16716, GSE20271, GSE78958 and LACE and Pathways cohorts from USA. Hispanics: average value between HM and Hispanics non-Mexican. Two-sided Fisher's exact test $p$-values with Benjamini–Hochberg (BH) FDR correction were computed for statistical comparisons (BH $p$-value <0.05 * vs. all Hispanic patients and, ✱ vs. integrated Hispanic Mexican patients). p: BH-adjusted $p$-value. H: Hispanic, HM: Hispanic-Mexican, N: Nigerian, AA: African-American, Cau: Caucasian. LumA: Luminal A, LumB: Luminal B, SCNA: Somatic Copy-Number Alteration, IHC: Immunohistochemistry.

0.315–0.866). This differences in subtypes frequency among different Hispanic groups could be partly ascribable to differences in the number of tumors evaluated, batch effects of the collection or to different admixture patterns within these cases.

As expected, we find some particular differences at PAM50 subtype level among population groups evaluated (Fig. 1c, d). LumA subtype has a higher prevalence among HM women than in AA and Nigerians (HM vs. AA: 43% vs. 26% Fisher's exact BH $p = 0.041$, HM vs. Nigerian: 43% vs. 17%. BH $p = < 0.001$). Further, basal-like subtype is clearly less represented in HM patients with respect to AA and Nigerian (OR AA vs. HM: 3.6, 95% CI: 2.2–5.8, BH $p = 0.001$; Nigerian vs. HM: 3.3, 95% CI: 1.8–6 BH $p = 0.002$) (Fig. 1d and Supplementary Data 2). Among the other subtypes, there was a substantial overlap in the frequencies between HM, Hispanic non-Mexican and non-Hispanic patients and differences did not reach statistical significance (Fig. 1d).

Recently, diverse studies have reported a higher proportion of women diagnosed with BC under 45 years in Latin America (20%) compared with their counterparts in USA and Europe (12%). Further, incidence rates in Mexico show that women with breast cancer are diagnosed on average 11 years earlier than in Caucasian women (51 years vs. 62 years)[15]. Thus, we interrogated if age at diagnosis was associated with ancestry background and BC subtypes in our dataset. We identified that HM (integrated set) and Hispanic non-Mexican women, had a significantly higher probability to present BC at a younger age (<45 years of age) than their Caucasian counterpart (OR HM vs. Caucasian: 1.5, 95% CI: 1–2.2, BH $p = 0.045$; Hispanic vs. Caucasian: 2, 95% CI: 1.4–2.9). Similarly, a higher rate of younger HM and Hispanic women with BC was found in comparison to Caucasian patients (HM: 28%, Hispanic non-Mexican 39% and 17% in Caucasian, Fisher BH $p < 0.1$) (Fig. 2a, Supplementary Fig. 3a, and Supplementary Data 2). Regarding the comparison among all Hispanic patients, we did not detect any significant difference between HM, Peruvian and US-Latinas women (Fig. 2a). Aggressive hormone receptor (HR) + /Her2+ and TN subtypes are significantly enriched in Hispanic young women (≤45 years of age) in comparison with AA, Asian and/or Caucasian patients analyzed (BH Fisher $p < 0.05$) (Fig. 2b and Supplementary Fig. 3b). In line with our findings, in a different sample collection from our group ($N = 97$, GSE86948), we had similarly observed an early-onsets in 37% of Mexican patients with triple-negative tumors (BH vs. non-Hispanic datasets $p < 0.05$). Moreover, in samples with available gene expression and age at diagnosis information, we observed an age-related disparity on the LumB and the aggressive Basal-like molecular subtypes, with an enrichment of HM and Hispanic non-Mexican women diagnosed at early-age (Fisher BH $p = <0.05$), compared with patients from non-Hispanic ancestries (Fig. 2c, Supplementary Fig. 3c, and Supplementary Data 2).

**TMB and cancer driver mutations across BC patients from multi-ancestry profiles.** Mutational load defined as the tumor mutational burden (TMB, mutation/Mb) is related with mutation rates stablished by intrinsic and non-intrinsic factors required for cancer development and might differ among subtypes or individuals from different ancestries, hence, TMB was computed for samples with available exome sequencing data in each sample collection, including 134 HM, 119 AA, 684 Caucasian, 185 Asian (Kan et al.[16]), 57 Asian-TCGA and 250 Japanese (Hatakeyama et al.[17]). Median TMB of tumors from HM-sequenced patients (median TMB:0.99, range:0.18–16.10) was significantly higher than those from Caucasian patients in TCGA (median TMB: 0.72, range:0–115.32, FDR, $p = <0.001$) and those samples from AA (median TMB of 0.86 mut/Mb, range: 0.06–3.67, FDR $p = 0.04$), but showed no significant difference with Asian patients in TCGA (median TMB: 0.92, range: 0.32–13.66) nor with Japanese patients (median TMB 1.08, range:0.06–18.16) (Fig. 3a and Supplementary Data 3). In contrast, HM women had a lower TMB than tumors from young Asian women (Kan et al.[16] median TMB: 1.39794, range: 0.47712–2.49969 FDR $p = 0.0021$), which exhibit higher mutation rates (Fig. 3a and Supplementary Data 3). After stratifying for HR and HER2 status, we only observed differences in median TMB on HR + /HER2- HM tumors (adj $p < 0.05$), which had higher values than Asian and Caucasian individuals included in TCGA (Supplementary Fig. 4a–d and Supplementary Data 3). Hypermutated tumors having a TMB value equal or higher to 4 occurred in 4% of all breast cancer samples evaluated (Supplementary Fig. 4e). The frequency of hypermutation was higher in HM women compared to AA and Asian Kan[16] women (8% vs. AA, BH $p < 0.05$). Additionally, although not significant, HM individuals were enriched in hypermutated phenotype in comparison to other groups (8% in HM patients, 5% in Asian Japanese carcinomas, 4% in Caucasian and 3% in Asian TCGA). Differences in the number of hypermutated tumors might be ascribed to different mutational process (Fig. 3a).

Of note, when samples are grouped by age, HM patients analyzed did not present statistically significant difference in median TMB values between younger (<45 years old) and older patients, as occurs in tumors from Caucasian and Asian ancestry (TCGA and Japanese dataset) (Fig. 3b) (adjusted $p < 0.05$). Nor we observed a significant correlation between somatic mutation burden and age in our HM tumor series (Supplementary Fig. 4f).

**Deciphering mutational signatures.** Although TMB is one of the main factors that characterizes cancer genomes and a potential marker of risk assessment, there are also particular mutational processes that generate unique signatures and, along with TMB, constitute emergent informative features of cancer. Therefore, we investigated which mutational signatures operate in HM tumors

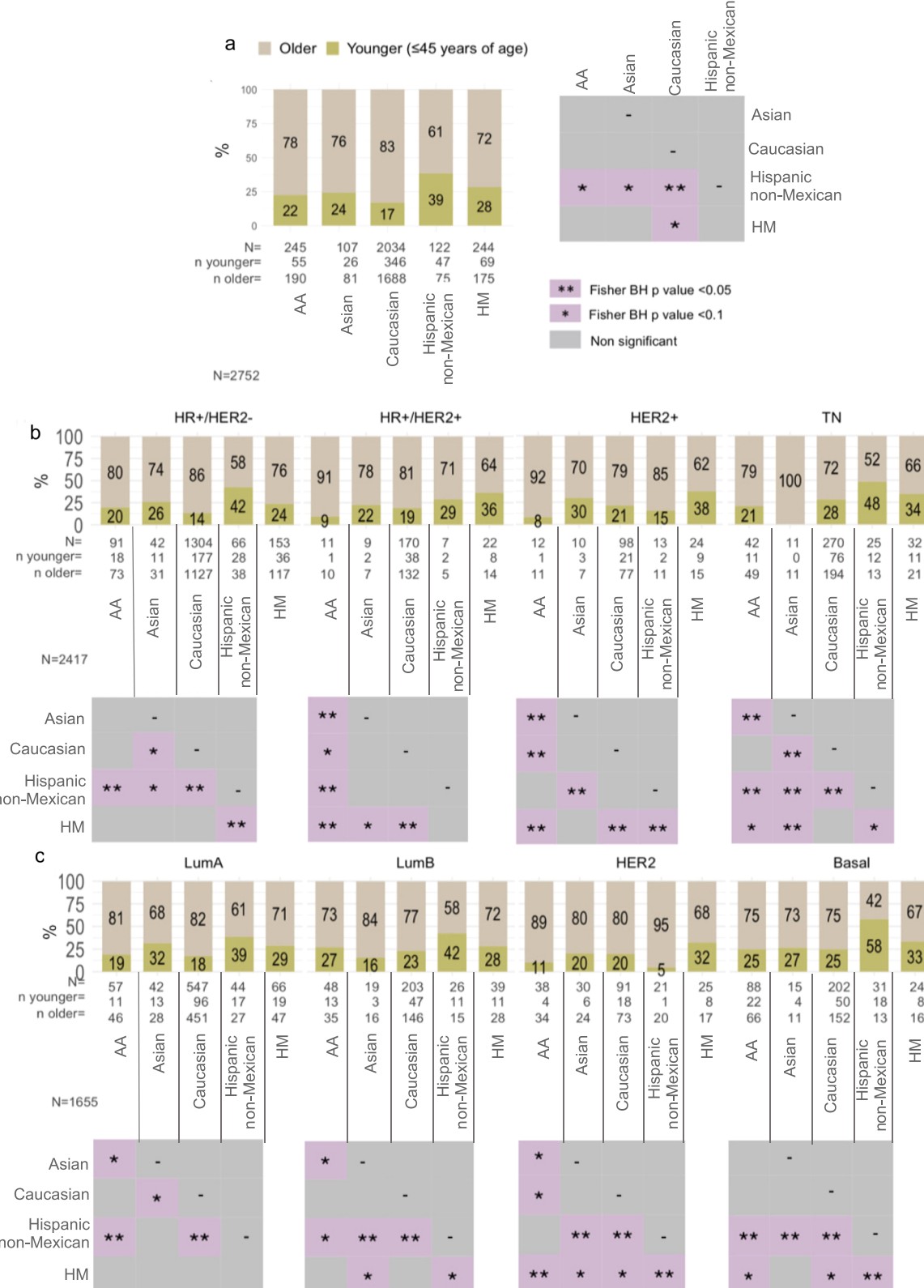

**Fig. 2 Age distribution in BC samples from diverse human populations and within IHC and molecular intrinsic subtypes. a** Frequency of younger (≤45 years of age) and elderly (>45 years of age) BC patients among ancestries. Frequency of BC **b** immunochemistry subtypes, and **c** PAM50 intrinsic molecular subtypes in patients from different ancestry diagnosis at early-age (≤45 years of age) or elderly-age (>45 years of age). Barplots represent proportion of age classes in each population group, while heatmaps represent the BH-adjusted *p*-values computed by a two-sided Fisher's exact test from multiple comparisons. Corresponding *p*-values are reported on Supplementary Data 2. HM: our profiled tumors integrated with GSE75678 Mexican tumors. Hispanic non-Mexican (H nM): Hispanic tumors from GSE16716, GSE78958, and GSE20271. **adjusted *p*-value < 0.05, *<0.1. HM: Hispanic-Mexican, H nM: Hispanic non-Mexican, AA: African-American, HR: Hormonal receptors (ER and PR).

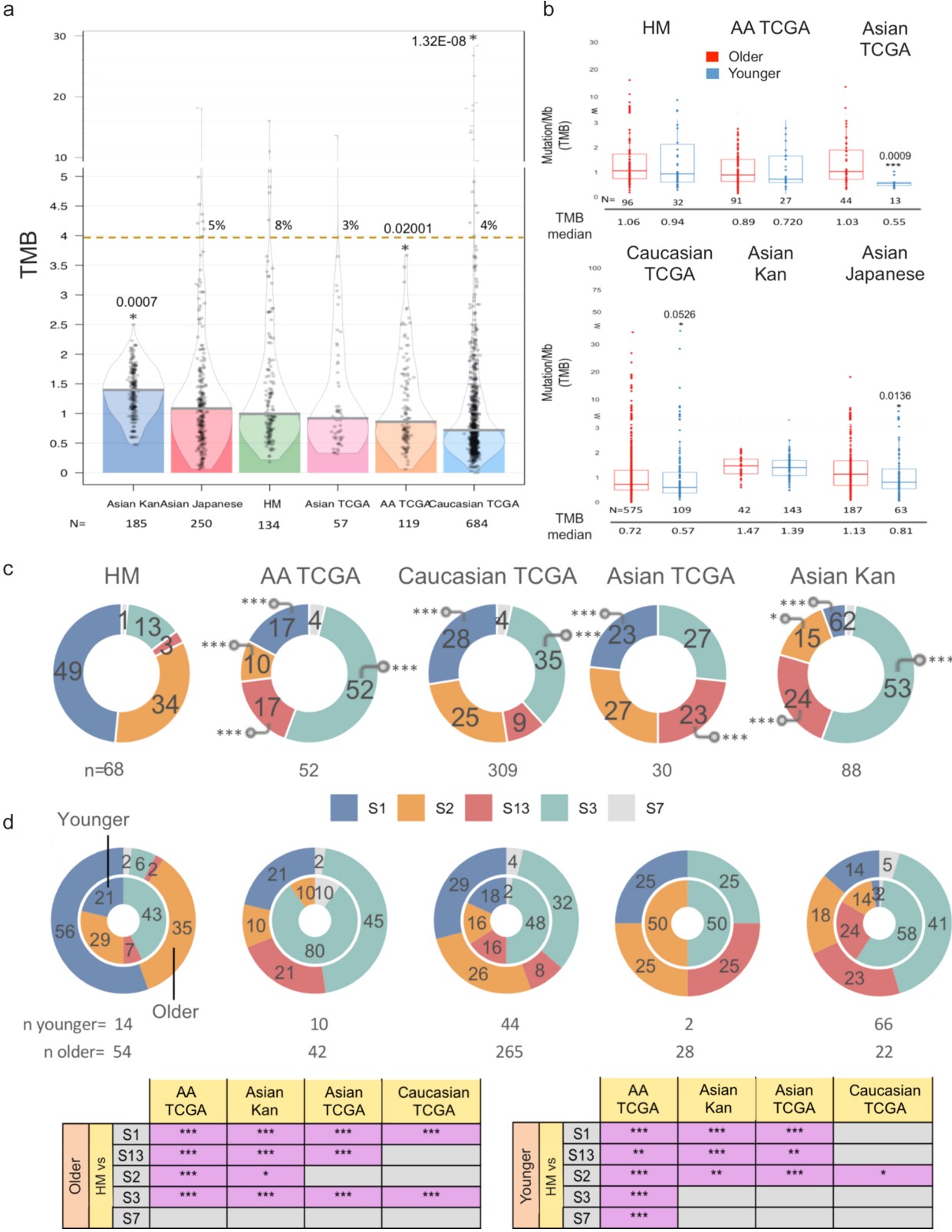

and their association with age and subtype prevalence. To gain further insights into these operative mutational processes, the contribution of mutational signatures were delineated using deconstructSigs[18] and SigFit[19] tools based on single nucleotide variations (SNV) in tumors harboring a TMB above the median value for each tumor collection ($n$ = HM 69, 52 AA, 30 Asian, 323 Caucasian, Asian Kan 94). The mutational landscape of each

signature deconstructed by the two algorithms showed a highly concordant result (Supplementary Fig. 5a). Most individual cancer exomes exhibit more than one mutational signature (Sig) and many different combinations of signatures were observed among BC tumors, across all populations (Supplementary Fig. 5b). In general, 8 signatures were robustly detected among BC tumors from the 30 COSMIC signatures v2[20], but it seems

**Fig. 3 Tumor mutational burden and tumors and mutational signatures identified in human breast cancers across ancestries. a** Violin plots showing the TMB (mut/Mb) distribution in each ancestry-group considering point non-silent mutation. Dot line represents hypermutated threshold (4mut/Mb) and the number indicate the percentage of hypermutated tumors in each dataset. Inserted barplots represent the median value, gray dots represent individual patient data. **b** Boxplot of TMB described in the evaluated datasets dividing patients population in younger (≤45 years of age) and older (>45 years of age). **c** Normalized proportions of the five most frequent COSMIC v2 trinucleotide mutational signatures in tumors from HM, AA, Caucasian, Asian TCGA and Asian Kan harboring a TMB equal or over the median in each dataset. Mutational signatures contributions to each individual sample were depicted by DeconstructSigs algorithm. **d** Donut plot reports the percentage of tumor samples harboring a particular predominant driving signature in younger (≤45 years of age–Inner donut chart) and elderly patients (>45 years of age outer donut chart). Bottom panel described statistical analysis of mutational signatures prevalence in each of the interrogated datasets. On panel **a** and **b**, boxplots represent median ± IQR (25th and 75th percentile) and whiskers correspond to maximum and minimum values. Statistical comparisons were assessed with a two-tailed Wilcoxon test considering HM dataset as reference. On panel **b** $p$-values are indicated near the corresponding asterisks. On panel **c** and **d** a two-sided Fisher's exact test was computed. $p$-values with BH FDR correction were computed for statistical comparisons. *(BH-adjusted $p$-value < 0.05). Corresponding $p$-values are reported on Supplementary Data 4. HM: Hispanic-Mexican, AA: African-American, S: signature.

that some tumors have a more complex repertoire of mutational processes than others (Supplementary Fig. 5b, Supplementary Data 4). Top frequently mutated signatures among patients from different ancestry deconstructed with both algorithms included: Sig1, Sig2, Sig3, Sig13 and Sig7 (Fig. 3c, Supplementary Fig. 5c, d, and Supplementary Data 4). Of note, 96% of the evaluated tumors presented one of these top signatures, while the remaining 4% did not present any of them.

The signature with the highest mutational probability was considered as the predominant driving signature in each sample (Fig. 3c and Supplementary Data 4). Overall, in our HM dataset, Signature 1, associated with cytosine deamination mutational processes and clock-like properties, was the most common predominant mutational signature contributing in average with 49% of the top mutational signatures (Fig. 3c). Sig1 is characterized by prominence of C > T changes (Supplementary Fig. 6a) and enriched distribution of hormone receptors markers (Supplementary Fig. 6b). HM tumors from young patients presented a higher Sig 1 prevalence than AA TCGA and Asian Kan individuals (HM 25% vs. 0% in AA and 3% in Asian Kan BH $p < 0.005$) (Fig. 3d). Moreover, mutations contributed by signature 1 showed a greater rate in HM older patients (>45 years old), which accounted for the highest frequency in comparison with other patients with different ancestries (HM 51% vs. 20% in AA, 28% in Caucasian, 31 in Asian TCGA and 14% in Asian Kan BH $p < 0.05$) (Fig. 3d). Overall, these data suggest the possibility that the chronological age does not completely recapitulate the biological age of tumors in the evaluated HM patients. Signature 2 was the next most common predominant mutational signature (31%) (Fig. 3c). Since etiologies of signature 2 and 13 are attributed to a common activity of the AID/APOBEC family, we collapsed these signatures into a single one (Signature 2/13)[21]. Signature 2/13 is characterized primarily by C > T and C > G mutations (Supplementary Fig. 6a) and is enriched in HR + /HER2− phenotype, while in the other populations evaluated a smaller percentage of this phenotype was observed in APOBEC-related signatures (Supplementary Fig. 6b). Tumors belonging to this signature present an overwhelming number of mutations in contrast to the other signatures (Supplementary Fig. 6c). On detail, HM tumors with high contribution of this signature were the most mutated tumors among AA, Caucasian and Asian Kan evaluated breast cancer cases (Wilcoxon BH $p < 0.05$) (Supplementary Fig. 6c). Signature 3, characterized by a BRCAness phenotype, was the third most frequent processes in our tumor collection (18%). This signature was preferentially enriched in triple-negative phenotype (Supplementary Fig. 6b). Finally, signature 7, resulting from DNA adducts formed by the action of UV light, were observed in 1% of HM patients. Overall, these results suggest that the studied HM population, shows distinctive spectrum of mutational signatures contributions that make up their mutational landscape.

**Mutational landscape in HM BC tumors.** We have observed that specific mutational signatures are particularly enriched among the analyzed HM tumors with specific molecular and demographic features and mutational burdens. Therefore, we hypothesized that some known somatic mutations in BC might be present in HM tumors but at a different frequency. To understand somatic mutations prevalence among HM women, we analyzed the non-silent mutations affecting protein-coding genes in these samples ($N = 134$) by computing MutSigCV[22] and compared them against patients from non-Hispanic ancestry. The significantly mutated cancer genes ($q$-value = <0.1) presented in more than 5% of HM individuals are *PIK3CA* (28%), *TP53* (20%), *AKT* (8%), and *MAP3K1* (5%) (Fig. 4a) (Supplementary Data 5). These genes have been reported by other studies as mutated in BC[16, 23, 24] (Supplementary Fig. 7a). Nonetheless, mutations in *CDH1* (2%) occur at a much lower frequency, while *AKT1* mutations have a higher occurrence (8%) in evaluated HM women (Supplementary Fig. 7a).

Of note, the Glu17Lys (E17K) mutation within the PHb domain of *AKT1*, was present in 10 out of the 11 tumors (8%) harboring *AKT1* mutations (Fig. 4a, b and Supplementary Data 5) and are predominantly HR + tumors (Fig. 4c). We found no evidence for *AKT1*[E17K] significant mutation (MutSigCV, $Q < 0.1$ and frequency >5%) in 4464 samples of non-Hispanic ancestry (Supplementary Fig. 7a–c), except for a group of young Korean patients (ICGC, breast cancer - very young women, BRCA-KR) where 8% of cases ($N = 4/50$) were mutated. E17K mutation results in a pathogenic activating mutation according to OncoKb, which in turn promotes an active PI3K/AKT/mTOR pathway signaling (Fig. 4d and Supplementary Fig. 7d). As a member of the PI3K/AKT/mTOR axis, only one *AKT1*[E17k] mutant sample (10%) is co-mutated in *PIK3CA*, thus the altered activation of this oncogenic signaling generally occur by any one of these means in an exclusively manner. Additionally, 40% of *AKT1*[E17k] mutant patients had no additional driver cancer mutations. While 60% had further alterations, other than *AKT1* mutation, which likely contribute to cancer development (e.g., truncating mutation in tumor suppressor or gain of oncogenic mutations) (Fig. 4d).

Other potentially novel mutated genes in the HM tumors exhibiting significant mutation prevalence (≥3.0%), non-previously reported in other datasets (Supplementary Data 5), includes *MRPL37* and *SLC16A8* ($q$-value = 2.72E-02 and 1.36E-02, respectively). Although, these novel mutations are predicted to be passenger alterations, most of them have a high deleterious oncogenic capacity and a functional consequence of missense or nonsense change (Supplementary Fig. 7e).

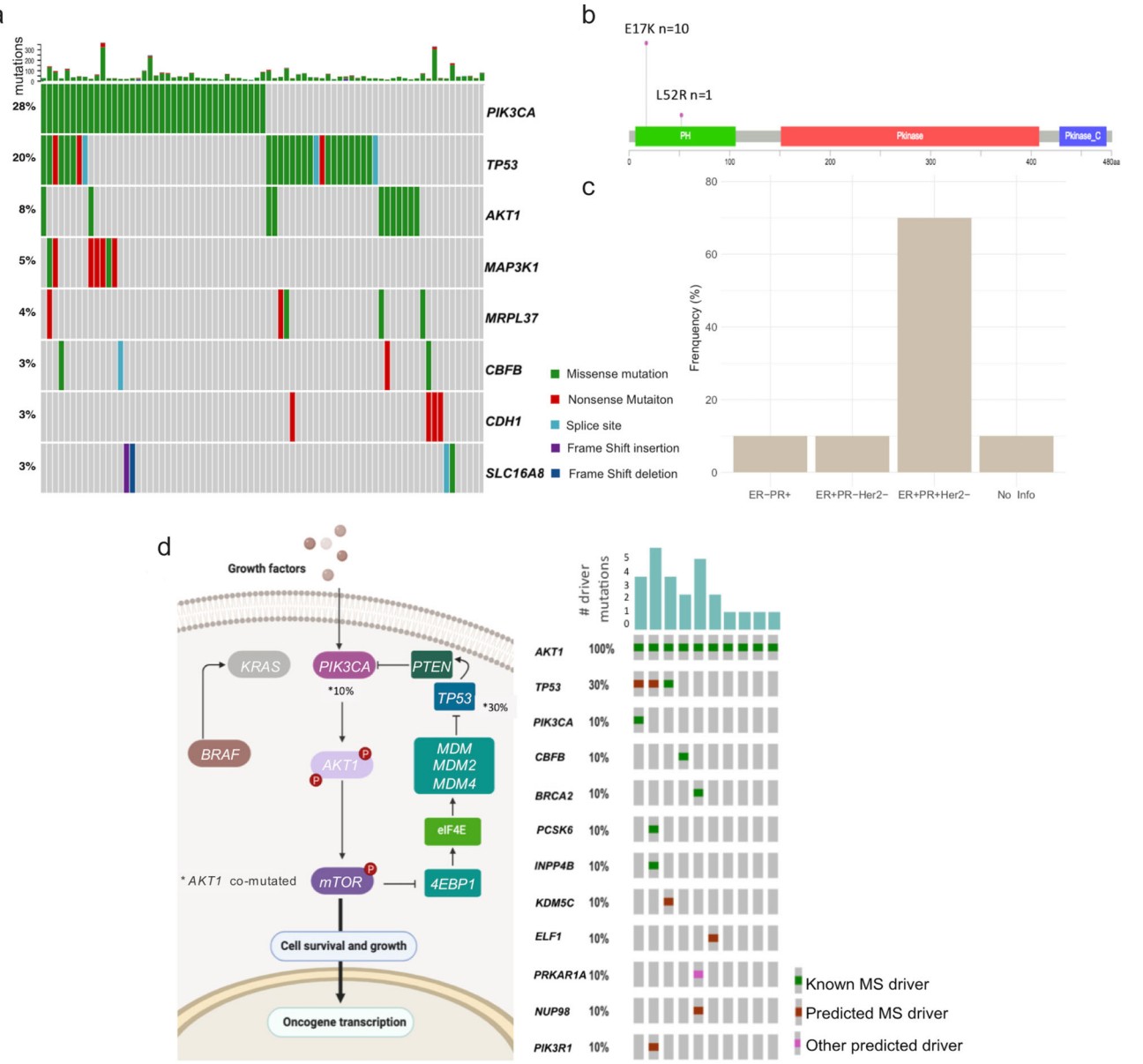

**Fig. 4 Mutational panorama of HM breast cancer tumors. a** Oncoplot of significantly non-silent mutated genes in HM tumors computed by MutSig (*q*-value < 0.1). The heatmap represents individual mutations in patient samples, color-coded by type of mutation as illustrated by the figure legend. Percentages refers to the fraction of tumors with at least one mutation in the specified gene. *p*-values were determined by testing if the observed mutations in a gene significantly exceed the expected background model. *p*-values were adjusted by false-discovery rates, only top 10 genes with adjusted *p*-value ≤ 0.1 were reported. **b** Lollipop plots of non-silent mutations detected in AKT1 gene in HM women and their distribution in the body gen. **c** Proportion of tumor samples in HM dataset, separated according to their immunochemical classification, harboring E17K-*AKT1* mutation. **d** Fraction of mutational alterations presented in the PIK3CA/AKT/mTOR axis (left panel, % of samples co-mutated with AKT) visualized with PathwayMapper and co-mutated driver alterations in AKT mutated tumors (right panel MS = Missense).

**Driver and clinically actionable genomic alterations in HM population**. Up to 78% of all HM tumors contain at least one driver point mutation, with an average of 2.65 driver mutation, in agreement to the rates computed for other ancestries[25] (Fig. 5a and Supplementary Data 6). The numbers of point driver mutations remain remarkably stable, even in highly mutated tumors (>1 mut/Mb) (Fig. 5b). Moreover, an important fraction of identified driver mutations is annotated as oncogenic (19%) or tumor suppressor alterations (21%) in HM tumors (Fig. 5c). Finally, the frequency of the top mutated cancer driver genes (>5%) in HM breast tumors varies among the evaluated ancestries, with *PIK3CA, MAP3K1* and *PTEN* having the highest similarities (Fig. 5d and Supplementary Data 6).

Subsequently, we analyzed the potential clinical implications of the mutation profiles, thus, we evaluated the frequency of clinically actionable mutations by retrieving annotations of targetable genomic alterations using the cancer hotspot database[26] and OncoKB classification system[27]. Forty-six percent of the HM tumors exhibited hotspot mutations, particularly in *PICK3CA* (H1047L, E542K, E545K, H1047L) and in *AKT1* (E17K) genes (Fig. 5e). When including also other potentially actionable variants other than hotspot mutations, 77% of the HM tumors analyzed harbored oncogenic alterations in 46 genes considered potentially targetable based on various clinical and preclinical evidence (Fig. 5f). 25% of them have either likely oncogenic or oncogenic status as reported by OncoKB (Fig. 5g).

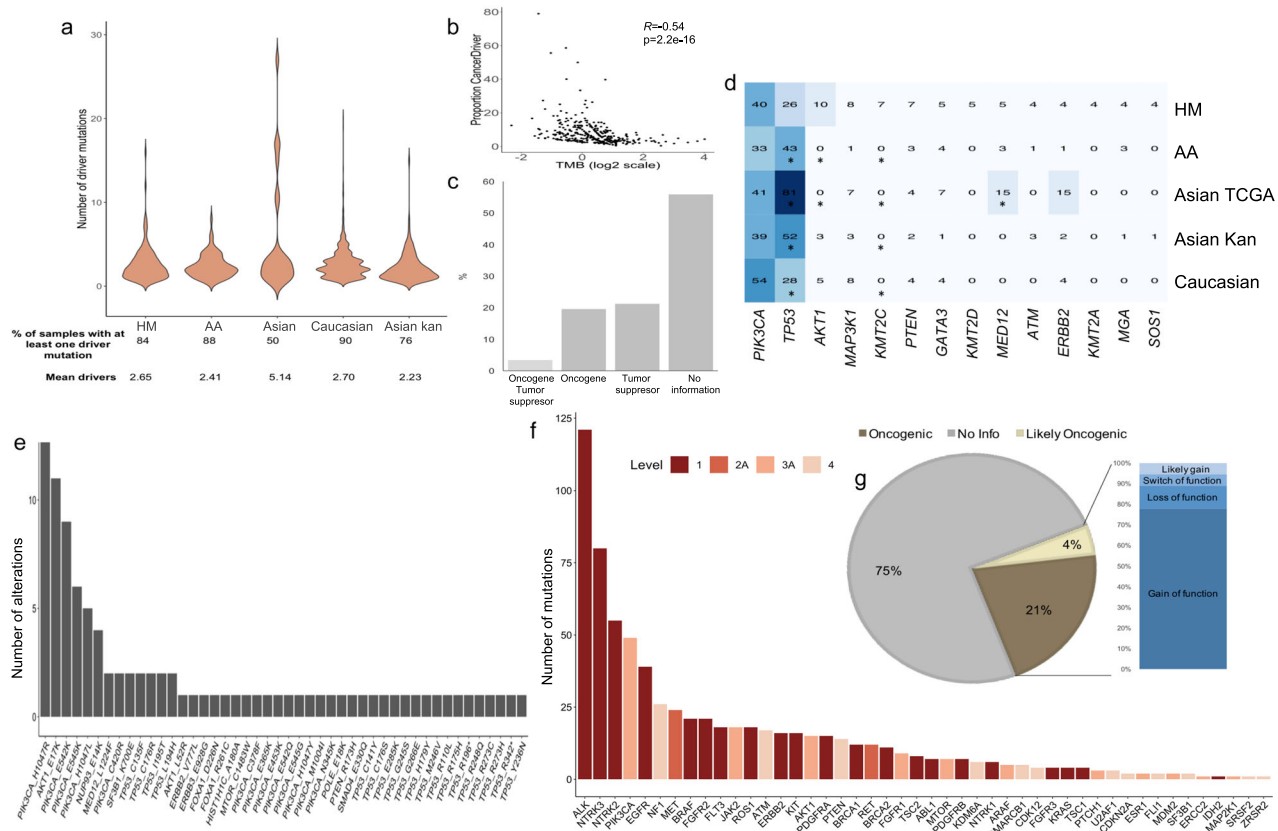

**Fig. 5 Mutational driver and actionable mutations in breast cancer samples from diverse ancestries. a** Violin plot showing the total number of known or predicted cancer driver mutations in each study. **b** Barplot showing the fraction of oncogenes or tumor suppressor genes among the known-driver or predicted driver-mutation in HM tumors. **c** Correlation between the proportion of driver mutations in tumors and their mutational burden (in logarithmic scale) presented as a scatter plot. Correlation coefficient computed using the Spearman method. **d** Most recurrent driver point mutation in the HM women. Heatmap showing the frequency mutation events in highly mutated genes across different tumor datasets. Color scale, from white to dark-blue, represents the percentage of events, which is also indicated inside each cell. Two-sided Fisher's exact test $p$-values with BH FDR correction were computed for statistical comparisons of panel **d**. Corresponding BH-adjusted $p$-values are reported on Supplementary Data 6. **e** Total number of hotspots mutations in HM breast cancer tumors. **f** Total number of mutations in genes annotated as actionable by OncoKb, classified on the basis of gene level evidence (i.e., 1 FDA drug approved, 2A FDA-approved standard care, 3A Compelling clinical evidence and 4 Compelling preclinical level) in HM evaluated samples. **g** Pie chart represents the frequency of tumors harboring actionable mutations categorized as oncogenic or likely oncogenic. The barplot on the right shows the frequency of oncogenic mutations, split according to their impact on protein function, that are present in the 25% of HM tumor samples. BH-adjusted $p$-value *<0.05. HM: Hispanic-Mexican, AA: African-American.

Some relevant actionable mutations include activation of *ERBB2* detected in four samples lacking *HER2* amplification, suggesting that these patients could benefit from anti-HER2 therapy such as Neratinib and Ado-Trastuzumab. Similarly, we found tumors harboring *AKT1*^E17K driver mutation, considered actionable, that may benefit from anti-AKT therapy (Supplementary Fig. 8).

**Analysis of somatic copy-number alterations in BC.** Based on our previous findings, we identified tumors that presented low mutated burden and did not harbor driver mutations, significantly mutated genes or high contributions of mutational signatures, suggesting that other events are contributing to their cancer genomic landscape. SCNAs are relevant alterations affecting larger fraction of the cancer genome[28]. In this way, to understand how SCNAs can also account for the biological features particularly observed in the HM women, we tested the hypothesis that there are some SCNA with a different frequency in HM with respect to other groups with a different ancestry background. GISTIC analysis on HM tumors, reveled relevant SCNA (retained as likely-significant +/−1 or significant +/−2) for a total median number of 9381 recurrent events in 140 regions (5075 gene amplifications and 3622 deletions were detected)

(Fig. 6a). The overall proportion of SCNA is similar to what is observed in the other datasets, exception made for AA tumors, which showed significantly a greater number of SCNA events (Wilcoxon FDR $p = 0.0095$) (Fig. 6a) (Supplementary Data 7). In general, we observed much more amplifications than deletions among all the sample sets evaluated (Supplementary Fig. 9a–c). Similarly, the median fraction of the cancer genome with copy-number changes, termed tumor SCNA burden (TSCB), is similar between the HM, Caucasian and Asian ancestries (HM 23%, Caucasian 23% and Asian 29%) and lower than the observed in the AA tumors (35%) (Fig. 6b). Of notice, we detected not previously reported SCNA events exclusively present in HM patients. Among the most significant, we found the amplification of the region 16p, which harbors genes such as *SNN, LITAF, ZC3H7A, TXNDC11, RMI2* and the oncogene *BCAR4*, implicated in endocrine resistance in human BC cell. Likewise, we detected the 17p amplification, where *SPECC1* gene is located (Fig. 6c, d, Supplementary Fig. 9d, and Supplementary Data 7). Additionally, HM women harbor well-known SCNA of BC, among which gain of chromosomic regions 8q, 11q, and 17q that contain oncogenes such as *MYC, CCND1* and *ERBB2*. Similarly, we detected losses on chromosomes 7q, 8p, 13q, 17p, containing *MLL3, CSMD1,*

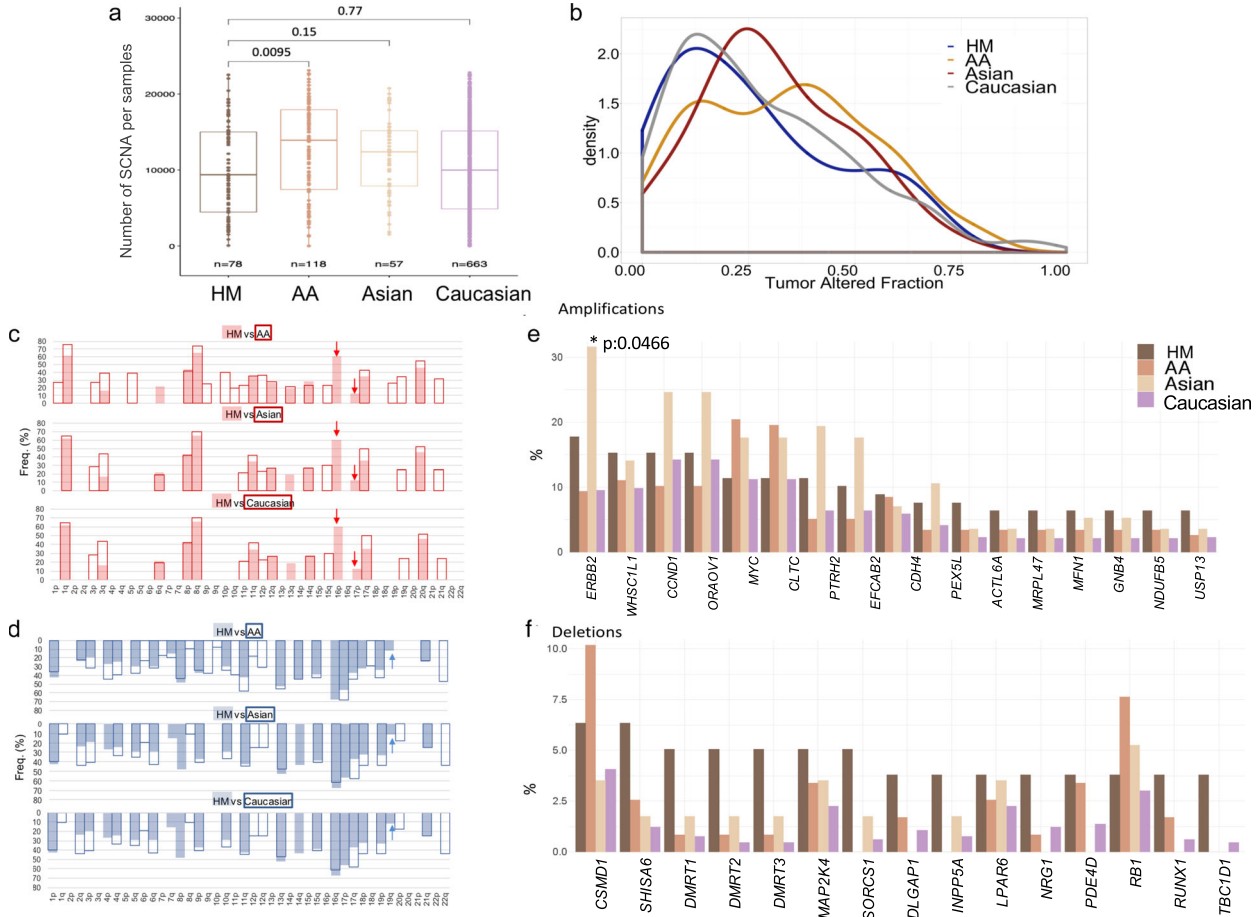

**Fig. 6 SCNAs landscape in breast tumors across ancestries. a** Boxplot shows the total number of SCNAs (including likely-significant (±1) or significant (±2) events) per sample among patients from HM ancestry and TCGA multi-ancestry data. SCNAs analysis was performed on SNP6 Affymetrix arrays data. Statistical comparisons based on two-tailed Wilcoxon test taking HM dataset as reference. Boxplots represent median ± IQR (25th and 75th percentile) and whiskers correspond to maximum and minimum values. **b** Histogram showing the comparative tumor SCNA burden (TSCB) distribution between ancestries. **c**, **d** Histogram showing the frequency of significant SCNAs at arm level across whole genome in different ancestries. Chromosomic regions are arranged on the x-axis. Gains are represented in red, above horizontal line (**c**), while losses are represented in blue, under horizontal line **d**). The comparison of HM (filled bars) and AA, Asian and Caucasian (lines bars) was calculated and plotted separately. Arrows indicate unique HM SCNA arm events. Frequency of the top most significant **e** amplifications and **f** deletions in HM patients, compared against their frequency in other ancestries. Two-sided Fisher's exact test p-values with BH FDR correction were computed for statistical comparisons *(BH-adjusted p-value < 0.05). HM: Hispanic-Mexican, AA: African-American, SCNA: Somatic Copy-Number Alteration.

*RB1* and *MAP2K4* genes. Even though most of the significant SCNA at arm level (q-value > 0.25) are also present in the other groups, they occur at different prevalence in HM patients (Fig. 6c, d and Supplementary Fig. 9c, d). On contrary, some amplifications (3p,11p,15q, 19q, 21q) or deletions (1q,6p,8q,12p/q and 22q) mostly detected in BC tumors from different populations were not identified in our HM dataset (Fig. 6c, d and Supplementary Data 7).

Gene-level amplification/deletion threshold values computed by GISTIC, considering only high-level amplifications (+2) and deep deletions (−2), were used to compare significant events against patients from other ancestries. Collectively, the most frequently altered genes by DNA copy-number alterations at focal level were the amplifications of *ERBB2* (17q17q12), *WHSC1L1* (8p11.23), *CCND1* (11q13.3), *ORAOV1* (11q13.3) and *MYC* (8q24.21) (Fig. 6e and Supplementary Data 8). Recurrent focal copy-number losses included *CSMD1* (8p23.2), *SHISA6* (17p12), and *DMRT1/2/3* genes (9p24.3) (Fig. 6f and Supplementary Data 8) (Supplementary Fig. 9d). Gained or lost regions as identified by GISTIC2 had significant corresponding peaks within the other ancestry groups evaluated.

**Dissecting the biological impact of genomic complexity alterations.** Given that tumor biology is the result of a variety of alterations, we focused our attention on integrating the different genomic features that characterizes HM tumors at mutation, SCNA and gene expression level, and combined them with available information on tumor subtype to dissect the coordinated mechanisms that would extend our comprehension on how these somatic events may impact tumor phenotypes. In our dataset, we confirmed a heterogeneous picture of DNA alterations, represented by correlation between TMB and the tumor altered fraction (TAF-SCNA events). Some tumors exhibited high number of mutations (>1mutation/Mb) or high numbers of copy-number alterations (>5% of tumor altered fraction), but not both (Fig. 7a). Interestingly, we also found tumors that did not present high TMB neither SCNA events, as well as cases exhibiting a relatively high TMB and TAF (Fig. 7a, b and Supplementary Data 9). This last molecular phenotype may be explained by clonal diversity that enhances higher intra-tumor heterogeneity and differences in genomic instability.

SCNAs have critical roles in activating oncogenes and in inactivating tumor suppressors. To determine the "cis"

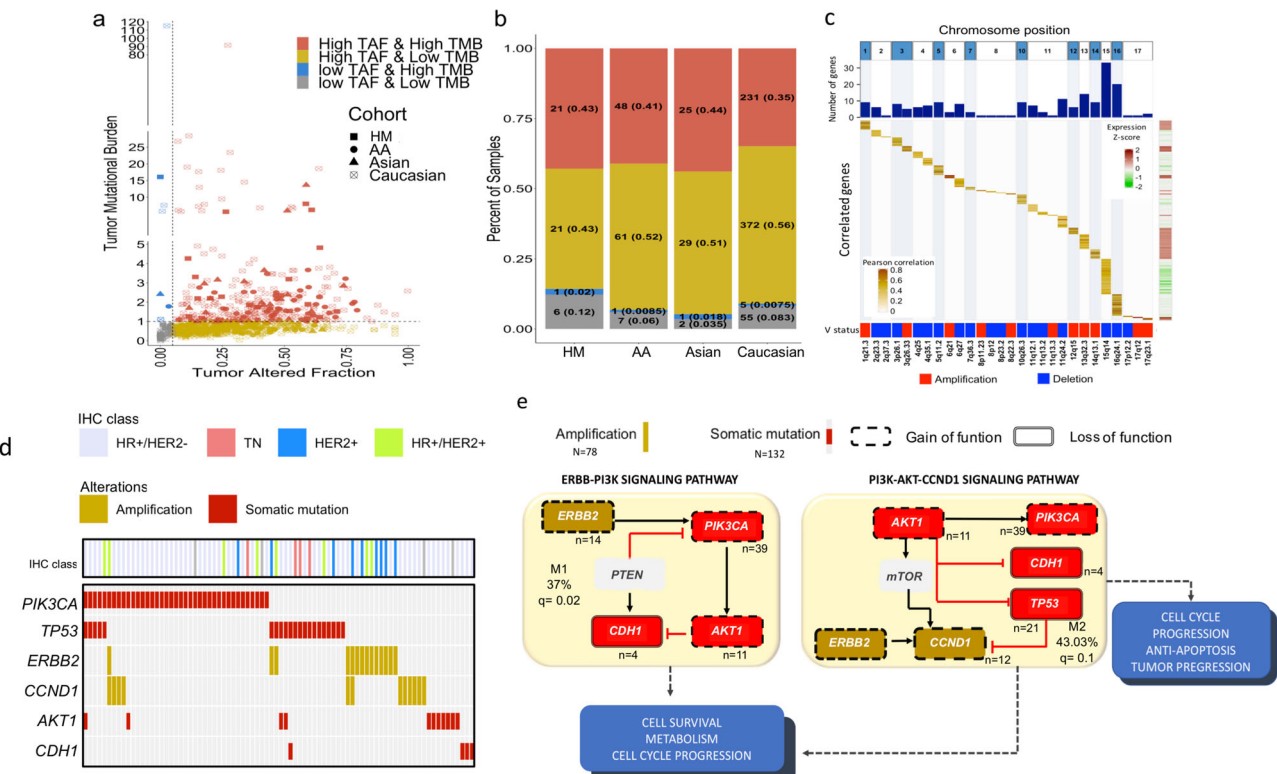

**Fig. 7 Integrative view of genomic and transcriptomic alteration in breast tumors carcinogenesis across ancestry groups. a** Correlation between tumor mutational burden (TMB) and tumor altered fraction (TAF) divided in four subclasses in accordance with the following cutoffs: high TMB > 1 and high TAF > 5%: High-TMB and Low-TAF, Low-TMB and High-TAF, High-TMB and High-TAF, Low-TMB and Low-TAF. **b** Prevalence of each biological group divided by TMB and TAF classes (above mentioned) among ancestry groups. **c** Cis effects of copy-number alterations on mRNA expression in HM. The heatmap shows significant correlated genes (Pearson R = 0.5, BH-adjusted p-value < 0.05) between SCNA and robust gene expression changes (gene expression profiles between altered and non-altered tumors, logFC:1.5, BH-adjusted p-value < 0.05) as illustrated in the sided heat map (over-expression: red, down-expression: green). The diagonal yellow line represents over-expressed genes located in amplified regions or down-regulated ones located in deleted loci. Genes are ordered by chromosome locations on x-axes. The lower heatmap shows the amplifications (red) and deletions (blue) events in each chromosome. **d** Altered signaling pathways in breast cancer tumors. Mutual exclusivity modules (MEMo) analysis identified multiple modules recapitulating *ERBB-PI3K*, *PIK3-Akt-CCND1* signaling (adjusted p-value = <0.1). Heatmap showing the distribution of mutated or amplified genes that make up MEMo modules. **e** Graphical representation of the top-scoring mutually exclusive modules. Nodes represents frequently altered genes in each module and edges connect them according to their reported activity in corresponding core signaling pathways. Amplifications are shown in yellow, somatic mutations events in red, black arrows represents activation and red arrows inhibition. Gain of function biological consequence in dot lines and loss of function in continuous line. MEMo p-values were estimated by comparing the observed alteration frequency of each module to those expected for the same module after randomly permuting the set of observed genomic alterations. q: FDR adjusted p-value. HM: Hispanic-Mexican, AA: African-American, IHC: immunochemistry.

consequences of SCNAs in HM population, a correlation analysis was computed between significant SCNA profiles and significant differential expression of genes (logFC > 1.3, adj p-value < 0.05) contained in the aberrant locus. A total of 184 CNA-mRNA pairs were significantly correlated (R = > 30%, adj p-value < 0.05) (Fig. 7c and Supplementary Data 9). Not surprisingly, genes in amplified regions are involved in cancer pathways, such as DNA repair mechanism, histone acetylation and chromatin remodeling complex (Supplementary Fig. 10). Conversely, multiple deletion events comprise genes with roles in the control of fatty-acid and amino acid metabolic pathways, traffic and localization of vesicles, regulation of cytoskeleton, among others (Supplementary Fig. 10). These observations suggest a convergence of multiple CNA targets on a common set of biological functions important to maintain different hallmarks of cancer.

Oncogenic networks with mutually exclusive genomic alterations between somatic-mutations and SCNA were identified through MEMo algorithm[29] in the profiled HM patients. Two significantly modules (adjusted p < 0.1), affecting a considerable proportion of samples were detected (Fig. 7d, e). Notably, ERBB2

(*HER2*) amplification tend to exclude *PIK3CA/AKT1* mutations (Fig. 7e). Of relevance, *HER2* amplification is highly correlated with its expression (R = 90%, adj p-value < 0.05, Fig. 7c) and might be able to activate PI3K/AKT signaling in a HER3 independent mechanism[30], suggesting that redundant alterations over the same pathway leads to a disadvantage for the cell. Likewise, *PIK3CA*, *AKT1* and *CDH1* gain-of-function mutations were mutually exclusive. Similarly, *TP53* inactivation was mutually exclusive with *PIK3CA/AKT1* mutations in most of the altered cases, nonetheless we found a 15% of co-occurrence (Fig. 7d, e). The oncogenic amplification of *CCND1*, a cell cycle regulatory molecule, was mutually exclusive with *TP53* mutation. Interestingly, amplification of *CCND1* is also correlated with its over-expression (R = 40%, adj p-value < 0.05, Fig. 7c) and consequently enhance G1-S progression through RB/E2F[31]. The observed mutual exclusivity among diverse alterations in TP53 tumor suppressor pathways indicates different manners to dismantle cell division[32]. Although a small proportion of *CCND1* amplified tumors co-occurred with HER2 amplification (3/12) and PIK3CA mutations (4/12), they were generally mutually

excluded. Of interest, in addition to *CCND1* amplification, *CCND1* can be up-regulated through other mechanisms such as ERBB2 driven-activation or by the PI3K-AKT-mTOR pathway, suggesting a functional redundancy of these alterations, and more than one of these alterations might be disadvantageous for the tumoral cell (Fig. 7d, e and Supplementary Data 9). These results pointed out the relevance of PI3K/AKT and ERBB2 signaling pathways alterations as key tumorigenic events, suggesting that these multiple alterations should be considered prior to determining therapy based on molecular characterization.

**Immune infiltration characterization of breast tumors along ancestries.** Our observations identified that TMB was significantly higher in HM patients than in AA and Caucasian women profiled, which could be largely attributed to APOBEC-related mutations. Interestingly, these phenotypes were particularly enriched in the HR + /HER2− (LumA) tumors. Further, a high TMB and the APOBEC signature have been associated with cytotoxic T-lymphocyte infiltration and cytolytic activity implying a stronger anti-tumor immune response. Thereafter, we identified immune attributes based on RNA profiles on the tumor set included in this study. No significant differences were detected among tumors from different populations regarding cytolytic activity (CYT), necessary for immune response[33], or tumor inflammatory signature (TIS)[34] (Fig. 8a). As expected, comparison across PAM50 subtypes revealed a significant enrichment of CYT and TIS in all basal tumors and to lesser extent HER2 tumors, meaning these subtypes were the most immunogenic tumors in all ancestries (Supplementary Fig. 11a). Even more, grouping samples by PAM50 subtype, showed that the CYT and TIS score were higher in LumA tumors from HM, Hispanics and AA individuals than their counterparts from Asian (FDR = 0.007) and Caucasian (FDR = 0.04) women (Fig. 8b), suggesting a more active immune reaction in Hispanic tumors evaluated. This data was corroborated through calculation of the immune score from the Estimated algorithm (FDR = < 0.05) (Supplementary Fig. 11b).

Immune-infiltrating cells portrait evaluated through ssGSEA of immune-cell signatures[35], displayed modest significant ancestry differences. We identified relevant changes between HM tumors vs. AA in enhancement of an immune excluded-like phenotype where T-cells (CD4 and T-helper Type 2) are rate-limiting by the over-representation of mast cells, macrophages and Tregs, thus rendering ineffective T-cells capability to infiltrate the tumor stroma (Supplementary Fig. 11c and Supplementary Data 10). Lastly, immune-phenotypes among molecular subtypes revealed HM LumA tumors as the most variable immune subset among evaluated ancestries (Fig. 8c, d). These results potentially indicate that LumA HM tumors harbors a higher cytotoxic activity that would promote a diverse immune infiltrate context that may impact the restriction of certain clonal tumors (Supplementary Data 10). We reasoned that different selective pressures represented by diverse immune-phenotypes and molecular tumoral subtypes could result in the positive selection of distinct immune mechanisms. These features will help to shed light on how tumors respond to immunotherapies, as well as to provide rationale for the development of novel therapeutic strategies. The meaning of these demographic associations remains unclear but provides evidence for the immune diversity based on genetic ancestry.

## Discussion

The number of samples analyzed in the literature from minorities in BC is still relatively small, which limits the ability to detect ancestry-specific molecular alterations[36]. To our best knowledge, this study represents the largest genomic analysis of BC among patients with HM ancestry residing in Mexico (Fig. 9). To describe a deep biological portrait of the molecular features of HM and Latin-Hispanic BC patients, we compared multi-omics profiles between our set of tumors and public data from other ancestries mainly consisting of Caucasian, Asian, African and Afro-American women.

Recent studies have shown that breast tumors in young women exhibit more aggressive characteristics than those occurring in older patients[37]. Public systems need to face this growing health problem, mainly in developing countries where the incidence of BC is rising[38] and a higher proportion of women debuts at younger stages[37]. In our analysis, younger women (<45 years of age) from Hispanic datasets accounted for a higher proportion compared with Caucasian patients. In accordance with the reported average age at diagnosis in Mexican women, that occurs a decade less than in the Caucasian population[39]. Notably, aggressive basal-like tumors, that are mainly composed by TN tumors, and HR + /HER2 + cancers are enriched in younger HM and Hispanic non-Mexican patients in comparison with patients from non-Hispanic ancestry. In accordance, it has been reported that Mexican young patients have a larger proportion of TN tumors than their counterparts in Europe, US and Asia[40].

The heterogeneous transcriptional phenotypes observed in women with BC belonging to different ancestries, is in part influenced by the alterations in cancer genomes such as mutations and SCNV. Even when a well concordance of SCNA profiles were observed with other ancestries, differences exist in the frequencies of these genomic alterations alongside the detection of unique SCNA in tumors from HM women.

Moreover, we identified recurrent alterations that particularly affect HM tumor genomes, such as the enrichment of $AKT1^{E17K}$ mutation in HR + tumors, with a prevalence of 8%. Interestingly, although this amino acid alteration was identified as a recurrent hotspot mutation in BC[26, 41], other profiles report a lower frequency, ranging from 1.4% to 5.9% in different ancestries[42–44], with a mean frequency of 3.8%. In accordance with these data, through our in-silico analysis of large-scale sequencing studies, we were only able to detect a similar frequency in young BC patients (<35 years) from South Korea, with 8% (4/50), but not in any other dataset of tumors analyzed, nor this has been previously reported by other study, to the best of our knowledge.

The PI3K–AKT–mTOR signaling pathway is one of the most frequently de-regulated pathways in human cancers[41], with repercussions in key cellular processes, such as metabolism, independent cell proliferation, cell invasion, endocrine receptor deregulation and resistance to therapy[45], and consequently supporting cancer cell programs. The pathway can be aberrantly activated through multiple mechanisms, including diverse AKT mutations[46]. E17K mutation activates AKT1 by recruiting it to the membrane through a PI3K-independent mechanism, resulting in the activation of PI3K/AKT/mTOR signaling pathway[47]. Thus, AKT1 mutations have emerged as an attractive druggable target and there is promising clinical data in ER + ductal BC patients harboring $AKT1^{E17K}$ mutation treated with the pan-AKT targeted inhibitors AZD5363, MK-220[46, 48] and ipatasertib, another ATP-competitive AKT inhibitor[49].

It is possible that biological and environmental factors may dictate evolutionary dynamics of a tumor. This assumption may explain the observed mutational signatures portraits and their differences between racial groups described in our study. Mutations contributed by signature 1, which exhibits clock-like properties generally correlated with age[50], were strongly overrepresented in individuals from HM ancestry in contrast to the other datasets analyzed, even still in younger Mexican patients. This particular configuration may be the result of a biological

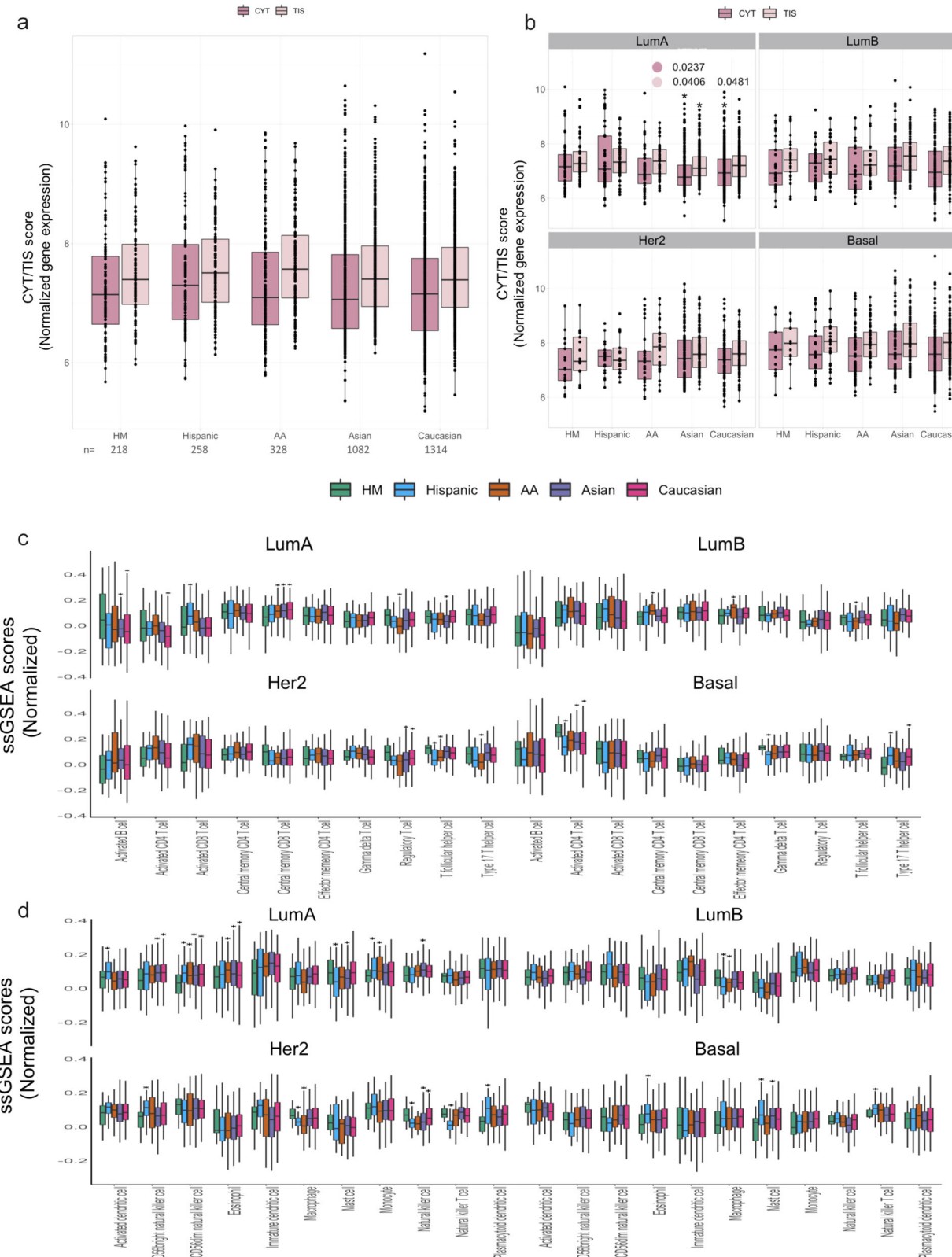

**Fig. 8 Immune landscape of breast cancer tumors. a** Boxplots showing the overall distribution of CYT and TIS scores across breast cancer samples and ancestries. **b** Boxplots describing the distribution of CYT and TIS score among PAM50 intrinsic subtypes in each ancestry-group. *p*-values are indicated near the corresponding asterisks in accordance to color legend (Purple: CYT and pink: TIS). Barplot showing the distribution of immune-cell population signature scores (ssGSEA) of **c** adaptive and **d** innate immune cells across PAM50 subtypes and ancestries. Statistical comparison based on two-tailed Wilcoxon sum of ranking test taking HM group as reference. Boxplots in panel **a–d** represent median ± IQR (25th and 75th percentile) and whiskers correspond to maximum and minimum values. BH-adjusted *p*-value* < 0.05. Statistics from panel **c** and **d** are described at supplementary data 10. HM: Hispanic-Mexican, AA: African-American, LumA: Luminal A, LumB: Luminal B, CYT: cytolytic activity score, TIS: Tumor inflammation signature.

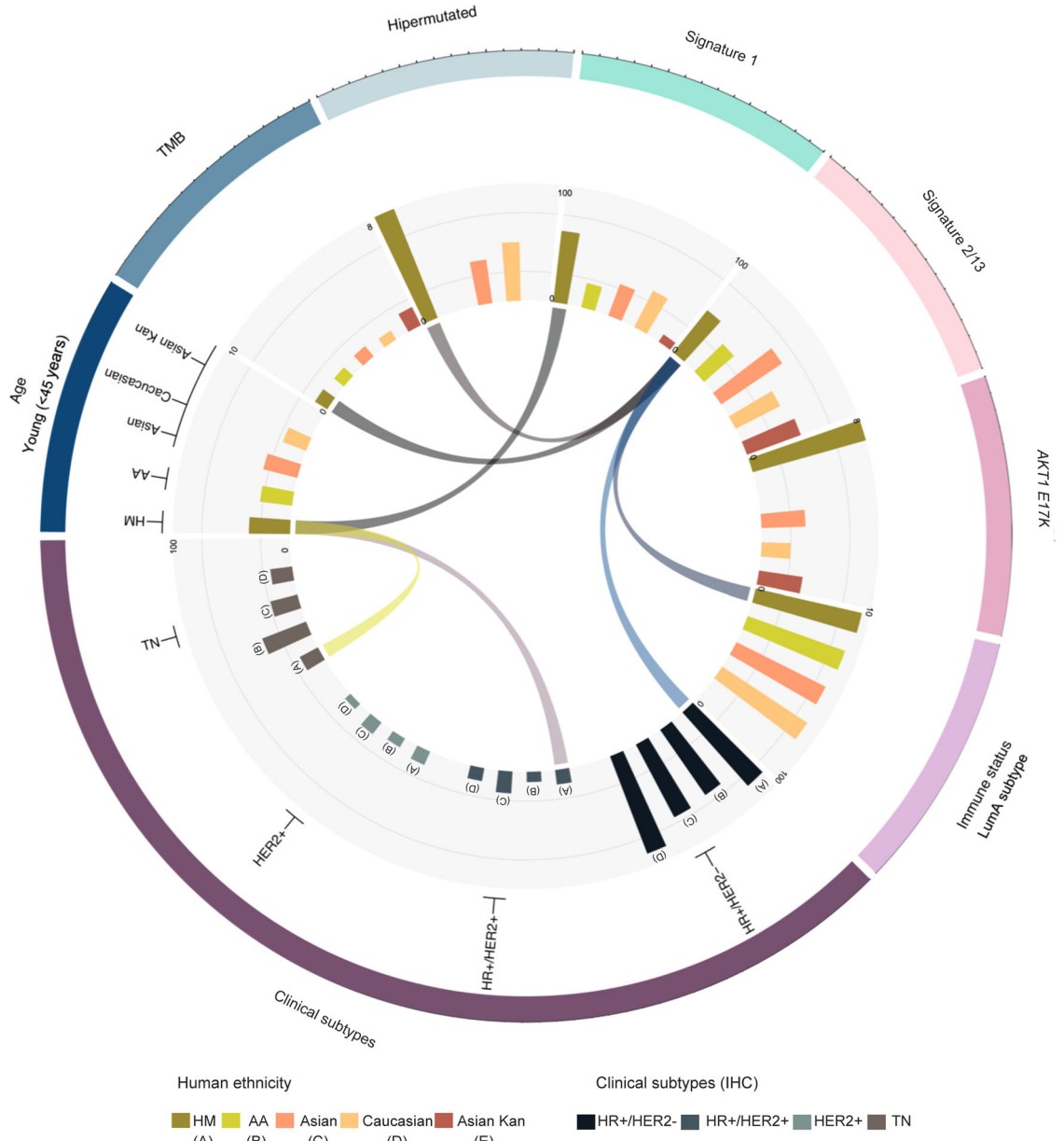

**Fig. 9 Summary of our remarkable findings in the evaluated HM tumors.** The outer circle was divided in each molecular alteration presenting a differential status detected in our evaluated HM tumor samples against other human population. Inner circle shows barplots indicating the frequency of each phenotype as indicated in the labels (y-axis and x-axis). Ribbon connections describe the proposed biological relation among the diverse altered "omics" and clinical features. LumA: Luminal A.

epigenetic clock mechanism as suggested by Kresovich and collaborators[51], that propose that age acceleration is associated with increased BC risk[52]. Thus, it's possible that chronological age not always correlates with biological age, presumably, by the accumulation of biological changes that undergo at different rates because of different carcinogen and environmental exposures or neoplastic changes[50]. Of note, when we considered age at diagnosis and TMB no differences were identified between Mexican younger and older patients. It is also possible that more than one signature contributes to the mutational process. Interestingly, at exploring this possibility, a significant co-occurrence of the contribution of signature 1 with the APOBEC-related signatures was only observed in HM-profiled tumors. Age-associated mutations in tumors reflect a decrease in tissular fitness and an accelerated-ageing that might be explained by intrinsic and exogenous risk factors such as obesity-associated changes in metabolism,

exposure to steroid sex hormones, inflammation and environmental and genetic background. These findings might partly explain why breast cancer is diagnosed at younger ages in HM women compared to what is reported for women with a different ancestry in other studies.

Cancer health disparities studies are often focused on the differences in frequency measures (e.g., incidence, prevalence, mortality, etc.) and even when it is indisputable the impact these factors have on the improvements in the clinical management of BC among different human groups, it is also undeniable that molecular characterization of cancer genomes of diverse populations has an important value in cancer research studies. In this respect, our data extend the knowledge and contribute towards the characterization of the biological and molecular factors in HM patients. There are certain limitations of this study including the number of analyzed samples and the limited number of

matched tumor samples across the "multiomic" characterizations. Even if, this report presents one of the first relatively large and comprehensive characterization (at genomic and transcriptomic level) of breast tumors in Mexican women, our observations can be only extended to the analyzed tumors and may not reflect all population rates and was not conducted in a epidemiologic design. As we took advantage of public data to have a robust multi-ancestry comparison and overcome size limitations, our study is constrained by available information, which impedes the assessment of clinical features impact, such as the tumoral grade. An important notion emerging from this work is that additional efforts to overcome the underrepresentation of Hispanic patients in multi-omics studies are still needed. In line with that notion, a multifactorial vision strategy that considers not only ancestry-related genomic and molecular features, but also socio-demographic issues, environmental exposures, and even public health policies would help refine our understanding of the factors contributing to disparities in BC outcomes among different groups of human populations, still a priority question to address in order to reduce cancer health burden.

## Methods

**Tumor sample collection and clinopathologial assessment of in-house molecular-profiled HM patients**. Mexican patients diagnosed with primary breast cancer, without a second tumor and treated with adjuvant therapy at the Institute of Breast Diseases (FUCAM) from 2008–2012 were convenient collected. Tumor and adjacent non-tumoral tissue, as well as peripheral blood were obtained (EDTA Vacutainer tubes, BD, 6 ml) from each patient after informed consent was obtained. After macroscopic inspection by the pathologist, sections of tumor and normal tissue were frozen in liquid nitrogen and store at −80 °C until further processing. A section of the tissue was formalin fixed and embedded in paraffin (FFPE) to confirm pathological diagnosis, as well as to assess tumor cell content and grade by hematoxylin eosin (H&E) staining. Only samples with tumor content values >60% were further analyzed. Blood samples were centrifuged to separate and isolate buffy and plasma components. Additionally, one hundred consecutive FFPE specimens with adjuvant surgical resection (2012–2016) were convenient collected at Anatomic Pathology Department of FUCAM fulfilling the inclusion criteria described above. Tumor specimens were evaluated by a pathologist to determine their histotype and evaluate cellularity, to then macrodisected the most enriched area with tumoral cells (>60% of tumor cells). The clinical characteristics of the HM in-house-profiled samples are shown in Supplementary Data 1. The protocol was reviewed and approved by the Ethics and Research committees of the National Institute of Genomic Medicine and FUCAM Institute in Mexico City (CE2009/11). All the studies were conducted in accordance with the Declaration of Helsinki.

**Immunohistochemistry**. Estrogen and progesterone receptors, HER2 and EGFR expression, was evaluated using the ER/RP pharmDX (Dako, Denmark, K1904, ready-to-Use), Estrogen receptor alpha (Dako, Denmark, M7047, 1:35), Progesterone receptor (Dako, Denmark, M3569, 1:50), HercepTest (Dako, Denmark, K5204) and DAKO EGFR pharmDxTM kit (Dako, Denmark K1492, ready-to-Use), respectively, following the manufacturer´s instructions. CK5/6 was evaluated with the mouse monoclonal anti-CK5/6 antibody (Dako, Denmark, M7237, 1:20, clone D5/16 B4), cytokeratin 14 (Novocastra, NCL-LL002, 1:20, clone LL02) and cytokeratin 17 (Dako, Denmark M7046, 1:2o, clone), as well as Claudin 1 (Abcam, UK, AB15099, 1:50) and Claudin 3 (Abcam, UK, AB15102, 1:50). Histopathological analysis was performed according to standard protocols by two trained pathologists.

**DNA/RNA extraction**. After tumor cell content evaluation, DNA and RNA were extracted from tumor tissue using the AllPrep DNA/RNA mini kit (Qiagen, Valencia, CA), while DNA from peripheral blood lymphocytes was extracted with the QIAamp DNA Blood Maxi Kit (Qiagen, Valencia, CA), according to manufacturer's instructions. DNA integrity was evaluated by 1% agarose gel electrophoresis and RNA integrity by capillary electrophoresis using the Bionalyzer system (Agilent, Santa Clara, CA). Only samples with RNA integrity number (RIN) >6.0 were used for microarrays analysis.

**mRNA expression profiles**. A total of 109 tumors were processes at the Affymetrix Unit of National Institute of Genomic Medicine (INMEGEN) in Mexico City. Global mRNA expression patterns were analyzed with the Affymetrix (CA, USA) microarray platform Human Gene 1.0 ST. Briefly, 100 to 300 ng of total RNA were obtained from adjacent and neoplastic mammary tissues and processed according to the Affymetrix protocol (Wt manually sense target). Microarrays were read in a scanner and high-resolution model 7Gimage analysis and quality control

was performed using the Affymetrix Expression Console software. The mRNA expression values were normalized with RMA and quantile algorithms in R (http://www.r-project.org/) with the oligo library v 1.46[53] from Bioconductor. Batch correction was done with the SVA package v 3.36.0.[54] using the function ComBat with parametric prior method. Quality controls, as boxplots and histograms of intensity values distribution and MA plots, were visualized with Oligo v 1.46 and limma v 3.38.3[55] tools in R environment. To facilitate comparison between all datasets, including our gene expression profiles and benchmark data, we applied z-score scaling to each gene. Microarray data are available in GEO with the number GSE86374.

**Publicly available datasets**. To have a robust representation of multi-ancestry studies including Caucasian, African-American (AA), Hispanic (Non-Mexicans) and Asian women, we integrated our data with LACE (n = 1635)[56], TGCA (n = 894 from Xena and cbioportal), Metabric (n = 832, cbioportal)[57], SMC (n = 166)[16], Nigerian (n = 96)[58] and Geo omnibus (n = 1830) portraits, including clinical annotations and multidimensional mutational, SCNV and gene expression profiles to develop a deep biological comprehensive of BC among different ancetries (Fig. 1a) (full details of datasets type and composition in Supplementary Data 2). For datasets retrieved from GEO, gene expression analysis was limited to Affymetrix platforms (U133, U133 v2, U133 plus, Gene st). We first recover all the raw data (cel file) from GEO data base, and samples from each dataset were normalized and RMA background corrected with oligo package in R. Quality control were performed and only samples with optimal QC parameters were included. We then annotate the probe sets using BioMart[59] on R package (biomaRt v2.38.0) and extract the common genes between all platforms resulting in 13,060 common genes. Then, all the common genes from all samples were putting together in a single matrix and corrected for batch effect with ComBat[60] in SVA R package with parametric prior method. For data retrieved from GEO processed in another microarray platform (Illumina or Agilent) only clinical data were downloaded to report it.

**Estimate**. As a quality control of samples ESTIMATE scores were computed, including purity, stromal and immune values, were calculated using the ESTIMATE R package v0.22.0[61]. Samples with a tumor purity score under 60% were discarded for gene expression profiles description and further analysis.

**Pam50 subtyping and triple-negative breast cancer molecular classification**. To obtain the intrinsic subtype classification, the PAM50 algorithm was applied as described by Parker[14]. The Pam50 subtype centroids were applied to the normalized data and the classification was carried out using the pbcmc package v 1.6.0[62] on Bioconductor using robust parameters (nPerm = 10,000, pCutoff = 0.01, where = fdr and corCutoff = 0.1), defining the following molecular subtypes: luminal A and B, HER2-enriched, Basal-like and Normal-Like for each sample in our study[63]. Samples classified as Normal-like were discarded due to possible contamination by normal tissue. For triple-negative samples, TNBC molecular subtypes were identified using TNBCtype v.1.0 (http://cbc.mc.vanderbilt.edu/tnbc/) to corroborate TN status.

**Genotyping arrays of samples from HM women**. DNA from tumor and peripheral blood was processed using Affymetrix Genome-Wide Human SNP 6.0 (Affymetrix, CA, USA) according to manufacturer's protocols. Briefly, DNA was digested with NspI and StyI enzymes (New England Biolabs, USA), ligated to the respective Affymetrix adapters using T4 DNA ligase (New England Biolabs, USA) amplified (Clontech-Takara Bio, USA), purified using magnetic beads (Agencourt, Beckman, USA), labeled, fragmented, and hybridized to the arrays. Following hybridization, the arrays were washed and stained with streptavidin-phycoerythrin (Invitrogen, USA). Arrays preparation and scanning was performed at the genotyping core laboratory of INMEGEN. Background correction and extraction of raw fluorescence intensity were performed with the Affymetrix Genotyping Console. Raw microarray data was deposited in GEO under the number GSE87048.

**Ancestry proportion analysis**. The ancestry estimation calculation of the Mexicans breast cancer patients of our dataset were performed using 299,411 SNPs from Affymetrix SNP6.0 microarray to inferred ancestry proportions based on the four main ancestries populations in the American Continent as reference. The first three ancestries populations were retrieved from the HapMap International Project: 27 individuals with northern European ancestry (CEU), 27 individuals with African ancestry from Yoruba in Ibadan, Nigeria and 41 East Asian ancestry (merge of Japan and China individuals) were included, together, with 37 additional Native Americans from Mexico (10 Zapotecas from Oaxaca, 13 Tepehuanos from Durango and 14 Mayas from Campeche) from the Mexican Genome Diversity Project (MGDP)[7, 8], finally we include 161 Mexican mestizo (admixed population) of six Mexican federal states to compare the ancestry mean between breast cancer patients and Mexican mestizo population. The ancestry proportions were calculated using a fast sequential quadratic programming algorithm and novel quasi-Newton acceleration method implemented in ADMIXTURE Software v.1.3.0[64, 65]. For samples in our collection that were not profiled by SNP6 Affymetrix Array, ancestry was reported as self-identified Mexican ancestry. For samples from public

datasets in GEO, ancestry was retrieved from sample information deposited within in each series matrix file, while TCGA ancestry data were recovered from clinical information deposited on cbioportal and from METABRIC tumors based on information reported on ref. [66].

**Whole-exome sequencing**. Whole-exome was captured through Agilent SureSelect ExomePlus bait system (Agilent Technologies, USA) and biotinylated RNA baits[67] process. ~155 Mb baits were target including the standard exome, plus intronic and promoter sequences for known cancer genes, relevant targets identified by cancer genomic studies, TCGA and the Cancer Cell Line Encyclopedia. Exome libraries were sequenced on Illumina GAII or HiSeq 2000 (Illumina, USA) sequencers with 76 base-paired-end reads achieving a mean of 141x. Alignment was performed with the Burrows–Wheeler Alignment tool[68] and standard Picard pipeline to the human genome assembly hg19/GRCh37.

**Somatic mutation analysis**. For HM patients, a dataset of aggregated mutations was generated by combining 54 MAF files retrieved from cBioPortal (Breast Invasive Carcinoma Broad, Nature 2012[11]) and 100 MAF files corresponding to Whole-exome sequencing data of breast tumors from HM patients[69]. Additionally, the annotated and filtered MAF files of 894 TCGA Breast Invasive carcinoma samples, were obtained from Firebrowse database (v1.1.40)[70]. Briefly, somatic mutations were identified via MuTect[71], with a standard panel of normal samples complemented with extra information from ExomePlus cohort. Local realignment and FFPE artifact filter were applied to remove alignment and technical artifacts. Variants were annotated with Oncotator[72] and TransVar[73]. Asian Kan data were downloaded from supplementary information[16]. Next, significantly mutated genes, for each sample collection, were identified using MutSigCV v1.3.01 algorithm[22] through GenePattern module. Since, coverage information file was not available for the analyzed datasets, we used the "full coverage" file, that is available on the GenePattern public server (exome_full192.coverage.txt) assuming full coverage. This file provides information of how the reference sequence of the human exome breaks down by gene, category and effect. For this analysis, the TCGA and public samples were independently analyzed based on ancestry. Only mutations alterations with a q-value < 0.05 (false-discovery rate) were considered significant. Visualization was performed with Maftools v 1.8.10[74].

**Hypermutation cutoff in breast cancer samples analyzed**. To define a hypermutated threshold in the set of breast cancer samples analyzed coming from different human populations we applied a segmented linear regression. Since very extreme values (outliers) can produce disproportionate effects on the slope of the regression equation, we first identified those influential points. Therefore, we applied the median and median absolute deviation method (MAD), that is an outlier detection approach and filter all samples with values over 15 mut/Mb ($N = 12$ excluded samples). We then computed the segmented function (control = seg. control, n.boot = 0, it.max = 1000, $K = 10$, psi missing) on the Segmented package implemented in R. The first breakpoint at which a statistically significant change occurred was at 4mt/Mb.

**Actionable genomic alterations**. Hotspot mutations have been identified with cancer hotspot database (http://cancerhotspots.org)[26, 75] using protein change annotation retrieved from TransVar[73].

Cancer hotspot tool provides useful resources for query and visualization of statistically significant hotspot mutations among diverse tumors. Then, to determine the clinical actionability and oncogenicity of identified mutations, OncoKB[27] knowledgebase was interrogated. OncoKB provides disease-specific levels of evidence for the actionability of mutant alleles and DNA copy-number alterations as follows: (1) level 1 alteration is an FDA-recognized biomarker in specific tumor type; (2) level 2 is a biomarker routinely used to guide prescribing of an FDA-approved drug, based on tumor type (2A) or other indication (2B); (3) level 3 demonstrates compelling clinical evidence supporting its use as a biomarker. OncoKb also categorized a specific gene as "oncogenic" or "likely oncogenic" in accordance with its alterations and biological consequence.

**Driver mutation discovery**. Detected mutations were annotated and classified as driver and passenger somatic mutations using the method implemented in OncodriveMUT algorithm[76] and the Cancer Genome Interpreter (https://www.cancergenomeinterpreter.org/home) framework, which allows to identify the most likely driver mutations of a tumor. The oncogenic classification: known and predicted mutation (in any neoplasia), were considered as driver alteration and taken into account for analysis.

**Structural 3D view of AKT1 mutations**. c protein visualization for *AKT* mutation was performed with RCSB PDB (rcsb.org)[77] for *AKT* E17K mutation (2UZS) with the NGL Viewer, and RCSB PDB[78].

**Mutational signatures**. Identification of mutational signatures in tumor samples was performed with deconstructSigs v1.8.0[18]. Briefly, the algorithm determines the linear combination of predefined signatures (COSMIC project v2, www.cancer.

sanger.ac.uk/cosmic/signaturesnd), that more accurately reconstructs the mutational profile of a single tumor sample using SNP type variants to estimate the contribution for each signature. To determine how much of each signature is present in each sample we ran the whichSignatures function, using the following parameters; tumor.ref = context matrix consisting of the counts of the mutations observed in each of the 96 possible trinucleotide contexts for each sample, signatures.ref = reference signatures matrix, which values represent the fraction of times a mutation is seen in each of the 96-trinucleotide contexts for each selected signature, with the following parameters tri.counts.method = exome2genome method as normalization strategy and signature.cutoff = 0.06. As a cross-validation method we computed mutational signatures with SigFit v2.0 package[19], which estimate signature contribution based on a Markov Chain Monte Carlo sampling and Non-Negative Matrix Factorization (NMF) model to fit the mutational input to a COSMIC mutational catalog. The parameters used were: iter = 6000, warmup = 3000, and hpd_prob = 0.90 to get the exposures and 90% highest posterior density intervals. To define the overall signature exposure in each sample, the calculated signature weights were multiplied by the COSMIC trinucleotide context probability within the corresponding signature and then normalized by the total number of mutations in that sample. The signature with the maximum exposure was considered as the one that mostly contributes in that sample. To maximize the algorithms performance, only samples harboring a TMB higher than the median in each datasets were included (HM median TMB:0.99 mut/Mb, Asian TCGA: 0.92, Asian Kan: 1.39794, AA TCGA 0.86 and Caucasian TCGA 0.72). A mutational signature was considered to be present if it contributes >5% towards the entire mutational load of a sample. The probability of co-occurrence (i.e., the frequency of occurrence) of any two predominant mutational signatures in each tumor dataset was estimated as the statistical association between signatures across samples in that tumor collection, using a Fisher Exact Test. Co-occurrence was considered to be present if an association was found with FDR-corrected p-value < 0.1.

**Copy-number profiles**. To generate the copy-number profiles, we used the "Copy-Number Inference" pipeline implemented in Genepattern[79] (http://genepattern.broadinstitute.org/gp/pages/index.jsf). In brief, probe-level signal intensities from Affymetrix SNP6 (cel files) were normalized to a uniform brightness and merged to obtain values for each probe set using SNP File Creator. These intensities measurements were converted into a copy-number call by the Copy-Number Inference model. After reducing the noise by subtracting out the variation that is also seen within the normal samples (peripheral blood), the CBS segmentation algorithm identifies regions in the genome that have a uniform underlying copy number, creating a segmentation file. Finally, to identify significant focal SCNAs in induvial genes of across different ancestries, GISTIC 2.0 (v2.0.23)[80] algorithm was computed in the Genepattern environment (Deletion Threshold = 0.1, cap-values = 1.5, broad length cutoff = 0.7, Remove X-Chromosome = 0, Confidence Level = 0.90, Join Segment Size = 4, Arm Level Peel Off = 1, Maximum Sample Segments = 2000, Gene GISTIC = 1). Segmentation file were used as input. Peaks with q-value < 0.15 were selected as significant. For the comparison of our dataset, TCGA segmentation data were downloaded from cbioportal, and processed as previously described in the GISTIC module for each ancestry.

**Frequency of altered genome**. The tumor altered fraction (TAF) was computed as reported previously[81]: the length of segments with log2 copy-number level (log2-ratios) larger than 0.2 divided by the length of all segments measured.

**Correlation between CNA and gene expression**. Pearson correlation between copy-number changes of significantly deleted or amplified regions, as inferred by GISTIC, and significantly differentially expressed genes located within those regions was performed on 49 MH tumor samples with CNA values and gene expression data available. Correlations for each possible region and its contained genes were computed and considered to be significant with a Pearson R coefficient > 0.3 and p-value < 0.05. Functional annotation of GO biological process and KEGG terms (GO_Biological_Process_2018 and KEGG_2019_Human) among significantly correlated genes was performed using the package enrichR v2.1[82] implemented in Bioconductor/R, with default parameters. We then selected the top 15 terms, ordered by raw p-value from each collection, considering genes in amplify and deleted regions separately.

**Mutually exclusivity modules in cancer**. To search for genomic alterations occurring in biological pathway context, we used the MEMo v1.1 program, which is a method for identifying network modules of significantly mutually exclusive alterations in member genes that show recurrent somatic mutations or CNA and are likely to belong to the same biological pathway or process. MEMo was run across all breast tumors profiled by exome sequencing and/or genome-wide arrays (i.e., samples with information on single nucleotide mutations and/or CNV data) selecting alterations affecting at least 2% of the samples. We used HRN2 as background reference network and the following parameters, mut_sig_q_value_threshold = 0.10 (i.e., the threshold of FDR correction computed with the Benjamini & Hochberg method) and min_number_of_alterations = 3, which selects the minimum number of alterations

in a gene to be include for analysis. MEMo statistically significant modules were selected with an FDR-corrected *p*-value ≤ 0.1.

**Immune characterization through CYT, TIS, and ssGSEA of immune signatures.** Cytolytic signature (CYT) that measure the activity of lymphocytes T-CD8 to punch the tumoral cells was calculated through a validated gene expression signature based on the geometric mean of normalized gene expression levels of granzyme A (GZMA) and perforin-1 (PRF1)[33]. Tumor Inflammation Signature (TIS) score[34], correlated with CD8, CD4, natural Killer and macrophage M2-like activity, was calculated as the average of continuous mean of log2-transformed normalized expression of the identified genes. Individual enrichment scores based on immune-related gene expression signatures[50] were computed with ssGSEA implemented in GSVA v1.36.2[83] Bioconductor library with min gene set size parameter of 5. Wilcoxon test were applied to define significant differences among HM vs. any of the other populations.

**Statistics.** To compare associations of the oncogenic alterations between datasets and intrinsic subtypes a Fisher exact test was performed, using the fisher.multcomp function on R, and *p*-values were controlled for false-discovery rate with the Benjamini & Hochberg method and a multivariate Cox proportional hazards analysis was computed with SAS University Edition Statistical. Two-tailed Kruskal–Wallis or Wilcoxon test were applied to define statistical differences between the conditions evaluated on continuous variables. Statistical significance was set as *p* = < 0.05.

**Reporting summary.** Further information on research design is available in the Nature Research Reporting Summary linked to this article.

## Data availability

Gene Expression data generated in this study have been deposited in the GEO (GSE86374). SNP array can be found in the GEO (GSE87048). Somatic mutations and annotation data from HM women can be retrieved from cbioportal under the URL [https://cbioportal-datahub.s3.amazonaws.com/brca_broad.tar.gz] and from the dbGAP database [https://www.ncbi.nlm.nih.gov/gap/] under accession number phs001250. v1.p1 (upon authorization request). Public gene expression datasets used in this study are available in the GEO database under the following accession codes: GSE78958, GSE16716, GSE20271, GSE37751, GSE48390, GSE54002, GSE15852, GSE2109, GSE75678, GSE113184, GSE59595. Public somatic mutations and annotation data from TGCA breast cancer samples are available through Firebrowse database under the URL [http://gdac.broadinstitute.org/runs/stddata__2016_01_28/data/BRCA/20160128/gdac. broadinstitute.org_BRCA.Mutation_Packager_Oncotated_Calls.Level_3.2016012800.0.0. tar.gz]. Copy-number data from TGCA breast cancer samples are available in Xenabrowser portal under the URL [https://xenabrowser.net/datapages/?dataset=TCGA-BRCA.cnv.tsv&host=https%3A%2F%2Fgdc.xenahubs.net&removeHub=https%3A%2F%2Fxena.treehouse.gi.ucsc.edu%3A443]. Clinical and molecular subtype data from METABRIC and Metastatic breast cancer (INSERM, *PLoS Med* 2016)[84] samples are available in cbioportal database under the URLs [https://www.cbioportal.org/study/clinicalData?id=brca_metabric] and [https://www.cbioportal.org/study/clinicalData?id=brca_igr_2015]. Clinical and molecular subtype from Nigerian breast cancer samples (Pitt et al. 2018)[58] are available as supplementary information in the URL [https://static-content.springer.com/esm/art%3A10.1038%2Fs41467-018-06616-0/MediaObjects/41467_2018_6616_MOESM4_ESM.xlsx]. Clinical and molecular subtype from Asian breast cancers samples (SMC Kan et al. 2018)[16] are available as supplementary information in the URL [https://www.nature.com/articles/s41467-018-04129-4#Sec24]. Clinical and molecular subtype from Japanese breast cancer samples (Hatakeyama et al. 2019)[17] are available as supplementary information in the URL (https://onlinelibrary.wiley.com/action/downloadSupplement?doi=10.1111%2Fcas.14087&file=cas14087-sup-0002-TableS1.xlsx). The remaining data are available within the article, supplementary data or available from the authors upon request.

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

## Acknowledgements

We want to thank all the patients that made possible this study, Dra. Esperanza Monterrubio Flores for her support in the immunohistochemical analysis. Paulina Michelle García Vargas and Rodrigo Bolado Hadad for their support to collect clinical data of HM dataset. Rosa Gloria Rebollar Vega is a doctoral student from Programa de Doctorado en Ciencias Biomédicas, Universidad Nacional Autónoma de México (UNAM), and received fellowship (CVU 294330) from CONACYT. S.R.C. and I.S.G. were supported by a postdoctoral fellowship from the Mexican Council of Science and Technology (CONACYT) and S.R.C. has been supported by Cátedra Salvador Zubirán, UNAM/INCMNSZ. This work was funded by the Mexican National Council of Science and Technology Basic Science grant (grant number 258936) and Frontiers in Science grant (number 1285).

## Author contributions

I.S-G., R.R-V., and S.L.R.-C. contributed equally to this work. S.L.R.-C. and A.H.-M. conceived and designed the study. R.R.-V., S.L.R.-C., and L.A.-R. collected tumoral specimens (Fresh-frozen and formalin-fixed paraffin-embedded tumor tissue). I.S.-G., R. R.-V., S.L.R.-C., and A.H.-M. designed the pipeline-analysis. R.R.-V., L.A.-R., and L.U.F. performed nucleic acid extraction and sample preparation for microarray analysis. I.S.-G. and S.L.R.-C. performed all the genomic analysis. I.S.-G. and S.L.R.-C. wrote the paper. R. R.-V., A.H.-M. and L.U.F. drafted, edit, discuss, and finalized the paper. I.S.-G., S.L.R.-C. and J.C.F.-L. provided bioinformatics support. V.B.-P., F.V.C., A.T.-T., C.D.-R. procured patient tumoral specimens, assisted the pathological assessment, and immunochemistry evaluation, and provided clinical features of analyzed samples. R.R.-V. performed immunohistochemical evaluation for triple-negative tumors (extended markers). R.R.-V. and L.A.-R. managed the clinical data bases. J.C.F.-L. performed ancestry analysis of HM tumors. All authors reviewed and approved the final manuscript.

## Competing interests

A.H.-M. received a grant from Astra Zeneca Mexico, outside the submitted work. All remaining authors declare no competing interests.
