## [Peer Review File · Nature Communications]

Reviewers' comments:

Reviewer #1 (Remarks to the Author):

The authors investigate molecular subtypes of breast cancer in a case-series from a hospital in Mexico (N=185) compared to publically available datasets.

While the analysis of diverse populations a very important area of research that needs to be addressed, gross sweeping claims attributed to whole population requires extreme caution, especially if not conducted in a epidemiologic study design. The authors claim to identify an age-related COSMIC signature 1 (which is not defined in the abstract, but presumably refers to https://cancer.sanger.ac.uk/cosmic/signatures_v2 catalogue of somatic mutations in cancer)

The details of these samples are extremely limited and its unclear how these subjects/cases were selected. I recognise that this is an issue in many subtyping studies of cancer, but note that guidelines have been developed, particularly the REMARK guidelines for such studies and its inaccurate to call these samples a cohort in the epidemiologic context. The material and methods are extremely weak in their reporting of patient characteristics beyond age (e.g. comorbidities, risk factor information and demographic characteristics education etc), inclusion and exclusion criteria--based on the website it looks like FUCAM diagnoses 1500 breast cancer cases/year so given that only 185 cases were used in this dataset diagnosed from 2008-2012-- its completely unclear how these subjects were selected for. Given that 185 represents <3% of all cases potentially diagnosed it is crucial that these details are noted as these subjects are not likely representative of the population. As we know breast cancer is a disease that is influenced by important risk factor characteristics, to make comparisons with other populations, such details are needed. As ethnicity/race is correlated with socioeconomic status and health status, these confounders were not obtained in this dataset (or were not reported on) and we must be careful making statements of associations of race/ethnicity with health outcomes without such details-- this is a limitation of such work that must be acknowledged.

While I think it still important to present and use such valuable data as was collected and analysed in this study, the authors need to refrain from making sweeping statements claiming that mutations in certain genes are more common compared to other ethnicities---I would focus just on what's new and not comment as to whether things are more or less common compared to other populations ---none of these samples are representative so who knows if they are more or less common all the samples are biased. immune signatures are an interesting area and so I guess focusing on that aspect an important contribution of this work to the literature.

It is clear that the authors performed a great amount of work and analysis--and I commend them for this. However, I must admit that I found it difficult to follow the logic and presentation of the analyses. It was not clear which hypotheses were being tested and how the different experiments were connected, especially since they start with presenting other publicly available data before describing their new dataset. I potential revised presentation could start with the focused analysis of their data and then comparison with other datasets--with the caveats noted above--the data are what they are, but we must be clear that they do not likely represent the general populations of any of the ethnicities/races.

Reviewer #2 (Remarks to the Author):

The manuscript by Salido-Guadarrama et al presents analyses of genomic data of breast cancer in Mexican-Hispanic women and contrasts these findings with data from other populations. Overall the manuscript reads more as a summary of a data together with many disparate analyses rather than a hypothesis driven research article, making almost impossible to quantify its contribution

beyond adding ~100 or so breast cancer samples of Mexican Hispanic ancestry to the field. More importantly, the vast majority of analyses presented in this work are not supported by statistical significance thus leading to suggestive statements at best; no attention is given to multiple testing hypotheses correction and thus results could be stochastic fluctuations in the data and not reflect true biology.

Major comments:

1. The methods section is unreadable and does not describe in sufficient detail the parameters of methods used in this work. This makes it impossible to replicate most of results presented here. The supplementary methods section presents some details on how data was generated but lacks detail on methodologies used throughout the manuscript.
2. Most of comparisons are not supported by significance p-values and/or confidence intervals making it impossible to differentiate between stochastic fluctuations in the data vs true biology. For example, the first paragraph of results notes several percentages of tumor types (without any confidence intervals) and cannot conclude if difference is due to number of tumors evaluated (i.e. random noise) or different admixture patterns. Similar statements exist throughout the entire manuscript.
3. Results second paragraph. Age diagnosis correlated with ethnicity; could this be due to the biased nature of the data presented here? In what ways is are the ~100 MH breast cancer samples representative of the general population of breast cancer in Mexican-Hispanic ancestry? It seems that with such a small sample size and no replication in other cohorts makes it impossible to draw conclusions about general population. Furthermore, how is the multiple testing correction handled? The authors perform several tests and should correct for multiple testing to avoid false positives.
4. Distribution of subtypes across Native American ancestry component. Why divide in 4 groups and not simply correlate with genome-wide ancestry proportion? Why present these results if they are not even marginally significant ($p\text{-val} > 0.05$)? How was ancestry inference performed? The supplementary note states that admixture was used with 4 components; are results robust to method/approach to infer ancestry from the genotype data?
5. IHC marker distribution. Again the p-values are not corrected for multiple testing and the results could be just due to random fluctuations in the data (not biological).
6. Several computational tools are used throughout with no details on the parameters used and how choice of parameters could alter the conclusions (e.g., MutSigCV, deconstructSig, etc).
7. Mutational signatures across ethnicities. Again no consideration is given to multiple testing; all p-values presented here are not significant after multiple testing correction.

Reviewers' comments:

Reviewer #1 (Remarks to the Author):

The authors investigate molecular subtypes of breast cancer in a case-series from a hospital in Mexico (N=185) compared to publicly available datasets.

While the analysis of diverse populations a very important area of research that needs to be addressed, gross sweeping claims attributed to whole population requires extreme caution, especially if not conducted in an epidemiologic study design. The authors claim to identify an age-related COSMIC signature 1 (which is not defined in the abstract, but presumably refers to https://cancer.sanger.ac.uk/cosmic/signatures_v2 catalogue of somatic mutations in cancer)

The details of these samples are extremely limited and its unclear how these subjects/cases were selected. I recognise that this is an issue in many subtyping studies of cancer, but note that guidelines have been developed, particularly the REMARK guidelines for such studies and its inaccurate to call these samples a cohort in the epidemiologic context. The material and methods are extremely weak in their reporting of patient characteristics beyond age (e.g. comorbidities, risk factor information and demographic characteristics education etc), inclusion and exclusion criteria--based on the website it looks like FUCAM diagnoses 1500 breast cancer cases/year so given that only 185 cases were used in this dataset diagnosed from 2008-2012-- its completely unclear how these subjects were selected for. Given that 185 represents <3% of all cases potentially diagnosed it is crucial that these details are noted as these subjects are not likely representative of the population. As we know breast cancer is a disease that is influenced by important risk factor characteristics, to make comparisons with other populations, such details are needed. As ethnicity/race is correlated with socioeconomic status and health status, these confounders were not obtained in this dataset (or were not reported on) and we must be careful making statements of associations of race/ethnicity with health outcomes without such details--this is a limitation of such work that must be acknowledged.

While I think it still important to present and use such valuable data as was collected and analysed in this study, the authors need to refrain from making sweeping statements claiming that mutations in certain genes are more common compared to other ethnicities-- -I would focus just on what's new and not comment as to whether things are more or less common compared to other populations ---none of these samples are representative so who knows if they are more or less common all the samples are biased. immune signatures are an interesting area and so I guess focusing on that aspect an important contribution of this work to the literature.

It is clear that the authors performed a great amount of work and analysis--and I commend them for this. However, I must admit that I found it difficult to follow the logic and presentation of the analyses. It was not clear which hypotheses were being tested and how

the different experiments were connected, especially since they start with presenting other publicly available data before describing their new dataset. A potential revised presentation could start with the focused analysis of their data and then comparison with other datasets--with the caveats noted above--the data are what they are, but we must be clear that they do not likely represent the general populations of any of the ethnicities/races.

Reviewer #2 (Remarks to the Author):

The manuscript by Salido-Guadarrama et al presents analyses of genomic data of breast cancer in Mexican-Hispanic women and contrasts these findings with data from other populations. Overall the manuscript reads more as a summary of a data together with many disparate analyses rather than a hypothesis driven research article, making almost impossible to quantify its contribution beyond adding ~100 or so breast cancer samples of Mexican Hispanic ancestry to the field. More importantly, the vast majority of analyses presented in this work are not supported by statistical significance thus leading to suggestive statements at best; no attention is given to multiple testing hypotheses correction and thus results could be stochastic fluctuations in the data and not reflect true biology.

Major comments:

1. The methods section is unreadable and does not describe in sufficient detail the parameters of methods used in this work. This makes it impossible to replicate most of results presented here. The supplementary methods section presents some details on how data was generated but lacks detail on methodologies used throughout the manuscript.

2. Most of comparisons are not supported by significance p-values and/or confidence intervals making it impossible to differentiate between stochastic fluctuations in the data vs true biology. For example, the first paragraph of results notes several percentages of tumor types (without any confidence intervals) and cannot conclude if difference is due to number of tumors evaluated (i.e. random noise) or different admixture patterns. Similar statements exist throughout the entire manuscript.

3. Results second paragraph. Age diagnosis correlated with ethnicity; could this be due to the biased nature of the data presented here? In what ways is are the ~100 MH breast cancer samples representative of the general population of breast cancer in Mexican-Hispanic ancestry? It seems that with such a small sample size and no replication in other cohorts makes it impossible to draw conclusions about general population. Furthermore, how is the multiple testing correction handled? The authors perform several tests and should correct for multiple testing to avoid false positives.

4. Distribution of subtypes across Native American ancestry component. Why divide in 4 groups and not simply correlate with genome-wide ancestry proportion? Why present these results if they are not even marginally significant ($p\text{-val} > 0.05$)? How was ancestry inference performed? The supplementary note states that admixture was used with 4 components; are results robust to method/approach to infer ancestry from the genotype data?

5. IHC marker distribution. Again, the p-values are not corrected for multiple testing and the results could be just due to random fluctuations in the data (not biological).

6. Several computational tools are used throughout with no details on the parameters used and how choice of parameters could alter the conclusions (e.g., MutSigCV, deconstructSig, etc).

7. Mutational signatures across ethnicities. Again, no consideration is given to multiple testing; all p-values presented here are not significant after multiple testing correction.

Response to reviewers' comments

Reviewer #1 (Remarks to the Author):

We fully appreciate the reviewer for his/her consideration and useful comments on it. In turn, we have performed a thorough revision of our study to try to address in the best possible way all concerns expressed by the reviewer. Please, find our point-by-point response below.

The authors investigate molecular subtypes of breast cancer in a case-series from a hospital in Mexico (N=185) compared to publically available datasets. While the analysis of diverse populations a very important area of research that needs to be addressed, gross sweeping claims attributed to whole population requires extreme caution, especially if not conducted in a epidemiologic study design.

We agree with this observation, however, of note, most of main international efforts to characterize breast cancer samples such as TCGA and METABRIC were convenient collected from biobanks located in multiple health centers. Those, TCGA and METABRIC samples do not reflect population features, and were not conducted in epidemiologic study designs.

We also agree with the reviewer that samples from our collection of Mexican Hispanic patient does not represent the diversity of Mexican population and taking into account your valuable comment, changes along the text has been made accordingly, stating that our findings only reflect the biological and molecular landscape of the profiled patients in this study and other public profiles, instead of being representative data from Mexican or other human population.

In specific, the tittle was adapted to a new one: "Comprehensive omic characterization of breast cancer in Mexican-Hispanic women". In the Introduction, line 77, we have included a sentence which we believe helps to clarify our scope, in which we pointed out the fact that our object is to leverage data about patients with a different ancestry to frame the findings within our sample set which is composed of Mexican patients with a Mexica-Hispanic admixture ancestry. The sentence reads:

"To get a comprehensive understanding of the molecular architecture of breast cancer in a set of Mexican patients, we evaluated somatic mutations, SCNA and gene expression patterns on 204 tumors from HM women and we comparatively describe their genomic context in contrast to patients with African, African-American, Asian and Caucasian ancestry"

Along results section we described our data restricting their interpretation to our profiled samples.

On the Discussion section in line 504, we inserted a sentence that reads:

"although here we report the first relatively large and comprehensive characterization (at genomic and transcriptomic level) of breast tumors in Mexican women, the study may not reflect all population rates and was not conducted in a epidemiologic design"

Importantly, we carefully sought to extend this concept along all our descriptions and conclusions along the manuscript.

Nonetheless, the acknowledged limitations, this dataset represents the first relatively larger and comprehensive characterization of breast tumors in Mexican Hispanic women residing in Mexico and although in itself it is not sufficient to support any epidemiological conclusion, which has been a limitation in most cancer genomics studies, we consider that our data provides a valuable contribution to the body of novel evidence and alterations not presented in other sample collections from different populations such as a robust enrichment in the proportion of clock-like related mutational signatures in the evaluated Mexican cases. For historical reasons, there is a lack of screening in Mexican patients that significantly limits our understanding of how differences in ancestry, lifestyle, environmental, and genetic factors may play a role in breast cancer. In light of these fact, we consider that our study has the objective to provide an important first step to delineate the genomic/molecular landscape of breast cancer in Mexican women, which as stated, still lacks representation among large studies conducted so far.

The authors claim to identify an age-related COSMIC signature 1(which is not defined in the abstract, but presumably refers to https://cancer.sanger.ac.uk/cosmic/signatures_v2 catalogue of somatic mutations in cancer)

We thank the reviewer for this correction. We clarify this information, indicating that we used the predefined COSMIC mutational signatures v2, where applicable. Specifically, in the abstract section, line 33, where it now reads: "age-related COSMIC (v2) signature 1"

The details of these samples are extremely limited and its unclear how these subjects/cases were selected. I recognise that this is an issue in many subtyping studies of cancer, but note

that guidelines have been developed, particularly the REMARK guidelines for such studies and its inaccurate to call these samples a cohort in the epidemiologic context. The material and methods are extremely weak in their reporting of patient characteristics beyond age (e.g. comorbidities, risk factor information and demographic characteristics education etc), inclusion and exclusion criteria--based on the website it looks like FUCAM diagnoses 1500 breast cancer cases/year so given that only 185 cases were used in this dataset diagnosed from 2008-2012-- its completely unclear how these subjects were selected for. Given that 185 represents <3% of all cases potentially diagnosed it is crucial that these details are noted as these subjects are not likely representative of the population. As we know breast cancer is a disease that is influenced by important risk factor characteristics, to make comparisons with other populations, such details are needed. As ethnicity/race is correlated with socioeconomic status and health status, these confounders were not obtained in this dataset (or were not reported on) and we must be careful making statements of associations of race/ethnicity with health outcomes without such details--this is a limitation of such work that must be acknowledged.

In accordance with Reviewer 1's suggestion we made a big effort to collect the most reliable clinical and demographic information of our profiled patients including: ER (IHC), PR (IHC), Her2 (IHC), Age, Histology, Grade, Tumor Size (mm), TNM, Age at menarche, number of pregnancies, menopause, Age at menopause and BMI. We kindly refer the reader to Supplementary table 1 within our text to verify all this information.

Since we are not conducting an epidemiological collection or study, which is now reflected within the text, we have cautiously avoided to refer our group of samples, or public dataset, as cohort.

Regarding the inclusion criteria for our biospecimen collection, we have included the following statement within the Methods section, line 525:

"Mexican patients diagnose with primary breast cancer, without a second tumor and treated with adjuvant therapy at the Institute of Breast Diseases (FUCAM) from 2008-2012 were convenient collected."

It is important to add that our original collection samples comprised about 400 samples, but part of these samples were discarded due to quality control process on tumor cell content evaluation, nucleic acid extraction, or microarray/WES library preparation. Of note, not all patients at FUCAM are treated in adjuvant chemotherapy since they are clinical managed in accordance to the diagnosis subtype, so tumor tissue collected on surgical procedure is not possible for all patients treated at FUCAM. Also, important, not all the patients diagnosed at FUCAM are followed up for surgical procedure in this institution, thus this reduces the universe of patients potentially candidate for sample collection.

We appreciate the helpful observation regarding the correlation between ethnicity/race, socioeconomic and health status. Accordingly, we now report the general information of

patient population attended at FUCAM, that may provide more elements to overcome this lack of information and to contextualize socioeconomic status of the patients that are part of this biobank. Specifically, in Methods, line 543, we have included a paragraph that reads:

“FUCAM provided services to vulnerable population covered by Seguro Popular de Salud (Popular Health Insurance), which aimed to extend health care coverage to the Mexican population, and gratuity interventions. Socioeconomic position measured by income among patients treated at FUCAM is reported as low income (\$218 dollars per month). Regarding the educational level of patients, institutional data showed that 6% are uneducated, 58% with an educational attainment less than high school, 22% with a high school education and 14% with some college education (data of the last two years about the general patient population of FUCAM)”

While I think it still important to present and use such valuable data as was collected and analyzed in this study, the authors need to refrain from making sweeping statements claiming that mutations in certain genes are more common compared to other ethnicities-- -I would focus just on what's new and not comment as to whether things are more or less common compared to other populations ---none of these samples are representative so who knows if they are more or less common all the samples are biased.

We appreciate reviewer's comment. As expected, Mexican breast cancer patients share important molecular features with Caucasian ones, but they also harbor features unique to Native-Mexican ancestry component, which together with a particular exposure to lifestyle, environmental, and genetic factors configure the particular oncogenic process occurring in Mexican Hispanic patients. To the best of our knowledge our study shows for the first time that a set of Mexican Hispanic patients have molecular (“omic”) features that are different from available Caucasian, African, African-American and Asian samples. Taking into consideration the reviewer suggestions, we have attempted to improve and better address our main findings, with a clearer take-away message of what we found in our HM profiled samples and the likelihood that these events were more prevalent among them with respect to samples from other samples with a different ancestry provided by different studies. In particular, we have focused our results and discussion on novel findings to address the frequency of particular well-known cancer mutations that were not immediately recognized in other datasets. Most importantly, through deeper exploration of differences in these genomic alterations we were able to discover that AKT1^{E17K} mutation were mainly seen in HM tumors. To reflect this, we have rephrased and complemented parts of our manuscript, to better outline this result.

Specifically, within the abstract in line 30 we have included a sentence that reads:

“we discovered a novel copy-number amplification in BCAR4 and the significant E17K hotspot mutation in AKT1 gene occurring in 8% of the HM tumors, a prevalence exclusively seen in this dataset”

In Results section, under Mutational landscape in HM BC tumors section, we included the following paragraph in line 235:

"Independent results from other studies were comparable with our gene mutated portraits^{9, 12, 29} (Supplementary Fig. 6a), except for a much lower frequency of CDH1 (2%) and a higher occurrence of AKT1 mutations (8%) in a subset of Mexican patients in our study (Supplementary Fig 6a)"

In line 243:

"We found no evidence for AKT1E17K significant mutation (MutSigCV, $Q < 0.1$ and frequency $>5\%$) in 4464 samples of non-Hispanic ancestry (Supplementary Fig. 6a-c), except for a group of young Korean patients³⁰ where 8% of cases ($N=4/50$) were mutated"

And line in line 249:

"Regarding the other components of the PI3K/AKT/mTOR axis, only one AKT1E17k mutant sample (10%) is co-mutated in PIK3CA, thus the altered activation of this oncogenic signaling generally occur by any one of these means in an exclusively manner. Additionally, 40% of AKT1E17k mutant patients had no additional drive cancer mutations. While 60% had further alterations, other than AKT1 mutation, which likely contribute to cancer development (e.g. truncating mutation in tumor suppressor or gain of oncogenic mutations) (Fig. 5d)"

Immune signatures are an interesting area and so I guess focusing on that aspect an important contribution of this work to the literature.

We agree with Reviewer 1. Ethnic differences in immune landscapes among populations is an interesting topic. By exploring dedicated integrative analyses via gene expression profiles, we found that Non-European ancestry women presented some modest differences with their Caucasian counterpart. There are diverse strategies to integrate DNA and RNA alterations to elucidate these immunogenic differences, but due to resource constraints, we could not perform a complete "omic" characterization on all the tumors analyzed. Based on the novelty of the current results, we're planning to conduct more research to describe in-deep these differences, mainly to provide further evidence in support of clock-like related mutational signatures; expecting that this novel knowledge will reveal more insight into the ethnic disparities of breast cancer, but this is beyond the scope of this initial report.

It is clear that the authors performed a great amount of work and analysis--and I commend them for this. However, I must admit that I found it difficult to follow the logic and presentation of the analyses. It was not clear which hypotheses were being tested and how the different experiments were connected, especially since they start with presenting other publicly available data before describing their new dataset. I potential revised presentation could start with the focused analysis of their data and then comparison with other datasets-

-with the caveats noted above--the data are what they are, but we must be clear that they do not likely represent the general populations of any of the ethnicities/races.

We thank the reviewer for this helpful observation. Accordingly, we have reorganized parts of our manuscript in hope to give a clearer picture of our main objectives and findings. In accordance to our main hypothesis, based on the plausibility that Mexican BC patients may present "omic" features unique to them. The origin of cancer is related to diverse abnormalities that can occur in many forms, including DNA mutation, deletions, amplifications, and altered gene expression programs, among others. These changes can promote abnormal cell process that can result in a cell malignant transformation. Thus cancer genomics takes advantage of recent technological to study the human genome, by analyzing the DNA and RNA of cancer cells and comparing them to normal tissues or different pathological states. This approach provides a more detail picture to understand which cell abnormalities are active or silenced in cancer cells, contributing to their uncontrolled growth. Taking advantage of a cancer genomics perspective we aimed to describe a deep genomic picture of a set of Mexican women with BC, which are not represented in the international efforts, to get comprehensive molecular portraits. As an integrative analysis of the correlation among alterations we have discovered, we have presented the section "Dissecting the biological impact of genomic complexity alterations" in which we describe the interplay between multiple DNA and RNA alterations in tumor cells. We consider that this findings represent a proof of concept of possible "omic" differences in Mexican BC patients, to encourage future extensive studies on Mexican Hispanic population.

In fact, when "omic" alterations were examined across datasets containing patients from diverse ancestries, novel findings came out that were not identified in larger datasets mainly composed by Caucasian patients. Although our HM dataset had relatively small numbers compared to the other larger efforts such as TCGA-Caucasian dataset, our sample size is similar in number to others that have been analyzed in paramount studies conducted in non-European and TCGA-African-American and TCGA-Asian samples ^{1,2}.

We have improved the presentation of our novel data throughout the manuscript to ensure a clearer description of the identified alteration in our profiled cases, that do not necessarily recapitulate ethnicities features. We first described on Results section: "Overview of the HM profiled tumors" the main features and methodological strategies our dataset; and for each "omic" evaluation we began with the description of the data obtained in the HM analyzed samples, to then compare this data with what we observed in other sample collections from diverse ancestries.

Reviewer #2 (Remarks to the Author):

The manuscript by Salido-Guadarrama et al presents analyses of genomic data of breast cancer in Mexican-Hispanic women and contrasts these findings with data from other populations. Overall the manuscript reads more as a summary of a data together with many disparate analyses rather than a hypothesis driven research article, making almost impossible

to quantify its contribution beyond adding ~100 or so breast cancer samples of Mexican Hispanic ancestry to the field. More importantly, the vast majority of analyses presented in this work are not supported by statistical significance thus leading to suggestive statements at best; no attention is given to multiple testing hypotheses correction and thus results could be stochastic fluctuations in the data and not reflect true biology.

We fully appreciate the reviewer for considering reviewing our manuscript and for his/her useful observations on it. In turn, we have performed a thorough revision of our study to try to address in the best possible way all concerns expressed by the reviewer. Please, find our point-by-point response below.

Major comments:

1. The methods section is unreadable and does not describe in sufficient detail the parameters of methods used in this work. This makes it impossible to replicate most of results presented here. The supplementary methods section presents some details on how data was generated but lacks detail on methodologies used throughout the manuscript.

We thank the reviewer for his helpful observations. Addressing the reviewer's 2 comment, in the revised version we have now included the methods section within the primary text, including a more detailed description for each method/algorithm and parameters we employed, in hope that this new information is now presented in a better way to allow the replication of bioinformatic analyses. Additionally, we believe this new version now complies with the requirements described on the Reporting Summary requested by nature research editorial.

2. Most of comparisons are not supported by significance p-values and/or confidence intervals making it impossible to differentiate between stochastic fluctuations in the data vs true biology. For example, the first paragraph of results notes several percentages of tumor types (without any confidence intervals) and cannot conclude if difference is due to number of tumors evaluated (i.e. random noise) or different admixture patterns. Similar statements exist throughout the entire manuscript.

In accordance with reviewer's observation, we now report on all the statistical methods and descriptive data. Furthermore, in an effort to provide a more robust evidence in support of the statistical significance of our results, we have included information on adjusted p-values for multiple hypotheses and confidence intervals where applicable.

Regarding the "*Distribution of intrinsic subtypes in HM population in comparison with other populations*" section to which the reviewer makes reference, we have applied comparisons of proportions among the different datasets evaluated by using a Fisher's exact test and computed p-values adjusted by Benjamini & Hochberg method through `fisher.multcomp` function on R. Pam50 subtypes proportions were described among the different datasets by an odds ratio analysis accompanied by confidence intervals. Further, differences in the intrinsic subtypes proportions among the analyzed datasets were described by Fisher exact

test and p value adjusted by Benjamini & Hochberg method. This information is reflected particularly in the following lines, within the indicated section:

On Line 115, where it reads:

"Fisher's exact Benjamini-Hochberg, BH, $p=0.041$) (Fig. 1b) and a significant reduced odd of having a Basal-like subtype (OR HM dataset vs Hispanics: 0.522, 95% CI: 0.315-0.866)"

On Line 115, where it reads:

"(HM vs AA: 46 vs 26% Fisher's exact 125 BH $p=0.041$, HM vs Nigerian: 46 vs 17%. BH $p<0.001$), where basal-like subtype showed 126 a clear enrichment (OR AA vs HM: 3.6, 95% CI: 2.2- 5.8, BH $p=0.001$; Nigerian vs HM: 127 3.3, 95% CI: 1.8-6 BH $p=0.002$)"

And on Line 115, where it reads:

"Caucasian women had significantly less probability to present a BC in an early-stage (<45 years of age) than women from HM and Hispanic populations (OR HM vs Cau: 1.5, 95% CI: 1- 2.2, BH $p=0.045$; Hispanic vs Cau: 2, 95% CI: 1.4- 2.9, BH $p= 0.008$).

Moreover, as already mention, we have made changes thorough the text to report statistically significant results on the basis of adjusted p values, similarly we now include confidence intervals.

3. Results second paragraph. Age diagnosis correlated with ethnicity; could this be due to the biased nature of the data presented here? In what ways is are the ~100 MH breast cancer samples representative of the general population of breast cancer in Mexican-Hispanic ancestry? It seems that with such a small sample size and no replication in other cohorts makes it impossible to draw conclusions about general population. Furthermore, how is the multiple testing correction handled? The authors perform several tests and should correct for multiple testing to avoid false positives.

We appreciate the thoughtful suggestion of this reviewer. We are in agreement with the impression of the reviewer that ~180 HM cases not necessarily represent the entire Mexican-Hispanic population with breast cancer. Therefore, we have reorganized and changed mayor parts of the section "Distribution of intrinsic subtypes in HM population in comparison with other populations", in line 101, which now describe our findings within the limits of their interpretation to our in-house set of samples and how they compare to other public studies in BC with samples with different ancestries and without attempting to extrapolate these results to the entire population. Similarly, we have made changes thorough the text to reflect this scope. In particular, this has been stated within the aforementioned subsection, in line 103, which reads:

"We evaluated PAM50 subtypes by defining the intrinsic clusters as described by Perou and Parker19 for BC samples in our in-house profiled dataset and other publicly available microarray gene expression data"

Regarding the correlation between age of diagnosis and ethnicity we observed in our ~180 HM cases describing an early-onset in HM patients and particular BC subtypes, are replicated in larger studies with representative samples for Mexican populations. This is now address in the Discussion section, second paragraph, line 444, which reads as follows:

"with Caucasian patients. Generally, the average age at diagnosis in Mexican women is 50 years, which represents a decade less than in the Caucasian population^{51, 52}. Notably, basal-like tumors, that are mainly composed by TN tumors, are enriched in younger patients in comparison with patients from Caucasian ethnicity. Of interest, according to Reynoso-Noverón, Mexican young patients have a larger proportion of TN tumors than their counterparts in Europe, US and Asia⁵³. Additionally, current evidence regarding racial disparities in triple-negative breast cancer conducted in a large study (N=1.15 million patients) identified 96,600 triple-negative cases and discovered that Hispanic women had higher odds of triple negative diagnosis when compared with white women⁵⁴"

Furthermore, another report by Villareal-Garza described thorough a literature review that in Latin American countries the proportion of BC in young patients aged <44 years reaches up 20%, a higher proportion compared to the incidence rates reported in developed countries³

To overcome any possible bias in the analysis of our data we have applied a Fisher exact test and p-value adjusted by Benjamini & Hochberg method to avoid false positives on the frequency of younger (<45 years at diagnosis) or older (>45 years) patients along evaluated data sets. As well, a Wilcoxon-test and adjusted p value (FDR) was computed to compare TMB distribution among the age classes. This is now further clarified in the Methods section, in line 796, which reads:

"To compare associations of the oncogenic alterations between datasets and intrinsic subtypes a Fisher exact test was performed, using the fisher.multcomp function on R, and p-values were controlled for false discovery rate with the Benjamini & Hochberg method and a multivariate Cox proportional hazards analysis was computed with SAS University Edition Statistical"

4. Distribution of subtypes across Native American ancestry component. Why divide in 4 groups and not simply correlate with genome-wide ancestry proportion? Why present these results if they are not even marginally significant (p-val>0.05)?

This is a helpful observation. We must admit that when computing a simple linear regression among genome ancestry contribution and subtype incidence we did not find any significant

result. In light of this observation and as you have mentioned, we now have avoided to report results with marginally significance. In this new version, ancestry components and their contribution are just described as part of the dataset characterization on the results section: "Overview of the HM profiled tumors" in line 86, which reads:

"A comprehensive molecular analysis of HM tumors was performed as following: whole exome sequencing (WES, n=134) to define the somatic mutation landscape, mRNA high-throughput microarray profiling (n =109) to evaluate molecular intrinsic subtypes and gene expression portraits, and finally DNA copy number profiling through genome-wide arrays (SNP6 Affymetrix arrays) (n =74 tumors and matched normal blood sample) (Fig. 1a). Clinical information of patients is shown in Table 1 and Supplementary Table 1. Latin American ancestry heterogeneity is the consequence of the admixture of pre-Columbian Native-Mexican, European and African populations¹⁵. Among the patients with genotyping information (SNP Array) in our data set and self-identified as Mexican women, Native-Mexican genetic ancestry component was identified to contribute 60% across the profiled patients, while the European component was estimated to lead 34%. A small contribution of African and Asian genetic ancestry was determined ranging from 5 to 1%, respectively"

In line with these changes, the corresponding Fig. 1, Supplementary Fig. 1a-b and Supplementary Table 1, has been modified accordingly.

How was ancestry inference performed? The supplementary note states that admixture was used with 4 components; are results robust to method/approach to infer ancestry from the genotype data?

The genomic ancestry of individuals evaluated in the HM dataset was calculated with well-known procedures for microarray genotype data reported in previous studies focused on describing genomic variation within Mexican individuals^{4,5,6}. The method has been robustly standardized at our research group.

To better explain the way Ancestry component analysis was performed. In Methods section, under the Ancestry component analysis subsection, in line 628 we have included a paragraph that reads as follows:

"A total of 77 SNP arrays (SNP 6.0, Affymetrix) were used for the ancestry analysis. Genotypes of 299,411 common SNPs evaluated in 3 genotyping platforms (SNP 6.0 and 500k arrays from Affymetrix and 1M from Illumina) were used to infer the population structure of Mexicans BC patients. Previously genotyped datasets from various sources were included to evaluate ancestry proportions in the general Mexican Population: A) 95 HapMap samples (CEU: northern European ancestry; YRI, Africans from Nigeria and CHB+JPT, east Asian population); B) 37 native Mexican samples from Tepehuano (Durango) group, Zapoteco (Oaxaca) group, and Mayas (Campeche) group from the Mexican Genome Diversity Project (MGDP)^{4,6}; and C) 161 Mexican mestizo sample from the Mexican States of Campeche, Zacatecas, Sonora, Yucatán, Tamaulipas, Guerrero, Guanajuato, and Veracruz

from MGD. Samples were subjected to four quality control tests: 1) Missing rate per person: individuals with more than 5% missing genotypes were excluded, 2) Missing rate per SNP: only SNPs with a 95% genotyping rate were included, 3) Identity by descent: (IBD) test were computed to assess quality on the full set of samples 4) Exclusion of markers that failed the Hardy-Weinberg Equilibrium test at 0.00001 significance threshold. All samples presented an optimal quality and no familial relationships were found. Principal components analysis (PCA) was used to detect population substructures based on genome wide data through ADMIXTURE v1.22 software. In accordance to the origin of the samples, the optimal K value was defined as 4, meaning that four parental groups (CEU, YRI, CHB+ JPT and NAT MEX) were considered to quantify ancestral contribution and to explain the major substructure in this set of 77 individuals. The ancestral component was identified based on the frequency of the differences between the relevant parental populations (European, Asian, African and Natives). The program STRUCTURE v2.3.4 was used to estimate the individual admixture proportions, as well as the average admixture proportions in each sample"

5. IHC marker distribution. Again the p-values are not corrected for multiple testing and the results could be just due to random fluctuations in the data (not biological). We appreciate this useful comment by the reviewer. As previously mentioned, we applied Fisher exact test and p-value adjusted by Benjamini & Hochberg method to avoid false positives on the frequency IHC subtypes, reporting only significant IHC distribution among tumors from different ancestries. Accordingly, we have inserted this information within the corresponding sections.

Specifically, in Results section, within the "*Immunochemistry (IHC) markers distribution among breast neoplasia*" subsection, in line 148, we have include the next paragraph that reads:

"Briefly, an age-related disparity on the aggressive TN and HR+/HER2+ IHC subtype was observed, with an enrichment of Hispanic women diagnosed at early-age (<45 years old, median age at diagnosis) (Fisher BH $p < 0.05$), compared with patients from non-Hispanic ancestries (Fig. 2b)"

And again, this is also stated in the Methods section, in line 796, which reads:

"To compare associations of the oncogenic alterations between datasets and intrinsic subtypes a Fisher exact test was performed, using the fisher.multcomp function on R, and p-values were controlled for false discovery rate with the Benjamini & Hochberg method and a multivariate Cox proportional hazards analysis was computed with SAS University Edition Statistical"

6. Several computational tools are used throughout with no details on the parameters used and how choice of parameters could alter the conclusions (e.g., MutSigCV, deconstructSig, etc).

We appreciate the observation. In this regard and with the intention of being as transparent as possible, we have now included a more detailed description within the methods section on all programs and steps undertaken for data collection and analysis, including programs version and relevant setting of parameters we employed in their execution and website info where applicable. Details are also provided within the document "nr-reporting-summary.pdf"

In particular, we the following modifications/insertions within the manuscript are highlighted.

In "Copy number profiles" subsection, the paragraph describing GISTIC, now includes the following description in line 667 :

"GISTIC 2.0 (v2.0.23)⁹⁴ algorithm was computed in the Genepattern environment (Deletion Threshold = 0.1, cap Values = 1.5, broad length cutoff = 0.7, Remove X-Chromosome = 0, Confidence Level = 0.90, Join Segment Size = 4, Arm Level Peel Off = 1, Maximum Sample Segments = 2000, Gene GISTIC = 1). Segmentation file were used as input. Peaks with q-value <0.15 were selected as significant"

For MutSigCV analysis, we have inserted the following information in "Somatic mutation analysis" subsection, line 693, which reads as follows:

"Next, significantly mutated genes for each ethnic group were identified using MutSigCV v1.3.01 algorithm through GenePattern module using parameters annotated in FireBrowser. Since, coverage information file was not available for the analyzed datasets, we used the "full coverage" file that is available on the GenePattern public server (exome_full192.coverage.txt) assuming full coverage. This file provides information of how the reference sequence of the human exome breaks down by gene, category and effect. In brief, the algorithm considers recurrence of mutations, genomic context, gene-expression. For this analysis, the TCGA and public samples were independently analyzed based in ethnicity. Only mutations alterations with a q value <0.05 (False Discovery Rate) were considered significant"

The paragraph describing the Mutational Signatures analysis using DeconstructSig and SigFit, line 730, now includes information on their implementation and parameters employed, which reads:

"Identification of mutational signatures in tumor samples was performed with deconstructSigs v1.8.0108. Briefly, the algorithm determines the linear combination of predefined signatures (COSMIC project v2, www.cancer.sanger.ac.uk/cosmic/signaturesnd), that more accurately reconstructs the mutational profile of a single tumor sample using SNP type variants to estimate the contribution for each signature. To determine how much of each signature is present in each sample we ran the whichSignatures function, using the

following parameters; *tumor.ref* = context matrix consisting of the counts of the mutations observed in each of the 96 possible trinucleotide contexts for each sample, *signatures.ref* = reference signatures matrix which values represent the fraction of times a mutation is seen in each of the 96-trinucleotide contexts for each selected signature, with the following parameters *tri.counts.method* = *exome2genome* method as normalization strategy and *signature.cutoff*=0.06 . We selected 14 profiles of tri-nucleotide signatures with greater biological relevance related to breast cancer. As a cross-validation method we computed mutational signatures with SigFit v2.0 package³⁵, which estimate signature contribution based on a Markov Chain Monte Carlo sampling and Non-Negative Matrix Factorization (NMF) model to fit the mutational input to a COSMIC mutational catalog. The parameters used were: *iter* = 6000, *warmup* = 3000, and *hpd_prob* = 0.90 to get the exposures and 90% highest posterior density intervals. To define the signature exposure in each sample, the calculated weights, for both used algorithms, were then multiplied by the total number of mutations in that sample. The signature with the maximum exposure was considered as the one that mostly contributes. To maximize the algorithms performance, only samples harboring a TMB higher than the median in each data sets were included (HM median TMB:0.99 mut/Mb, Asian TCGA: 0.92, Asian Kan: 1.39794, AA TCGA 0.86 and Caucasian TCGA 0.72). A mutational signature was considered to be present if it contributes more than 5% toward the entire mutational load of a sample. The probability of co-occurrence (i.e., the frequency of occurrence) of any two predominant mutational signatures in each tumor dataset was estimated as the statistical association between signatures across samples in that tumor collection, using a Fisher Exact Test. Co-occurrence was considered to be present if an association was found with FDR corrected *p*-value <0.1."

In "Mutually Exclusivity Modules in Cancer" subsection, the paragraph in line 773 has been modified to include further details on the parameters uses to run MEMo program:

*"To search for genomic alterations occurring in biological pathway context, we used the MEMo v1.1 program³⁹, which is a method for identifying network modules of significantly mutually exclusive alterations in member genes that show recurrent somatic mutations or CNA and are likely to belong to the same biological pathway or process. MEMo was run across all breast tumors profiled by genome-wide arrays and/or samples with exome sequencing, selecting alterations affecting at least 2% of the samples. MEMo was computed on single nucleotide mutations and CNV data with the following parameters *mut_sig_q_value_threshold*=0.10, (i.e., the threshold of FDR correction computed with the Benjamini & Hochberg method) and *min_number_of_alterations*=3, which selects the minimum number of alterations in a gene to be include for analysis. MEMo statistically significant modules were selected with an FDR-corrected *p*-value ≤ 0.1 ."*

7. Mutational signatures across ethnicities. Again no consideration is given to multiple testing; all *p*-values presented here are not significant after multiple testing correction.

As for the previous sections, we have now revised our analyses to include a proper multiple testing correction with the aim to assess statistical significance. Regarding the mutational signatures selection, as we have pointed out, we have done a major restructuring of this section. In this revised version, we first computed two dedicated algorithms to define and cross-validated mutational signatures landscape. This modification is pointed out in the Method section describing the Mutational Signatures analysis, in line 730, which has been previously cited. Concordantly, within the Results, in the section "Deciphering mutational signatures", in line 282, we now include the following paragraph:

"To gain further insights into the operative mutational processes in BC, the contribution of mutational signatures³³ were delineated using deconstructSigs³⁴ and SigFit³⁵ tools based on single nucleotide variations (SNV). The mutational landscape of each signature deconstructed by the two algorithms showed a highly concordant result"

For downstream analyses we focused on reliable signatures (top5), that closely resembled signatures previously identified in breast cancer. All top 5 signature proportions were analyzed through Fisher exact test and p-value adjusted by Benjamini & Hochberg method for multiple hypothesis. In line 290, reads as:

"Top frequently mutated signatures among patients from different ancestry deconstructed with both algorithms included: Sig1, Sig2, Sig3, Sig13 and Sig7 (Supplementary Fig. 8c)"

And we highlight our main results in the paragraph in line 300, which reads:

"HM tumors from young patients presented a higher Sig 1 prevalence than AA TCGA and Asian Kant individuals (HM 25% vs 0% in AA and 3% in Asian Kan BH $p < 0.005$) (Fig. 6b). Although not differences were observed in the proportion of Sig1 among HM and Caucasian young patients, Mexican patients enriched in Sig1 were diagnosed at an earlier age (HM median age 39 years, while in Caucasian patients 44 years). Given the limited sample size of young patients in Asian TCGA collection we did not make a comparison against them (N=1)"

In addition, all of these changes are now properly reflected within corresponding figures and supplementary data, namely, Fig. 6a, Supplementary Fig.8b and Supplementary Table 9. Similarly is addressed within the Discussion section, in line 484:

"Mutations contributed by signature 1, which exhibits clock-like properties generally correlated with age^{34, 73, 74}, were strongly represented in individuals from HM ancestry in contrast to the rest breast cancer women evaluated, even though, our set of tumors was enriched in younger patients. Also, a significant co-occurrence of the contribution of signature 1 with the APOBEC-related signatures were only observed in HM profiled tumors"

Finally, we have included a new paragraph in line 326, addressing the probability of co-occurrence (i.e., the frequency of occurrence) of any two predominant mutational signatures in each tumor dataset, which reads as follows:

"We hypothesized that exploring the interplay between the contribution of mutational processes will further improve our understanding to uncover new relations between cancer drivers. To this end, we compute the co-currency of the two high-confidence predominant mutational signatures in each sample. 22% of the evaluated samples did not present any co-occurrent event among the top 5 mutational signatures (Fig. 6c). For the remaining 78%, particularly in the HM profiled samples, the mayor co-occurrent event was between Sig1::Sig2/13 (32%, OR: 5.7 BH p:0.008), in contrast to tumors belonging to other ancestries where the most common co-occurrent events are Sig2/13::Sig3 or Sig3::Sig2/13 (Fig. 6d, Supplementary table 9). Distinctive co-occurrence pattern might underline different associations with the exogenous or endogenous mutagenic processes, as well as, unique variability in the subclones abundances across samples. Overall, these results suggest that the HM population, shows distinctive differences in the spectrum of mutational signatures contributions that make up their mutational landscape"

1. Pitt JJ, *et al.* Characterization of Nigerian breast cancer reveals prevalent homologous recombination deficiency and aggressive molecular features. *Nat Commun* **9**, 4181 (2018).
2. Kan Z, *et al.* Multi-omics profiling of younger Asian breast cancers reveals distinctive molecular signatures. *Nat Commun* **9**, 1725 (2018).
3. Villarreal-Garza C, Lopez-Martinez EA, Munoz-Lozano JF, Unger-Saldana K. Locally advanced breast cancer in young women in Latin America. *Ecancermedicalscience* **13**, 894 (2019).
4. Silva-Zolezzi I, *et al.* Analysis of genomic diversity in Mexican Mestizo populations to develop genomic medicine in Mexico. *Proc Natl Acad Sci U S A* **106**, 8611-8616 (2009).
5. Reich D, *et al.* Reconstructing Native American population history. *Nature* **488**, 370-374 (2012).
6. Moreno-Estrada A, *et al.* Human genetics. The genetics of Mexico recapitulates Native American substructure and affects biomedical traits. *Science* **344**, 1280-1285 (2014).

For your convenience we include the text with the tracked changes

Comprehensive omic characterization of breast cancer in Mexican-Hispanic women

Sandra L. Romero-Córdoba^{1,2+*}, Ivan Salido-Guadarrama³⁺, Rosa Rebollar-Vega^{2,4,7+},
Veronica Bautista-Piña⁵, Carlos Dominguez-Reyes⁵, Alberto Tenorio-Torres⁵, Felipe
Villegas-Carlos⁵, Juan C. Fernández- López⁶, Laura Uribe-Figueroa², Luis Alfaro-Ruiz^{2*},
Alfredo Hidalgo-Miranda^{2*}

¹ Biochemistry Department, Instituto Nacional de Ciencias Médicas y Nutrición Salvador Zubirán, Mexico City, Mexico. ² Cancer Genomics Laboratory, National Institute of Genomic Medicine, México. ³ Computational biology core, Instituto Nacional de Enfermedades Respiratorias Ismael Cosío Villegas. ⁴ Genomics Laboratory, Red de Apoyo a la Investigación, Universidad Nacional Autónoma de México-Instituto Nacional de Ciencias Médicas y Nutrición Salvador Zubirán, México City 14080, México. ⁵ Instituto de Enfermedades de la Mama FUCAM, México. ⁶ Computational genomics laboratory, National Institute of Genomic Medicine. ⁷ Biomedical Sciences PhD Program, Autonomous National University of Mexico, Mexico City, Mexico.

+Equal contribution

*Corresponding authors

Abstract:

Breast cancer is a public health problem that affects millions of women worldwide. It is recognized as a clinically and molecular heterogeneous pathology, but the genomic basis of its variability remains poorly understood, with important shortcomings regarding the consequences of ethnicity disparities in genomics studies. Moreover, most of the comprehensive molecular characterization is predominantly centered on Caucasian population. Through omics portraits of DNA and RNA we explored the molecular features of breast cancers in Mexican-Hispanic (HM) women and confronted them against to those observed in benchmarked public multi-ethnic datasets. HM patients presented an earlier onset of the disease, in particular in the aggressive Basal-like/triple negative and Luminal B subtypes. Among recurrent DNA alterations we discovered a novel copy-number amplification in BCAR4 and the significant E17K hotspot mutation in AKT1 gene occurring in 8% of the HM tumors, a prevalence exclusively seen in this dataset. An integrative analysis revealed an important role of the AKT1/PIK3CA axis as a potentially druggable target. Computational analysis allowed us to identify age-related COSMIC (v2) signature 1 to be present in a much broader spectrum of HM women, even in young patients, than in those from other ethnicities. On the basis of transcriptome immunogenomic analysis across human population we identified that luminal breast tumors with HM ancestry presented an enhanced immunogenic phenotype than their Asiatic and Caucasian counterparts. Our data provides novel insights into molecular phenotypes and mechanisms underlying carcinogenesis in a group of Mexican Hispanic women. In conclusion, this study provides a conceptual framework to better understand ethnic disparities of breast cancer.

Introduction

Breast cancer (BC) is the most common neoplasia in women around the globe and it represents an increasingly urgent health problem worldwide. Out of 19.7 million cases over the next ten years, 10.6 million will occur in low-and middle-income countries¹. In Mexico, breast tumors represent the main cause of cancer in women and epidemiological projections estimate the number of new cases and the mortality rates will increase in the next years².

BC is a heterogeneous disease, both at clinical and molecular levels, as evidenced by the
various therapeutic rate responses and by the identification of intrinsic biological and
molecular subtypes that present unique features which may enhance cancer cell fitness
and increase clinical risk and therapeutic resistance^{3, 4, 5, 6}. In the clinical practice, the
immunohistochemical markers estrogen (ER); progesterone (PR) and human epidermal
growth factor receptor 2 (HER2) are used to guide diagnosis and treatment decisions⁷.
Molecular classification of breast tumors based on gene expression patterns has also
been successfully translated into tests to support clinical decisions. Gene expression
profiling, particularly the PAM50 intrinsic subtyping signature, has identified at least five
categories of breast tumors: luminal A (LumA), luminal B (LumB), HER2-enriched (HER2),
basal-like (basal) and claudin-low tumors^{3, 4, 6, 8}, each one with distinctive oncogenic
features. Further, increased genomic instability is a relevant feature of breast tumors, both
at point mutation and at somatic DNA copy number alterations (SCNA) levels. These two
processes play an important role in activating oncogenes or in inactivating tumor
suppressors genes^{9, 10, 11}, and, they can also provide important information about BC
biology^{12, 13}.

During the last years, Hispanic/Latino populations represents the largest and fastest
growing minority population in the United States¹⁴. In this study, we referred as
hispanic/latino population to those BC patients from Mexico, Caribbean, Central and South
America that share similar admixtures of Native American, European, and African
ancestries¹⁵. Hispanics constituted 18.1% of the total population in United States in 2017,
and this will increase to 35% by 2050¹⁴. However, there is limited information about the
genomic alterations of BC in Mexican-Hispanic (HM) and Latino populations¹⁶. Most of the
genomics discoveries and exploratory studies focus on data obtained almost
predominantly from Caucasian populations¹⁷. A more comprehensive description of the
biological landscape of breast tumors between different ethnicities could provide important
insights into the BC disparities¹⁸. To get a comprehensive understanding of the molecular
architecture of breast cancer in a set of Mexican patients, we evaluated somatic mutations,
SCNA and gene expression patterns on 204 tumors from HM women and we
comparatively describe their genomic context in contrast to patients with African, African-
American, Asian and Caucasian ancestry. To the best of our knowledge, this dataset
represents the largest breast cancer genomics characterization of breast tumors in

Mexican women, and, the results from this work highlights unique molecular features of
HM breast cancers, as well as, characteristics common to all BC cases.

**Results**

**Overview of the HM profiled tumors**

A comprehensive molecular analysis of HM tumors was performed as following:
whole exome sequencing (WES, n=134) to define the somatic mutation
landscape, mRNA high-throughput microarray profiling (n =109) to evaluate molecular
intrinsic subtypes and gene expression portraits, and finally DNA copy number profiling
through genome-wide arrays (SNP6 Affymetrix arrays) (n =74 tumors and matched normal
blood sample) (Fig. 1a). Clinical information of patients is shown in Table 1 and
Supplementary Table 1. Latin American ancestry heterogeneity is the consequence of the
admixture of pre-Columbian Native-Mexican, European and African populations¹⁵. Among
the patients with genotyping information (SNP Array) in our data set and self-identified as
Mexican women, Native-Mexican genetic ancestry component was identified to contribute
60% across the profiled patients, while the European component was estimated to lead
34%. A small contribution of African and Asian genetic ancestry was determined ranging
from 5 to 1%, respectively (Supplementary Fig. 1a-b, Supplementary Table 1).

**Distribution of intrinsic subtypes in HM population in comparison with other** 102 **populations**

We evaluated PAM50 subtypes by defining the intrinsic clusters as described by Perou
and Parker¹⁹ for BC samples in our in-house profiled dataset and other publicly available
microarray gene expression data (Fig. 1a). PAM50 classification identified four different
tumor subtypes (Supplementary Fig. 2a) and normal-like samples, which were discarded
due to possible normal cell majority content. Of note, our HM tumors (N=109) contain 46%
LumA tumors, 28% LumB, 15% HER2-enriched and 12% Basal-like (Fig. 1b). Compared
to other biospecimen collections including Hispanic patients (N=313) from M.D. Anderson
Cancer Center (GSE16716, N=41, Basal-like: 23%), Instituto Nacional de Enfermedades
Neoplasicas in Lima, Peru together with the Centro Medico Nacional de Occidente in
Guadalajara, Mexico (GSE20271, N=81, Basal-like: 29%), Hospital San Jose Tec de
Monterrey, Mexico (GSE75678, N=53, Basal-like: 22%) and LACE and Pathways cohorts
from USA (N=138, Basal-like:27%), our data set presented a smaller proportion of Basal-
like tumors (12% vs 26% -media proportions of other studies-, Fisher's exact Benjamini-
Hochberg, BH, p=0.041) (Fig. 1b) and a significant reduced odds of having a Basal-like

subtype (OR HM dataset vs Hispanics: 0.522, 95% CI: 0.315-0.866) (Fig. 1b,
Supplementary Table 2). This differences in subtypes frequency among different Hispanic
groups could be partly ascribable to differences in the number of tumors evaluated or to
different admixture patterns within these cases.

When comparing the distribution of particular intrinsic subtypes in breast tumors among all
the human populations, we encountered that, LumA was the predominant subtype in all
ethnic groups except in AA and Nigerian populations (HM vs AA: 46 vs 26% Fisher's exact
BH p=0.041, HM vs Nigerian: 46 vs 17%. BH p<0.001), where basal-like subtype showed
a clear enrichment (OR AA vs HM: 3.6, 95% CI: 2.2- 5.8, BH p=0.001; Nigerian vs HM:
3.3, 95% CI: 1.8-6 BH p=0.002) (Fig 1b-c). Among the other subtypes, there was a
substantial overlap in the frequencies between Hispanic and non-Hispanic patients and
differences did not reach statistical significance (Fig 1b-c). Age at diagnosis of the
evaluated women was significantly correlated with ethnicity (Fig. 1d, Supplementary Fig.
2b), for instance Caucasian women had significantly less probability to present a BC in an
early-stage (<45 years of age) than women from HM and Hispanic populations (OR HM vs
Cau: 1.5, 95% CI: 1- 2.2, BH p=0.045; Hispanic vs Cau: 2, 95% CI: 1.4- 2.9, BH p= 0.008).
Regarding the comparison among all Hispanic populations, we did not detect any
significant difference between HM, Peruvian and US-Latinas women. From samples with
available gene expression and age at diagnosis information, aggressive Basal-like subtype
is significantly enriched in Hispanic young women (<=45 years of age) in comparison with
Caucasian and Asian patients (BH Fisher p=0.009) (Supplementary Fig. 2c).

**Immunochemistry (IHC) markers distribution among breast neoplasias**

We found high (~70%) overlapping between IHC-defined subtypes (based on St. Gallen
guidelines²⁰) and RNA-based PAM50 centroids intrinsic subtypes (Supplementary Table 2
and Supplementary Fig. 3a-b). In tumors for which there is available IHC information we
observed a strong enrichment of triple negative (TN) subtype in AA (38% vs 20 in
Hispanic, BH p=0.007) and Nigerian (43%, BH P: 0.001) population, as previously
described in the literature^{21, 22} (Fig. 2a). We then evaluated the associations between
ancestry and clinical characteristics. Briefly, an age-related disparity on the aggressive TN
and HR+/HER2+ IHC subtype was observed, with an enrichment of Hispanic women
diagnosed at early-age (<45 years old, median age at diagnosis) (Fisher BH p<0.05),
compared with patients from non-Hispanic ancestries (Fig. 2b).

**Analysis of somatic copy number alterations in BC**

SCNAs are important alterations affecting larger fraction of the cancer genome than do
any other type of somatic genetic alteration, particularly in BC²³. GISTIC analysis on HM
tumors, revealed relevant SCNA (retained as likely-significant +1 or significant +2) for a total
median number of 9381 recurrent events in 140 regions (3622 gene amplifications and
5075 deletions were detected) (Fig. 3a). The overall proportion of SCNA is similar to what
is observed in the other data sets, exception made for AA tumors which showed
significantly more number of SCNA events than tumors from HM ancestry (Wilcoxon FDR
$p=0.0095$) (Fig. 3a) (Supplementary Table 3). In general, we observed much more
amplifications than deletions among all the ethnicities evaluated (Supplementary Fig. 4a-
c). Similarly, the median fraction of the cancer genome with copy number changes, termed
tumor SCNA burden (TSCB), is similar between the HM, Caucasian and Asian ethnicities
(HM 23%, Caucasian 23% and Asian 29%) and lower than the observed in the AA tumors
(35%) (Fig. 3b). This can suggest a higher dependency of AA tumors on SCNA alterations.

Of notice, we detected novel SCNA events exclusively present in HM patients. Among the
most significant, we found the amplification of the region 16p, which harbors genes such
as *SNN*, *LITAF*, *ZC3H7A*, *TXNDC11*, *RMI2* and the oncogene *BCAR4*, implicated in
endocrine resistance in human BC cells^{24,25}. Likewise, we detected the 17p amplification,
where *SPECC1* gene is located (Fig. 3c-d, Supplementary Fig. 4d, Supplementary Table
4). Additionally, HM tumors harbor well-known SCNA of BC, among which gain of
chromosomal regions 8q, 11q and 17q that contain oncogenes such as *MYC*, *CCND1* and
*ERBB2*. Similarly, we detected losses on chromosomes 7q, 8p, 13q, 17p,
containing *MLL3*, *CSMD1*, *RB1* and *MAP2K4* genes. Even though most of the significant
SCNA at arm level (q value > 0.25) are also present in the other ethnical groups, they
occur at different prevalence in HM samples (Fig. 3c-d, Supplementary Fig. 4c-d). On
contrary, some amplifications (3p,11p,15q, 19q, 21q) or deletions (1q,6p,8q,12p/q and
22q) mostly detected in BC tumors from different populations were not identified in our HM
dataset (Fig. 3c-d, Supplementary Table 4).

Gene-level amplification/deletion threshold values computed by GISTIC, considering only
high-level amplifications (+2) and deep deletions (-2), were used to compare significant
events against patients from other ethnicities. Collectively, the most frequently altered

genes by DNA copy-number alterations at focal level were the amplifications of
*ERBB2* (17q17q12), *WHSC1L1* (8p11.23), *CCND1*(11q13.3), *ORAOV1*(11q13.3) and
*MYC* (8q24.21) (Fig. 3e, Supplementary Table 4). Recurrent focal copy number
losses included *CSMD1* (8p23.2), *SHISA6* (17p12), and *DMRT1/2/3* genes (9p24.3) (Fig.
3f, Supplementary Table 5) (Supplementary Fig. 4d). Gained or lost regions as identified
by GISTIC2 had significant corresponding peaks within the other ethnic groups evaluated.
A collection of functional relevant amplifications and deletions was curated by integrating
the GISTIC 2.0 analysis and the OncoKB database. Gene-specific copy number variants
that were labeled as oncogenic, likely oncogenic or predicted oncogenic are reported in
supplementary table 5.

**TMB and cancer driver mutations across BC patients from multi-ethnic profiles**

Tumor mutation burden (TMB, mutation/Mb) was computed for samples with available
exome sequencing data in each sample collection, including 132 HM, 119 AA, 684
Caucasian, 185 Asian (Kan. et.al), 57 Asian-TCGA and 250 Japanese (Hatakeyama,
et.al²⁶). Median TMB of tumors from HM patients (median TMB:0.99, range:0.18-16.10)
was significantly higher than those from Caucasian patients in TCGA (median TMB: 0.72,
range:0-115.32, FDR, $p < 0.001$) and those from AA samples (median TMB of 0.86
mut/Mb, range: 0.06- 3.67, FDR $p = 0.04$), but showed no significant difference with Asian
tumors in TCGA (median TMB: 0.92, range: 0.32-13.66) nor with Japanese patients
(median TMB 1.08, range:0.06-18.16) (Fig. 4a, Supplementary Table 6). In contrast, HM
samples had a lower TBM than tumors from young Asian women (Kan *et. al.* median TMB:
1.39794, range: 0.47712-2.49969 FDR $p = 0.0021$) which exhibit higher mutation rates (Fig.
4a, Supplementary Table 6), and this might explain the occurrence of breast cancer at
early life-time.

We, then decided to explore in more detail the relationship between TMB and age. HM
patients did not present statistically significant difference in median TMB values between
younger (<45 years old) and older patients, as occurs in AA and Asian Kan data sets, and
contrarily to what is observed in tumors from Caucasian and Asian ancestry (TCGA and
Japanese dataset) (Fig. 4b) (adjusted $p < 0.05$). Further, we only observed a significant
correlation between somatic mutation burden and age in Asian data sets ($R = 50\%$ and
24% $p < 0.05$) (Supplementary Fig. 5).

Next, we sought to estimate the number of mutations detected in tumors potentially acting
as drivers, whose deregulation contributes directly to tumor progression²³. Up to 84% of all
HM tumors contain at least one driver point mutation, with an average of 2.65 driver
mutations, as recently published²⁷ and in agreement to the driver mutation rates computed
for the other ancestries (Fig. 4c, Supplementary Table 7). Interestingly, the numbers of
point driver mutations remain remarkably stable, even in highly mutated tumors (>1
mut/Mb) (Fig. 4d). It is relevant to point out that an important fraction of the total driver
mutations is annotated as oncogenic (19%) or tumor suppressor alterations (21%) in HM
tumors (Fig 4e). Finally, the frequency of the top mutated cancer driver genes (> 5%) in
HM breast tumors varies among the evaluated ethnicities, with *PIK3CA*, *TP53*, *MAP3K1*
and *PTEN* having the highest similarities (Fig. 4f, Supplementary Table 7).

**Mutational landscape in HM BC tumors**

We then analyzed the non-silent coding-protein mutations affecting HM tumors (N=132) by
computing MutSigCV²⁸. The significantly mutated cancer genes (q.value=<0.1) presented
in more than 5% of HM samples are *PIK3CA* (28%), *TP53* (20%), *AKT* (8%) and *MAP3K1*
(5%) (Fig. 5a) (Supplementary Table 8). Independent results from other studies were
comparable with our gene mutated portraits^{9, 12, 29} (Supplementary Fig. 6a), except for a
much lower frequency of *CDH1* (2%) and a higher occurrence of *AKT1* mutations (8%) in a
subset of Mexican patients in our study (Supplementary Fig 6a).

Of note, 11 (8%) HM cases harbored non-silent mutations in *AKT1*, from which 10 tumors
present the mutation Glu17Lys (E17K) in the PHb domain (Fig. 5a-b, Supplementary Table
8). E17K mutation were most common in the positive hormone receptors tumors (HR+)
of the HM dataset (Fig. 5c). We found no evidence for *AKT1*^{E17K} significant mutation
(MutSigCV, Q < 0.1 and frequency >5%) in 4464 samples of non-Hispanic ancestry
(Supplementary Fig. 6a-c), except for a group of young Korean patients³⁰ where 8% of
cases (N=4/50) were mutated. E17K mutation results in a pathogenic activating mutation
according to OncoKb what in turn promotes an active PI3K/AKT/mTOR pathway signaling
(Fig. 5d, Supplementary Fig. 6d). Regarding the other components of the
PI3K/AKT/mTOR axis, only one *AKT1*^{E17k} mutant sample (10%) is co-mutated in *PIK3CA*,
thus the altered activation of this oncogenic signaling generally occur by any one of these
means in an exclusively manner. Additionally, 40% of *AKT1*^{E17k} mutant patients had no
additional drive cancer mutations. While 60% had further alterations, other

than AKT1 mutation, which likely contribute to cancer development (e.g. truncating
mutation in tumor suppressor or gain of oncogenic mutations) (Fig. 5d).

We also identified potentially novel recurrently mutated genes in the HM tumors exhibiting
significant mutation prevalence ($\geq 3.0\%$), including *MRPL37* and *SLC16A8* (q value=
2.72E-02 and 1.36E-02, respectively), but none of them have been previously reported in
other datasets (Supplementary Table 8). Although, all the novel mutations are predicted to
be passenger alterations, most of them have a high deleterious oncogenic capacity and a
functional consequence of missense or nonsense change (Supplementary Fig. 6e).

**Clinically actionable genomic alterations in HM population**

To analyze the frequency of clinically actionable mutations, we retrieved annotations of
targetable genomic alterations using the cancer hotspot database³¹ and OncoKB
classification system³². 46% of the HM tumors exhibited hotspot mutations, particularly in
*PICK3CA* (H1047L, E542K, E545K, H1047L) and in *AKT1* (E17K) genes (Fig. 5e). Beyond
hotspot mutations, other potentially actionable variants were also evaluated. Therefore,
77% of HM tumors harbored oncogenic alterations in 46 genes considered potentially
targetable based on various clinical and preclinical evidences (Fig. 5f), with 25% of them
having either likely-oncogenic or oncogenic status as reported by OncoKB (i.e., gain of
function (75%) or loss of function (10%) mutations) (Fig. 5g).

Some worth-mention actionable mutations include activating ERBB2 mutations detected in
4 samples lacking HER2 amplification which suggest that these patients could benefit from
anti-HER2 therapy such as Neratinib and Ado-Trastuzumab. Similarly, we found tumors
harboring *AKT1*^{E17K} driver mutation, considered actionable, that may benefit from anti-AKT
therapy (Supplementary Fig. 7).

**Deciphering mutational signatures**

To gain further insights into the operative mutational processes in BC, the contribution of
mutational signatures³³ were delineated using deconstructSigs³⁴ and SigFit³⁵ tools based
on single nucleotide variations (SNV). The mutational landscape of each signature
deconstructed by the two algorithms showed a highly concordant result (Supplementary
Fig. 8a). Most individual cancer exomes exhibit more than one mutational signature (Sig)
and many different combinations of signatures were observed among BC tumors, across
all populations (Supplementary Fig 8b). In general, 8 signatures were robustly detected

among BC tumors from the 30 COSMIC signatures v_2^{36} , but it seems that some tumors
have a more complex repertoire of mutational processes than others (Supplementary Fig.
8b, Supplementary Table 9). Top frequently mutated signatures among patients from
different ancestry deconstructed with both algorithms included: Sig1, Sig2, Sig3, Sig13 and
Sig7 (Supplementary Fig. 8c).

The signature with the highest mutational probability was considered as the predominant
driving signature in each sample (Fig. 6a). Overall in our HM dataset, Signature 1,
associated to cytosine deamination mutational processes and clock-like properties, was
the most common predominant mutational signature contributing in average with 45% of
the top mutational signatures (Fig. 6a). Sig1 is characterized by prominence of C>T
changes (Supplementary Fig. 9a) and enriched distribution of hormone receptors markers
(Supplementary Fig. 9b). HM tumors from young patients presented a higher Sig 1
prevalence than AA TCGA and Asian Kan individuals (HM 25% vs 0% in AA and 3% in
Asian Kan BH $p < 0.005$) (Fig. 6b). Although not differences were observed in the
proportion of Sig1 among HM and Caucasian young patients, Mexican patients enriched in
Sig1 were diagnosed at an earlier age (HM median age 39 years, while in Caucasian
patients 44 years). Given the limited sample size of young patients in Asian TCGA
collection we did not make a comparison against them (N=1). Moreover, mutations
contributed by signature 1 showed a greater rate in HM older patients (>45 years old)
which accounted for the highest frequency in comparison with other patients with different
ancestries (HM 51% vs 20% in AA, 28% in Cau, 31 in Asian TCGA and 14% in Asian Kan
BH $p < 0.05$) (Fig. 6b). Overall these data suggest the possibility that the chronological age
does not completely recapitulate the biological age of tumors in the evaluated HM patients.
Signature 2 was the next most common predominant mutational signature (31%) (Fig. 6a).
Since etiologies of signature 2 and 13 are attributed to a common activity of the
AID/APOBEC family, we collapsed these signatures into a single one (Signature 2/13)³⁷.
Signature 2/13 is characterized primarily by C>T and C>G mutations (Supplementary Fig.
9a) and is enriched in HR+ phenotype (Supplementary Fig. 9b). Tumors belonging to this
signature present an overwhelming number of mutations in contrast to the other signatures
(Supplementary Fig. 9c). On detail, HM tumors with high contribution of this signature were
the most mutated tumors among AA, Caucasian and Asian Kan evaluated breast cancer
cases (Wilcoxon BH $p < 0.05$) (Supplementary Fig 9c). Signature 3, characterized by a
BRCAness phenotype, was the third most frequent processes in our tumor collection

(18%). This signature was preferentially enriched in triple negative phenotype
(Supplementary Fig. 9b). Finally, signature 7, resulting from DNA adducts formed by the
action of UV light, were observed in 1% of HM samples.

We hypothesized that exploring the interplay between the contribution of mutational
processes will further improve our understanding to uncover new relations between cancer
drivers. To this end, we compute the co-currency of the two high-confidence predominant
mutational signatures in each sample. 22% of the evaluated samples did not present any
co-occurrent event among the top 5 mutational signatures (Fig. 6c). For the remaining
78%, particularly in the HM profiled samples, the mayor co-occurrent event was between
Sig1::Sig2/13 (32%, OR: 5.7 BH p:0.008), in contrast to tumors belonging to other
ancestries where the most common co-occurrent events are Sig2/13::Sig3 or Sig3::Sig2/13
(Fig. 6d, Supplementary table 9). Distinctive co-occurrence pattern might underline
different associations with the exogenous or endogenous mutagenic processes, as well
as, unique variability in the subclones abundances across samples. Overall, these results
suggest that the HM population, shows distinctive differences in the spectrum of
mutational signatures contributions that make up their mutational landscape.

**Dissecting the biological impact of genomic complexity alterations**

Dissecting the coordinated mechanisms by which some specific genomic alterations can
alter tumor biology would extend our comprehension on how these somatic events may
impact tumor phenotypes. In our dataset, we confirmed a heterogeneous picture of DNA
alterations, represented by correlation between TMB and the tumor altered fraction (TAF -
SCNA events). Some tumors exhibited either high number of mutations (>1mutation/Mb)
or high numbers of copy number alterations (>5% of tumor altered fraction), but not both
(Fig. 7a). Interestingly, we also found tumors exhibiting a relatively high TMB and TAF, as
well as tumors without any high mutational or SCNA events (i.e., low TMB and TAF) (Fig.
7a). Prevalence of high TAF and high TMB phenotype was similarly observed in HM, AA
and Asian women (~ 43% in each dataset), but less frequent in Caucasian population
(35%) (Fig. 7b, Supplementary Table 10). This molecular phenotype may be explained by
clonal diversity that enhances higher intra-tumor heterogeneity and differences in genomic
instability.

SCNAs have critical roles in activating oncogenes and in inactivating tumor suppressors.
To determine the “cis” consequences of SCNAs in HM population, a correlation analysis
was computed between significant SCNA profiles and significant differential expression of
genes ($\log_{2}FC > 1.3$, $\text{adj } p\text{-value} < 0.05$) contained in the aberrant locus. A total of 184
CNA-mRNA pairs were significantly correlated ($R=30\%$, $\text{adj } p\text{-value}= 0.05$) (Fig. 7c,
Supplementary Table 10). Most positive cis-associations were found for genes on
chromosome 1, 3, 12,13, 14 and 15. To better understand the functional consequences of
these aberrations, we investigated the GO Biological processes and KEGG terms to which
correlated genes are annotated. Not surprisingly, significantly enriched terms in amplified
regions included cancer pathways, such as DNA repair mechanism, histone acetylation
and chromatin remodeling complex (Supplementary Fig 10). Conversely, multiple deletion
events comprises genes with roles in the control of fatty-acid and amino acid metabolic
pathways, traffic and localization of vesicles, regulation of cytoskeleton, among others
(Supplementary Fig 10). These observations suggest a convergence of multiple CNA
targets on a common set of biological functions important to maintain different hallmarks of
cancer³⁸.

[revised manuscript text omitted]

**Discussion**

The number of samples analyzed in the literature from ethnic minorities in BC is still
relatively small, which limits the ability to detect ethnicity-specific molecular alterations^{18,}
433⁴⁷. To our best knowledge, this study represents the largest genomic analysis of BC among
434 patients with HM ancestry residing in Mexico. To describe a deep biological portrait of the
435 molecular features of HM and American-Hispanic BC patients, we compared multi-omics
profiles between our set of tumors and public data from other ethnicities mainly consisting
of Caucasian, Asian, African and Afro-American.

Recent studies have shown that breast tumors in young women exhibit more aggressive
characteristics than those occurring in older patients^{48,49}. Public systems need to face this
growing health problem, mainly in developing countries where the incidence of BC is rising
442⁵⁰ and a higher proportion of women debuts at younger stages⁴⁸. In our analysis, younger
women (<45 years of age) from Hispanic datasets accounted for a higher proportion
compared with Caucasian patients. Generally, the average age at diagnosis in Mexican
women is 50 years, which represents a decade less than in the Caucasian population^{51,52}.

Notably, basal-like tumors, that are mainly composed by TN tumors, are enriched in
younger patients in comparison with patients from Caucasian ethnicity. Of interest,
according to Reynoso-Noverón, Mexican young patients have a larger proportion of TN
tumors than their counterparts in Europe, US and Asia⁵³. Additionally, current evidence
regarding racial disparities in triple-negative breast cancer conducted in a large study
(N=1.15 million patients) identified 96,600 triple-negative cases and discovered that
Hispanic women had higher odds of triple negative diagnosis when compared with white
women⁵⁴.

The heterogeneous transcriptional phenotypes observed in women with BC belonging to
different ethnicities, is in part influenced by the alterations in cancer genomes such as

mutations and SCN. Even when a well concordance of SCNA profiles were observed
with other ethnicities, differences exist in the frequencies of these genomic alterations
alongside the detection of unique SCNA in tumors from HM women.

Moreover, we identified recurrent alterations that particular affect HM tumor genomes;
such as the enrichment of AKT1^{E17K} mutation in HR+ tumors, with a prevalence of 8%.
Interestingly, although this amino acid alteration was identified as a recurrent hotspot
mutation in BC^{31, 55}, other profiles report a lower frequency, ranging from 1.4% to 5.9% in
different ethnicities^{56, 57, 58}, with a mean frequency of 3.8 %. In accordance with these data,
our in-silico analysis of large-scale sequencing studies failed to detect frequently
significantly mutations at AKT gene in other populations. We only detected a similar
frequency in young BC patients (<35 years) from South Korea, with 8% (4/50)³⁰.

The PI3K–AKT–mTOR signaling pathway is one of the most frequently dysregulated
pathways in human cancers^{55, 59, 60}, with repercussions in key cellular processes, such as
metabolism, independent cell proliferation⁶¹, cell invasion⁶², endocrine receptor
deregulation^{63, 64} and resistance to therapy⁶²; and consequently supporting cancer cell
programs⁶⁵. The pathway can be aberrantly activated through multiple mechanisms,
including diverse AKT mutations^{66, 67}. E17K mutation activates AKT1 by recruiting it to the
membrane through a PI3K-independent mechanism, resulting in the activation of
PI3K/AKT/mTOR signaling pathway⁶⁸. Thus, AKT1 mutations have emerged as an
attractive druggable target and there is promising clinical data in ER+ ductal BC patients
harboring AKT1^{E17K} mutation treated with the pan-AKT targeted inhibitors AZD5363 and
MK-2206^{66, 69, 70, 71}. More recently, ipatasertib, another ATP-competitive AKT inhibitor, in
combination with paclitaxel increase progression free survival time⁷².

It is possible that biological and environmental factors may dictate evolutionary dynamics
of a tumor. This assumption may explain the observed mutational signatures portraits and
their differences between racial groups. Mutations contributed by signature 1, which
exhibits clock-like properties generally correlated with age^{34, 73, 74}, were strongly
represented in individuals from HM ancestry in contrast to the rest breast cancer women
evaluated, even though, our set of tumors was enriched in younger patients. Also, a
significant co-occurrence of the contribution of signature 1 with the APOBEC-related
signatures were only observed in HM profiled tumors. This may be explained by a
biological epigenetic clock as suggested by Kresovich and collaborators⁷⁵, that propose

that age acceleration is associated with increased BC risk⁷⁶. Thus, it's possible that
chronological age not always correlates with biological age, presumably, by the
accumulation of biological changes in individuals that undergo at different rates because of
different carcinogen and environmental exposures or neoplastic changes^{73, 77}.
Interestingly, when contrast age at diagnosis and TMB no differences were identified
between Mexican younger and older patients, suggesting that cancer drivers are not yet
completely described, mainly in younger patients. This can explain in part why in Mexico
breast cancer is diagnosed at younger ages compared to what is reported in other
populations.

Understanding the utility of molecular characterization, as well as its drawbacks is crucial
to take informed decisions regarding clinical care⁷⁸. Our data extends the knowledge of
Hispanic population and contributes with the understanding of the biological ethnic
disparities observed in the clinical management. There are certain limitations to this study
including the number of samples, although here we report the first relatively large and
comprehensive characterization (at genomic and transcriptomic level) of breast tumors in
Mexican women, the study may not reflect all population rates and was not conducted in a
epidemiologic design. Also, omics characterization has limited number of matched tumor
samples. As we took advantage of public data, to have a robust multiethnic comparison
and overcome size limitations, our study is constrained by available information, for
instance the clinical data in a considerable number of samples was incomplete, which
impedes the assessment of clinical features impact, such as the tumoral grade. Moreover,
to compare gene expression data profiled with different array versions we needed to limit
the number of evaluated genes to the common transcripts among all the arrays. Overall,
our results reinforce the need for future studies integrating molecular characterization,
over-coming the lack of populations representation in genetic research, to better
understand the causes of disparities in BC outcomes and minimize the inequitable access
to precision medicine through a more effective intervention that may reduce health
disparities. Therefore, additional efforts are still urgently needed to identify large sample
data sets of underrepresented Hispanic patients that include socioeconomic and
environmental information that may influence breast cancer biology.

522 **Methods**

***Tumor sample collection and clinopathological assessment of in-house molecular*** 524 ***profiled HM patients***

Mexican patients diagnose with primary breast cancer, without a second tumor and treated
with adjuvant therapy at the Institute of Breast Diseases (FUCAM) from 2008-2012 were
convenient collected. Tumor and adjacent non-tumoral tissue, as well as peripheral blood
were obtained (EDTA Vacutainer tubes, BD, 6ml) from each patient after informed consent
was obtained. After macroscopic inspection by the pathologist, sections of tumor and
normal tissue were frozen in liquid nitrogen and store at -80° C until further processing. A
section of the tissue was formalin and embedded in paraffin (FFPE) to confirm pathological
diagnosis as well as to assess tumor cell content and grade by hematoxylin eosin (H&E)
staining. Only samples with tumor content values >60% were further analyzed. Blood
samples were centrifuged to separate and isolate buffy and plasma components.
Additionally, one hundred consecutive FFPE specimens with adjuvant surgical resection
(2012-2016) were convenient collected at Anatomic Pathology Department of FUCAM
fulfilling the inclusion criteria described above. Tumor specimens were evaluated by a
pathologist to determine their histotype and evaluate cellularity, to then macrodissected the
most enriched area with tumoral cells (>60% of tumor cells). The clinical characteristics of
the HM in-house profiled samples are shown in Table 1. The protocol was reviewed and
approved by the Ethics and Research committees of the National Institute of Genomic
Medicine and FUCAM Institute in Mexico City (CE2009/11). All the studies were conducted
in accordance with the Declaration of Helsinki. FUCAM provided services to vulnerable
population covered by Seguro Popular de Salud (Popular Health Insurance), which aimed
to extend health care coverage to the Mexican population, and gratuity interventions.
Socioeconomic position measured by income among patients treated at FUCAM is
reported as low income (\$218 dollars per month). Regarding the educational level of
patients, institutional data showed that 6% are uneducated, 58% with an educational
attainment less than high school, 22% with a high school education and 14% with some
college education (data of the last two years about the general patient population of
FUCAM).

***Immunohistochemistry***

Estrogen and progesterone receptors, HER2 and EGFR expression, was evaluated using
the ER/RP pharmDX (Dako, Denmark, K4071), HercepTest (Dako, Denmark, K5204) and
DAKO EGFR pharmDxTM kit (Dako, Denmark K1492), respectively, following the
manufacturer's instructions. CK5/6 was evaluated with the mouse monoclonal anti-CK5/6
antibody (Dako, Denmark, M7046, D5/16 B4), cytokeratin 14 (Dako, Denmark, E3) and
cytokeratin 17 (Dako, Denmark M7046), as well as, Claudin 1 (Abcam, UK, AB15099) and

[revised manuscript text omitted]

**Ancestry component analysis**

A total of 77 SNP arrays (SNP 6.0, Affymetrix) were used for the ancestry analysis.
Genotypes of 299,411 common SNPs evaluated in 3 genotyping platforms (SNP 6.0 and
500k arrays from Affymetrix and 1M from Illumina) were used to infer the population
structure of Mexican BC patients. Previously genotyped datasets from various sources
were included to evaluate ancestry proportions in the general Mexican Population: A) 95
HapMap samples (CEU: northern European ancestry; YRI,
Africans from Nigeria and CHB+JPT, east Asian population);
B) 37 native Mexican samples from Tepehuano (Durango) group, the Zapoteco
(Oaxaca) group, and Mayas (Campeche) from the Mexican Genome Diversity Project
(MGDP)^{15, 91}; and C) 161 Mexican mestizo sample from the Mexican States of Campeche,
Zacatecas, Sonora, Yucatán, Tamaulipas, Guerrero, Guanajuato, and Veracruz from
MGDP. Samples were subjected to four quality control tests: 1) Missing rate per person:
individuals with more than 5% missing genotypes were excluded, 2) Missing rate per SNP:
only SNPs with a 95% genotyping rate were included, 3) Identity by descent: (IBD) test
were computed to assess quality on the full set of samples 4)
Exclusion of markers that failed the Hardy-
Weinberg Equilibrium test at 0.00001 significance threshold. All samples presented
an optimal quality and no familial relationships were found. Principal components analysis
(PCA) was used to detect population substructures based on genome wide data through
ADMIXTURE v1.22 software. In accordance to the origin of the samples, the optimal K
value was defined as 4, meaning that four parental groups (CEU, YRI, CHB+ JPT and
NAT MEX) were considered to quantify ancestral contribution and to explain the major
substructure in this set of 77 individuals. The ancestral component was identified based on
the frequency of the differences between the relevant parental populations (European,
Asian, African and Natives). The program STRUCTURE v2.3.4 was used to estimate the
individual admixture proportions, as well as the average admixture proportions in each
sample. Ancestry of samples from GEO series were retrieved from deposited sample

[revised manuscript text omitted]

**Availability of data.**

Gene expression and copy number data was deposited in GEO as Super Series under the
accession number GSE87049. MAF files can be download from cBioPortal (Breast
Invasive Carcinoma, Broad, Nature 2012) and dbGAP ([https://](https://https://www.ncbi.nlm.nih.gov/gap/)
<https://www.ncbi.nlm.nih.gov/gap/>) under accession number phs001250. v1.p1 (upon
authorization request).

**Figure legends**

**Figure 1. “Omic” characterization of multi-ethnic breast cancer molecular profiles.**

a) Graphical workflow of breast cancer molecular characterization of in-house profiled HM
samples and in-silico benchmarked data including Hispanic (Peruvian and US Latina
women), African-American, African (Nigerian), Asian and Caucasian breast cancer
patients. Briefly, genomic and transcriptomic data were analyzed to get a deep biological
landscape describing the mutational and copy number alterations, as well as gene
expression profiling of breast cancer. Each molecular platform was then integrated to get a
more robust oncogenic picture of breast tumors and their similarities and differences
between ethnicities. b) Frequency of PAM50 intrinsic molecular subtype in each breast
cancer dataset (N=5418), including in-house profiled Mexican samples (GSE87049 n=109)
and public available data. c) Median frequency of PAM50 intrinsic molecular subtypes
among ethnicities d) Frequency of younger (≤ 45 years of age) and elderly (> 45 years of
age) BC patients among ethnicities. Fisher's exact test p-values with B-H FDR correction
were computed for statistical comparisons *(adj.p.value < 0.05).

**Figure2: Immunohistochemistry classification of breast tumors.** a) Frequency of
immunohistochemistry subtypes routinely evaluated through hormone receptors (HR -
estrogen and progesterone receptor) and HER2 markers. Triple negative (TN) among
ethnic-groups. b) Frequency of BC immunohistochemistry subtypes and patients diagnosis at
early-age (≤ 45 years of age) or elderly-age (> 45 years of age). Fisher's exact test p-values
with B-H FDR correction were computed for statistical comparisons *(adj.p.value < 0.05).

**Figure 3: SCNAs landscape in breast tumors across ethnicities.** a) Boxplot shows the
total number of SCNAs (including likely-significant (± 1) or significant (± 2) events) per
sample among patients from HM ancestry and TCGA multi-ethnic data. SCNAs analysis
was performed on SNP6 Affymetrix arrays data. Statistical comparisons based on
Wilcoxon test taking HM dataset as reference. b) Histogram showing the comparative

tumor SCNA burden (TSCB) distribution between ethnicities. c-d) Histogram showing the
frequency of significant SCNAs at arm level across whole genome in different ethnicities.
Chromosomal regions are arranged on the x-axis. Gains are represented in red, above
horizontal line (c), while losses are represented in blue, under horizontal line d). The
comparison of HM (filled bars) and AA, Asian and Caucasian (lines bars) was calculated
and plotted separately. Arrows indicate unique HM SCNA arm events. Frequency of the
top most significant e) amplifications and f) deletions in HM patients, compared against
their frequency in other ethnicities. Fisher's exact test p-values with B-H FDR correction
were computed for statistical comparisons *(adj.p.value <0.05).

**Figure 4: Tumor mutational burden and cancer driver landscapes of breast cancer**
**tumors.** a) Violin plots showing the TMB (mut/Mb) distribution in each ethnic-group
considering point non-silent mutation. Statistical comparisons were assessed with a
Wilcoxon test considering HM dataset as reference. B) Boxplot of TMB described in the
evaluated data sets dividing patients population in younger (≤ 45 years of age) and older
(> 45 years of age). c) Violin plot showing the total number of known or predicted cancer
driver mutations in each study. d) Barplot showing the fraction of oncogenes or tumor
suppressor genes among the known-driver or predicted driver-mutation in HM tumors. e)
Correlation between the proportion of driver mutations in tumors and their mutational
burden (in logarithmic scale) presented as a scatter plot. Correlation coefficient computed
using the Spearman method. f) Most recurrent driver point mutation in the HM samples.
Heatmap showing the frequency mutation events in highly mutated genes across different
tumor datasets. Color scale, from white to dark-blue, represents the percentage of events,
which is also indicated inside each cell.

**Figure 5: Mutational panorama of HM breast cancer tumors.** a) Oncoplot of
significantly non-silent mutated genes in HM tumors (n=159) computed by MutSig(qvalue
< 0.1). The heatmap represents individual mutations in patient samples, color-coded by
type of mutation as illustrated by the figure legend. Percentages refers to the fraction of
tumors with at least one mutation in the specified gene. b) Lollipop plots of non-silent
mutations detected in AKT1 gene in HM women and their distribution in the body gen. c)
Proportion of tumor samples in HM dataset, separated according to their immunochemical
classification, harboring E17K-AKT1 mutation. d) Fraction of mutational alterations
presented in the PIK3CA/AKT/mTOR axis (left panel, *% of samples co-mutated with AKT)
visualized with PathwayMapper and co-mutated driver alterations in AKT mutated tumors
(MS=Missense). e) Total number of hotspots mutations in HM breast cancer tumors. f)
Total number of mutations in genes annotated as actionable by OncoKb, classified on the
basis of gene level evidence (i.e., 1 FDA drug approved, 2A FDA-approved standard care,
3A Compelling clinical evidence and 4 Compelling preclinical level) in HM evaluated
samples. g) Pie chart represents the frequency of tumors harboring actionable mutations
categorized as oncogenic or likely oncogenic. The barplot on the right shows the
frequency of oncogenic mutations, split according to their impact on protein function, that
are present in the 25% of HM tumor samples. Fisher's exact test p-values with B-H FDR
correction were computed for statistical comparisons *(adj.p.value <0.05)..

**Figure 6: Signatures of mutational processes identified in human breast cancers**
**across ethnicities.** Normalized proportions of the 5 most frequent COSMIC v2
trinucleotide mutational signatures in tumors from HM, AA, Caucasian, Asian TCGA and
Asian Kan harboring a TMB equal or over the median in each data set. Mutational
signatures contributions to each individual sample were depicted by deconstructSigs

algorithms. Donut plot reports the percentage of tumor samples harboring a particular
predominant driving signature described in a) all individuals of each ethnic group or in b)
younger (≤ 45 years of age – Inner donut chart) and elderly patients (> 45 years of age
outer donut chart). Bottom panel described statistical analysis of mutational signatures
prevalence in each of the interrogated data sets. Number of analyzed patients HM
Younger: 16, Older: 62; AA TCGA Younger: 10, Older: 40; Caucasian TCGA Younger: 42,
Older: 267, Asian: TCGA Younger 3, Older 29; Asian Kan Younger: 55, Older: 33. c)
Donut plot reported the percent of tumor samples with co-occurrent mutational signatures
events. d) Sankey diagram showing the co-occurrence of two predominant driving
signature in each sample. Percentage represent the proportion of samples with the co-
occurring event. Fisher's exact test p-values with B-H FDR correction were computed for
statistical comparisons *(adj.p.value < 0.05).

**Figure 7: Integrative view of genomic and transcriptomic alteration in breast tumors**
**carcinogenesis across ethnic-groups.** a) Correlation between tumor mutational burden
(TMB) and tumor altered fraction (TAF) divided in four subclasses in accordance with the
following cutoffs: high TMB > 1 and high TAF $> 5\%$: High-TMB and Low TAF, Low-TMB and
High-TAF, High-TMB and High-TAF, Low-TMB and Low-TAF. b) Prevalence of each
biological group divided by TMB and TAF classes (above mentioned) among ethnic-
groups. c) Cis effects of copy-number alterations on mRNA expression. The heatmap
shows significant correlated genes (Pearson $R=0.5$, adj.p.value < 0.05) between SCNA and
robust gene expression changes (gene expression profiles between altered and non-
altered tumors, logFC:1.5, adj.p.value < 0.05) as illustrated in the sided heat map (over-
expression: red, down-expression: green). The diagonal yellow line represents over-
expressed genes located in amplified regions or down-regulated ones located in deleted
loci. Genes are ordered by chromosome locations on x-axes. The lower heatmap shows
the amplifications (red) and deletions (blue) events in each chromosome. Altered signaling
pathways in breast cancer tumors. Mutual exclusivity modules (MEMo) analysis identified
multiple modules recapitulating ERBB-PI3K, PIK3-Akt-CCND1 signaling (adj.p.value
$= < 0.1$). d) Heatmap showing the distribution of mutated or amplified genes that make up
MEMo modules. e) Graphical representation of the top-scoring mutually exclusive
modules. Nodes represents frequently altered genes in each module and edges connect
them according to their reported activity in corresponding core signaling pathways.
Amplifications are shown in yellow, somatic mutations events in red, black arrows
represents activation and red arrows inhibition. Gain of function biological consequence in
dot lines and loss of function in continuous line.

**Figure 8: Immune landscape of breast cancer tumors.** a) Boxplots showing the overall
distribution of CYT and TIS scores across breast cancer samples and ethnicities b)
Boxplots describing the distribution of CYT and TIS score among PAM50 intrinsic
subtypes in each ethnic-group. Barplot showing the distribution of immune-cell population
signature scores (ssGSEA) of c) adaptive and d) innate immune cells across PAM50
subtypes and ethnicities. Statistical comparison based on Wilcoxon sum of ranking test
taking HM group as reference. * adj.p.value < 0.05 , ** adj.p.value < 0.005

**Acknowledgments**

We want to thank all the patients that made possible this study, Dra. Esperanza
Monterrubio Flores for her support in the immunohistochemical analysis. Paulina Michelle
García Vargas and Rodrigo Bolado Hadad for their support to collect clinical data of HM

dataset. Rosa Gloria Rebollar Vega was supported by a fellowship from CONACYT (CVU:
294330), this study is part of her doctoral thesis from the Biomedical Sciences Doctorate
Program, Faculty of Medicine, Universidad Nacional Autonoma de Mexico. SRC and ISG
were supported by a postdoctoral fellowship from the Mexican Council of Science and
Technology (CONACYT) and SRC has been supported by Cátedra Salvador Zubirán,
UNAM/INCMNSZ.

**Funding**

This work was funded by the Mexican National Council of Science and Technology Basic
Science grant (grant number 258936) and Frontiers in Science grant (number 1285).

**Contributions**

S.L.R-C and A.H-M conceived and designed the study. I.S-G., R.R-G. and S.L.R-C.
contributed equally to this work. R.R-G., S.L.R-C. and L.A-R. collected tumoral specimens
(Fresh-frozen and formalin-fixed paraffin-embedded tumor tissue). I.S-G., R.R-G., S.L.R-C
and A.H-M designed the pipeline-analysis. R.R-G., L.A-R and L.U.F performed nucleic
acid extraction and sample preparation for microarray analysis. I.S-G and S.L.R-C
performed all the genomic analysis. I.S-G and S.L.R-C wrote the paper. R.R-G., A.H-M.
and L.U.F drafted, edit, discuss and finalized the paper. I.S-G., S.L.R-C and J.C.F-L.
provided bioinformatics support. V.B-P., F.V.C., A.T-T., C.D-R. procured patient tumoral
specimens, assisted the pathological assessment and immunochemistry evaluation and
provided clinical features of analyzed samples. R.R-G performed immunohistochemical
evaluation for triple-negative tumors (extended markers). R.R-G and L.A-R managed the
clinical data bases. J.C.F-L performed ancestry analysis of HM tumors. All authors
reviewed and approved the final manuscript.

**Competing interests**

964 A.H-M. received a grant from Astra Zeneca Mexico, outside the submitted work. All
965 remaining authors declare no competing interest.

**Corresponding authors**

Correspondence to Sandra L. Romero-Cordoba or Alfredo Hidalgo-Miranda.

Table 1:
statistics of
features of
HM patients

Clinical feature	N=	%
Age (range) (n=204)		
29-40	19	9.3
41-50	56	27.5
51-60	58	28.4
61-70	44	22.1
71-80	11	4.9
81-90	3	1.5
NA	13	6.4
Younger (<45 years of age)	47	24.5
Older (>45 years of age)	144	75.5
Histologic type (n=178)		
Ductal	146	82
Lobular	16	9
Mixed ductal/lobular	7	3.9
Others	9	5.1
Tumor grade (N=158)		
I	12	7.6
II	121	76.58
III	25	15.82
Tumor size (N=182)		
1-15mm	35	19.2
16-30mm	113	62.08
>30mm	45	24.73
IHC (n=189)		

Summary
clinical
evaluated

ER+	142	75.1
ER-	47	24.9
PR+	121	64
PR-	66	34.9
PR+ No info	2	1.1
Her2+	31	16.4
Her2-	155	82
Her2 No info	3	1.6
Triple negative tumors dedicated IHC (N=11)		
EGFR+ Cit5/6+ Cit14+ Cit17+ CLD1+ CLD3+	3	27
EGFR+ Cit5/6- Cit14+ Cit17+ CLD1+ CLD3+	1	9
EGFR- Cit5/6- Cit14+ Cit17+ CLD1+ CLD3+	4	37
EGFR- Cit5/6- Cit14- Cit17- CLD1+ CLD3+	3	27

ER positive required at least 1% staining nuclei.
 PR positive required at least 1% staining nuclei.
 HER2 positive were 3+ IHC or 2+ and CISH confirmed.

References

1. Anderson BO. Breast cancer--thinking globally. *Science* **343**, 1403 (2014).

2. Knaul FM, Nigenda G, Lozano R, Arreola-Ornelas H, Langer A, Frenk J. [Breast cancer in
Mexico: an urgent priority]. *Salud publica de Mexico* **51 Suppl 2**, s335-344 (2009).
3. Perou CM, *et al.* Molecular portraits of human breast tumours. *Nature* **406**, 747-752
(2000).
4. Sorlie T, *et al.* Gene expression patterns of breast carcinomas distinguish tumor
subclasses with clinical implications. *Proceedings of the National Academy of Sciences*
*of the United States of America* **98**, 10869-10874 (2001).
5. van 't Veer LJ, *et al.* Gene expression profiling predicts clinical outcome of breast
cancer. *Nature* **415**, 530-536 (2002).
6. Polyak K. Heterogeneity in breast cancer. *The Journal of clinical investigation* **121**,
3786-3788 (2011).
7. Zaha DC. Significance of immunohistochemistry in breast cancer. *World journal of*
*clinical oncology* **5**, 382-392 (2014).
8. Rouzier R, *et al.* Breast cancer molecular subtypes respond differently to preoperative
chemotherapy. *Clinical cancer research : an official journal of the American Association*
*for Cancer Research* **11**, 5678-5685 (2005).
9. Cancer Genome Atlas N. Comprehensive molecular portraits of human breast tumours.
*Nature* **490**, 61-70 (2012).
10. Stephens PJ, *et al.* Complex landscapes of somatic rearrangement in human breast
cancer genomes. *Nature* **462**, 1005-1010 (2009).
11. Zack TI, *et al.* Pan-cancer patterns of somatic copy number alteration. *Nature genetics*
**45**, 1134-1140 (2013).
12. Banerji S, *et al.* Sequence analysis of mutations and translocations across breast cancer
subtypes. *Nature* **486**, 405-409 (2012).
13. Bergamaschi A, *et al.* Distinct patterns of DNA copy number alteration are associated
with different clinicopathological features and gene-expression subtypes of breast
cancer. *Genes, chromosomes & cancer* **45**, 1033-1040 (2006).
14. Bureau USC. Hispanic Heritage Month 2018. (ed^(eds) (2018).
15. Silva-Zolezzi I, *et al.* Analysis of genomic diversity in Mexican Mestizo populations to
develop genomic medicine in Mexico. *Proc Natl Acad Sci U S A* **106**, 8611-8616 (2009).
16. Kwan ML, *et al.* Epidemiology of breast cancer subtypes in two prospective cohort
studies of breast cancer survivors. *Breast cancer research : BCR* **11**, R31 (2009).
17. Popejoy AB, Fullerton SM. Genomics is failing on diversity. *Nature* **538**, 161-164
(2016).

18. Spratt DE, *et al.* Racial/Ethnic Disparities in Genomic Sequencing. *JAMA Oncol* **2**, 1070-
1074 (2016).
19. Parker JS, *et al.* Supervised risk predictor of breast cancer based on intrinsic subtypes.
*J Clin Oncol* **27**, 1160-1167 (2009).
20. Goldhirsch A, *et al.* Personalizing the treatment of women with early breast cancer:
highlights of the St Gallen International Expert Consensus on the Primary Therapy of
Early Breast Cancer 2013. *Ann Oncol* **24**, 2206-2223 (2013).
21. Bauer KR, Brown M, Cress RD, Parise CA, Caggiano V. Descriptive analysis of estrogen
receptor (ER)-negative, progesterone receptor (PR)-negative, and HER2-negative
invasive breast cancer, the so-called triple-negative phenotype: a population-based
study from the California cancer Registry. *Cancer* **109**, 1721-1728 (2007).
22. Dietze EC, Sistrunk C, Miranda-Carboni G, O'Regan R, Seewaldt VL. Triple-negative
breast cancer in African-American women: disparities versus biology. *Nat Rev Cancer*
**15**, 248-254 (2015).
23. Stratton MR, Campbell PJ, Futreal PA. The cancer genome. *Nature* **458**, 719-724
(2009).
24. Godinho MF, *et al.* BCAR4 induces antioestrogen resistance but sensitises breast
cancer to lapatinib. *Br J Cancer* **107**, 947-955 (2012).
25. Breast cancer anti-estrogen resistance 4 (BCAR4). *Science-Business eXchange* **7**, 1370-
1370 (2014).
26. Hatakeyama K, *et al.* Mutational burden and signatures in 4000 Japanese cancers
provide insights into tumorigenesis and response to therapy. *Cancer Sci* **110**, 2620-
2628 (2019).
27. Consortium ITP-CAoWG. Pan-cancer analysis of whole genomes. *Nature* **578**, 82-93
(2020).
28. Lawrence MS, *et al.* Mutational heterogeneity in cancer and the search for new cancer-
associated genes. *Nature* **499**, 214-218 (2013).
29. Kan Z, *et al.* Multi-omics profiling of younger Asian breast cancers reveals distinctive
molecular signatures. *Nat Commun* **9**, 1725 (2018).
30. Lee ES. Breast Cancer - Very young women (BRCA-KR). (ed[^](eds) (2015).
31. Chang MT, *et al.* Identifying recurrent mutations in cancer reveals widespread lineage
diversity and mutational specificity. *Nat Biotechnol* **34**, 155-163 (2016).
32. Chakravarty D, *et al.* OncoKB: A Precision Oncology Knowledge Base. *JCO Precis Oncol*
**2017**, (2017).

33. Nik-Zainal S, *et al.* Mutational processes molding the genomes of 21 breast cancers. *Cell* **149**, 979-993 (2012).
34. Alexandrov LB, Nik-Zainal S, Wedge DC, Campbell PJ, Stratton MR. Deciphering
signatures of mutational processes operative in human cancer. *Cell Rep* **3**, 246-259
(2013).
35. Gori KB-O, Adrian. sigfit: flexible Bayesian inference of mutational signatures. (2018).
36. Forbes SA, *et al.* COSMIC: mining complete cancer genomes in the Catalogue of Somatic
Mutations in Cancer. *Nucleic Acids Res* **39**, D945-950 (2011).
37. Roberts SA, *et al.* An APOBEC cytidine deaminase mutagenesis pattern is widespread
in human cancers. *Nat Genet* **45**, 970-976 (2013).
38. Hanahan D, Weinberg RA. Hallmarks of cancer: the next generation. *Cell* **144**, 646-674
(2011).
39. Ciriello G, Cerami E, Sander C, Schultz N. Mutual exclusivity analysis identifies
oncogenic network modules. *Genome Res* **22**, 398-406 (2012).
40. Ruiz-Saenz A, Dreyer C, Campbell MR, Steri V, Gulizia N, Moasser MM. HER2
Amplification in Tumors Activates PI3K/Akt Signaling Independent of HER3. *Cancer*
*Res* **78**, 3645-3658 (2018).
41. Inoue K, Fry EA. Aberrant expression of cyclin D1 in cancer. *Sign Transduct Insights* **4**,
1-13 (2015).
42. Sherr CJ, McCormick F. The RB and p53 pathways in cancer. *Cancer Cell* **2**, 103-112
(2002).
43. Rooney MS, Shukla SA, Wu CJ, Getz G, Hacohen N. Molecular and genetic properties of
tumors associated with local immune cytolytic activity. *Cell* **160**, 48-61 (2015).
44. Ayers M, *et al.* IFN-gamma-related mRNA profile predicts clinical response to PD-1
blockade. *J Clin Invest* **127**, 2930-2940 (2017).
45. Danaher P, *et al.* Pan-cancer adaptive immune resistance as defined by the Tumor
Inflammation Signature (TIS): results from The Cancer Genome Atlas (TCGA). *J*
*Immunother Cancer* **6**, 63 (2018).
46. Charoentong P, *et al.* Pan-cancer Immunogenomic Analyses Reveal Genotype-
Immunophenotype Relationships and Predictors of Response to Checkpoint Blockade.
*Cell Rep* **18**, 248-262 (2017).
47. Guerrero S, *et al.* Analysis of Racial/Ethnic Representation in Select Basic and Applied
Cancer Research Studies. *Sci Rep* **8**, 13978 (2018).
48. Villarreal-Garza C, *et al.* Molecular Subtypes and Prognosis in Young Mexican Women
With Breast Cancer. *Clin Breast Cancer* **17**, e95-e102 (2017).

49. Villarreal-Garza C, Lopez-Martinez EA, Munoz-Lozano JF, Unger-Saldana K. Locally
advanced breast cancer in young women in Latin America. *Ecancermedicalscience* **13**,
894 (2019).
50. Tfayli A, Temraz S, Abou Mrad R, Shamseddine A. Breast cancer in low- and middle-
income countries: an emerging and challenging epidemic. *J Oncol* **2010**, 490631
(2010).
51. Villarreal-Garza C, *et al.* Breast cancer in young women in Latin America: an unmet,
growing burden. *Oncologist* **18**, 1298-1306 (2013).
52. Society AC. Breast cancer facts & figures 2011- 2012. (ed^(eds). Atlanta, Georgia:
American Cancer Society (2013).
53. Reynoso-Noveron N, *et al.* Clinical and Epidemiological Profile of Breast Cancer in
Mexico: Results of the Seguro Popular. *J Glob Oncol* **3**, 757-764 (2017).
54. Scott LC, Mobley LR, Kuo TM, Il'iasova D. Update on triple-negative breast cancer
disparities for the United States: A population-based study from the United States
Cancer Statistics database, 2010 through 2014. *Cancer* **125**, 3412-3417 (2019).
55. Ciriello G, Miller ML, Aksoy BA, Senbabaoglu Y, Schultz N, Sander C. Emerging
landscape of oncogenic signatures across human cancers. *Nat Genet* **45**, 1127-1133
(2013).
56. Lopez-Cortes A, *et al.* Mutational Analysis of Oncogenic AKT1 Gene Associated with
Breast Cancer Risk in the High Altitude Ecuadorian Mestizo Population. *Biomed Res Int*
**2018**, 7463832 (2018).
57. Li G, *et al.* Prevalence and spectrum of AKT1, PIK3CA, PTEN and TP53 somatic
mutations in Chinese breast cancer patients. *PLoS One* **13**, e0203495 (2018).
58. Rudolph M, *et al.* AKT1 (E17K) mutation profiling in breast cancer: prevalence,
concurrent oncogenic alterations, and blood-based detection. *BMC Cancer* **16**, 622
(2016).
59. Fruman DA, Rommel C. PI3K and cancer: lessons, challenges and opportunities. *Nat*
*Rev Drug Discov* **13**, 140-156 (2014).
60. Janku F, Yap TA, Meric-Bernstam F. Targeting the PI3K pathway in cancer: are we
making headway? *Nat Rev Clin Oncol* **15**, 273-291 (2018).
61. Toker A. Achieving specificity in Akt signaling in cancer. *Adv Biol Regul* **52**, 78-87
(2012).
62. Vandermoere F, *et al.* Proteomics exploration reveals that actin is a signaling target of
the kinase Akt. *Mol Cell Proteomics* **6**, 114-124 (2007).

63. Polo ML, Arnoni MV, Riggio M, Wargon V, Lanari C, Novaro V. Responsiveness to PI3K
and MEK inhibitors in breast cancer. Use of a 3D culture system to study pathways
related to hormone independence in mice. *PLoS One* **5**, e10786 (2010).
64. Riggio M, *et al.* PI3K/AKT pathway regulates phosphorylation of steroid receptors,
hormone independence and tumor differentiation in breast cancer. *Carcinogenesis* **33**,
509-518 (2012).
65. Thorpe LM, Yuzugullu H, Zhao JJ. PI3K in cancer: divergent roles of isoforms, modes of
activation and therapeutic targeting. *Nat Rev Cancer* **15**, 7-24 (2015).
66. Hyman DM, *et al.* AKT Inhibition in Solid Tumors With AKT1 Mutations. *J Clin Oncol*
**35**, 2251-2259 (2017).
67. Janku F, *et al.* PIK3CA mutations in patients with advanced cancers treated with
PI3K/AKT/mTOR axis inhibitors. *Mol Cancer Ther* **10**, 558-565 (2011).
68. Carpten JD, *et al.* A transforming mutation in the pleckstrin homology domain of AKT1
in cancer. *Nature* **448**, 439-444 (2007).
69. Capivasertib Active against AKT1-Mutated Cancers. *Cancer Discov* **9**, OF7 (2019).
70. Park JW, *et al.* Adaptive Randomization of Neratinib in Early Breast Cancer. *N Engl J*
*Med* **375**, 11-22 (2016).
71. Rugo HS, *et al.* Adaptive Randomization of Veliparib-Carboplatin Treatment in Breast
Cancer. *N Engl J Med* **375**, 23-34 (2016).
72. Kim SB, *et al.* Ipatasertib plus paclitaxel versus placebo plus paclitaxel as first-line
therapy for metastatic triple-negative breast cancer (LOTUS): a multicentre,
randomised, double-blind, placebo-controlled, phase 2 trial. *Lancet Oncol* **18**, 1360-
1372 (2017).
73. Alexandrov LB, *et al.* Clock-like mutational processes in human somatic cells. *Nat*
*Genet* **47**, 1402-1407 (2015).
74. Alexandrov LB, *et al.* Signatures of mutational processes in human cancer. *Nature* **500**,
415-421 (2013).
75. Kresovich JK, Xu Z, O'Brien KM, Weinberg CR, Sandler DP, Taylor JA. Methylation-
based biological age and breast cancer risk. *J Natl Cancer Inst*, (2019).
76. Issa JP. Aging and epigenetic drift: a vicious cycle. *J Clin Invest* **124**, 24-29 (2014).
77. Belsky DW, *et al.* Quantification of biological aging in young adults. *Proc Natl Acad Sci*
*U S A* **112**, E4104-4110 (2015).
78. Adaniel C, Jhaveri K, Heguy A, Esteva FJ. Genome-based risk prediction for early stage
breast cancer. *The oncologist* **19**, 1019-1027 (2014).

- 79. Carvalho BS, Irizarry RA. A framework for oligonucleotide microarray preprocessing.
*Bioinformatics* **26**, 2363-2367 (2010).
- 80. Leek JT JW, Parker HS, Fertig EJ, Jaffe AE, Storey JD, Zhang Y, Torres LC sva: Surrogate
Variable Analysis. R package version 3.32.1. (ed^(eds) (2019).
- 81. Ritchie ME, *et al.* limma powers differential expression analyses for RNA-sequencing
and microarray studies. *Nucleic Acids Res* **43**, e47 (2015).
- 82. Kroenke CH, *et al.* Race and breast cancer survival by intrinsic subtype based on
PAM50 gene expression. *Breast Cancer Res Treat* **144**, 689-699 (2014).
- 83. Pereira B, *et al.* The somatic mutation profiles of 2,433 breast cancers refines their
genomic and transcriptomic landscapes. *Nat Commun* **7**, 11479 (2016).
- 84. Pitt JJ, *et al.* Characterization of Nigerian breast cancer reveals prevalent homologous
recombination deficiency and aggressive molecular features. *Nat Commun* **9**, 4181
(2018).
- 85. Barrett T, *et al.* NCBI GEO: archive for functional genomics data sets--update. *Nucleic*
*Acids Res* **41**, D991-995 (2013).
- 86. Durinck S, Spellman PT, Birney E, Huber W. Mapping identifiers for the integration of
genomic datasets with the R/Bioconductor package biomaRt. *Nat Protoc* **4**, 1184-1191
(2009).
- 87. Johnson WE, Li C, Rabinovic A. Adjusting batch effects in microarray expression data
using empirical Bayes methods. *Biostatistics* **8**, 118-127 (2007).
- 88. Yoshihara K, *et al.* Inferring tumour purity and stromal and immune cell admixture
from expression data. *Nat Commun* **4**, 2612 (2013).
- 89. Fresno C GG, Llera AS, Fernandez EA pbcmc: Permutation-Based Confidence for
Molecular Classification. R package version 1.9.0. (ed^(eds) (2018).
- 90. Prat A, *et al.* Phenotypic and molecular characterization of the claudin-low intrinsic
subtype of breast cancer. *Breast Cancer Res* **12**, R68 (2010).
- 91. Moreno-Estrada A, *et al.* Human genetics. The genetics of Mexico recapitulates Native
American substructure and affects biomedical traits. *Science* **344**, 1280-1285 (2014).
- 92. Curtis C, *et al.* The genomic and transcriptomic architecture of 2,000 breast tumours
reveals novel subgroups. *Nature* **486**, 346-352 (2012).
- 93. Reich M, Liefeld T, Gould J, Lerner J, Tamayo P, Mesirov JP. GenePattern 2.0. *Nat Genet*
**38**, 500-501 (2006).
- 94. Mermel CH, Schumacher SE, Hill B, Meyerson ML, Beroukheim R, Getz G. GISTIC2.0
facilitates sensitive and confident localization of the targets of focal somatic copy-
number alteration in human cancers. *Genome Biol* **12**, R41 (2011).

95. Gnirke A, *et al.* Solution hybrid selection with ultra-long oligonucleotides for massively
parallel targeted sequencing. *Nat Biotechnol* **27**, 182-189 (2009).
96. Li H, Durbin R. Fast and accurate short read alignment with Burrows-Wheeler
transform. *Bioinformatics* **25**, 1754-1760 (2009).
97. Rheinbay E, *et al.* Recurrent and functional regulatory mutations in breast cancer.
*Nature* **547**, 55-60 (2017).
98. Gao J, *et al.* Integrative analysis of complex cancer genomics and clinical profiles using
the cBioPortal. *Sci Signal* **6**, pl1 (2013).
99. Cerami E, *et al.* The cBio cancer genomics portal: an open platform for exploring
multidimensional cancer genomics data. *Cancer Discov* **2**, 401-404 (2012).
100. Cibulskis K, *et al.* Sensitive detection of somatic point mutations in impure and
heterogeneous cancer samples. *Nat Biotechnol* **31**, 213-219 (2013).
101. Ramos AH, *et al.* Oncotator: cancer variant annotation tool. *Hum Mutat* **36**, E2423-
2429 (2015).
102. Zhou W, *et al.* TransVar: a multilevel variant annotator for precision genomics. *Nat*
*Methods* **12**, 1002-1003 (2015).
103. Mayakonda A, Lin DC, Assenov Y, Plass C, Koeffler HP. Maftools: efficient and
comprehensive analysis of somatic variants in cancer. *Genome Res* **28**, 1747-1756
(2018).
104. Chang MT, *et al.* Accelerating Discovery of Functional Mutant Alleles in Cancer. *Cancer*
*Discov* **8**, 174-183 (2018).
105. Gonzalez-Perez A, Lopez-Bigas N. Functional impact bias reveals cancer drivers.
*Nucleic Acids Res* **40**, e169 (2012).
106. Berman HM, *et al.* The Protein Data Bank. *Nucleic Acids Res* **28**, 235-242 (2000).
107. Rose AS, Bradley AR, Valasatava Y, Duarte JM, Prlic A, Rose PW. NGL viewer: web-
based molecular graphics for large complexes. *Bioinformatics* **34**, 3755-3758 (2018).
108. Rosenthal R, McGranahan N, Herrero J, Taylor BS, Swanton C. DeconstructSigs:
delineating mutational processes in single tumors distinguishes DNA repair
deficiencies and patterns of carcinoma evolution. *Genome Biol* **17**, 31 (2016).
109. Plantalech O. Fraction of Genome Altered and Total Mutations added to cBioPortal
Plots tab. In: *thehyve* (ed[^](eds) (2018).
110. Yaeger R, *et al.* Clinical Sequencing Defines the Genomic Landscape of Metastatic
Colorectal Cancer. *Cancer Cell* **33**, 125-136 e123 (2018).

- 111. Chen EY, *et al.* Enrichr: interactive and collaborative HTML5 gene list enrichment
analysis tool. *BMC Bioinformatics* **14**, 128 (2013).
- 112. Kuleshov MV, *et al.* Enrichr: a comprehensive gene set enrichment analysis web server
2016 update. *Nucleic Acids Res* **44**, W90-97 (2016).
- 113. Hanzelmann S, Castelo R, Guinney J. GSEA: gene set variation analysis for microarray
and RNA-seq data. *BMC Bioinformatics* **14**, 7 (2013).

a

FIG 1

Omic landscape and clinical data

Normalization, processing, significance
Batch effect correction
Significance

Mutational landscape

Significant SCNA

Sample classification and GEP

Tumor immune-contexture

b

N	AA	Asian	Caucasian	Hispanic
---	----	-------	-----------	----------

c

d

a

Her2+ HR+/Her2- HR+/Her2+ TN

**b**

f

40	26	10	8	7	7	5	5	5	4	4	4	4	4	HM
33	43*	0*	1	0*	3	4	0	3	1	1	0	3	0	AA
41	61*	0*	7	0*	4	7	0	15*	0	15	0	0	0	Asian-TCGA
39	52*	3	3	0*	2	1	0	0	3	2	0	1	1	Asian-SMC
54	28*	5	8	0*	4	4	0	0	0	4	0	0	0	Caucasian
PIK3CA	TP53	AKT1	MAP3K1	KMT2C	PTEN	GATA3	KMT2D	MED12	ATM	ERBB2	KMT2A	MGA	SOS1	

a

b

◆ <0.05 ◆ <0.005 ■ <0.0005 ■ Non significant

c

d

a**b**
Legend for ethnic groups: HM (green), Hispanic (blue), AA (orange), Asian (purple), Caucasian (pink).

c**d**
REVIEWER COMMENTS

Reviewer #1 (Remarks to the Author): Expert in breast cancer genomics and epidemiology

The manuscript presents data on molecular features of breast cancers in a sample of 204 Mexican-Hispanic (HM) patients. They first compare these data with publicly available datasets and state that HM presented an earlier onset of the disease, in particular in the aggressive Basal-like/triple negative and Luminal B subtypes. These two statements do not seem to be supported by the data proposed and its not clear what comparison they are making to make these claims. In table 1 you see that in their data they have about 6% TN which are usually enriched for basal breast cancers and 24% of cases are diagnosed under 45 years of age--which does not really support the claim of an earlier onset or perhaps they are comparing it to something that was not clearly indicated. The presentation is still muddled and a list of multiple different bioinformatic analysis without a clear hypothesis and it seems like a fishing expedition. They conclude that this study provides a conceptual framework to better understand ethnic disparities of breast cancer--but I do not think this is shown in their paper.

Other comments. Data presentation of tables should include the total possible participants and missing category for data not available not multiple different N's as this is confusing.

Reviewer #2 (Remarks to the Author): Expert in cancer ancestry

The authors have comprehensively answered all previously raised concerns. Several claims from the initial submission have been removed as they did not pass rigorous testing. The writing and rigor in describing the methods and results has been greatly fixed, and thus I find the revision as a significant improvement. Although results are not that exciting and data set is too small to be representative of the entire Mexican-Hispanic women, there is a strong need for diverse genomic data and as such I support the publication of this report.

Comments:

1. The text still overstates in various places the extendability of this data to the entirety of Mexican-Hispanic women. The authors could tone this claim down in text by strictly referring to the set of n=204 tumors presented here.
2. The authors claim that PCA was ran using ADMIXTURE software whereas STRUCTURE was used to estimate individual ancestry proportions. I suspect a typo/error. ADMIXTURE is a software that estimates individual level ancestry proportions (not PCA); ADMIXTURE and STRUCTURE solve exactly the same problem.
3. Several methodological details to replicate the analysis are still missing; e.g., "Enrichment analysis of GO biological process terms among significantly correlated genes was performed using the package Enrichr v2.1111, 112 implemented in Bioconductor/R." What parameters were provided to the software?
4. How many tests where controlled for in the Benjamini&Hochberg procedure? Are p-values reported throughout the main text adjusted or raw?

REVIEWER COMMENTS

Reviewer #1 (Remarks to the Author): Expert in breast cancer genomics and epidemiology

The manuscript presents data on molecular features of breast cancers in a sample of 204 Mexican-Hispanic (HM) patients. They first compare these data with publicly available datasets and state that HM presented an earlier onset of the disease, in particular in the aggressive Basal-like/triple negative and Luminal B subtypes. These two statements do not seem to be supported by the data proposed and its not clear what comparison they are making to make these claims. In table 1 you see that in their data they have about 6% TN which are usually enriched for basal breast cancers and 24% of cases are diagnosed under 45 years of age--which does not really support the claim of an earlier onset or perhaps they are comparing it to something that was not clearly indicated.

The presentation is still muddled and a list of multiple different bioinformatic analysis without a clear hypothesis and it seems like a fishing expedition. They conclude that this study provides a conceptual framework to better understand ethnic disparities of breast cancer--but I do not think this is shown in their paper.

Other comments. Data presentation of tables should include the total possible participants and missing category for data not available not multiple different N's as this is confusing.

Reviewer #2 (Remarks to the Author): Expert in cancer ancestry

The authors have comprehensively answered all previously raised concerns. Several claims from the initial submission have been removed as they did not pass rigorous testing. The writing and rigor in describing the methods and results has been greatly fixed, and thus I find the revision as a significant improvement. Although results are not that exciting and data set is too small to be representative of the entire Mexican-Hispanic women, there is a strong need for diverse genomic data and as such I support the publication of this report.

Comments:

1. The text still overstates in various places the extendability of this data to the entirety of Mexican-Hispanic women. The authors could tone this claim down in text by strictly referring to the set of n=204 tumors presented here.
2. The authors claim that PCA was ran using ADMIXTURE software whereas STRUCTURE was used to estimate individual ancestry proportions. I suspect a typo/error. ADMIXTURE is a software that estimates individual level ancestry proportions (not PCA); ADMIXTURE and STRUCTURE solve exactly the same problem.
3. Several methodological details to replicate the analysis are still missing; e.g., "Enrichment analysis of GO biological process terms among significantly correlated genes was performed using the package Enrichr v2.1111, 112 implemented in Bioconductor/R." What parameters were provided to the software?
4. How many tests where controlled for in the Benjamini&Hochberg procedure? Are p-values reported throughout the main text adjusted or raw?

Response to reviewers' comments

Reviewer #1 (Remarks to the Author):

First of all, we would like to thank again the reviewer for considering our manuscript and for his/her comments about it. Below, we submit our reply.

They first compare these data with publicly available datasets and state that HM presented an earlier onset of the disease, in particular in the aggressive Basal-like/triple negative and Luminal B subtypes. These two statements do not seem to be supported by the data proposed and its not clear what comparison they are making to make these claims. In table 1 you see that in their data they have about 6% TN which are usually enriched for basal breast cancers and 24% of cases are diagnosed under 45 years of age--which does not really support the claim of an earlier onset or perhaps they are comparing it to something that was not clearly indicated.

We appreciate your comments. This was probably a confusion since in supplementary table 1 we had reported 21 TN, which account for 11% (21/186) of the total tumors analyzed, out of which 7 correspond to patients younger than 45 years representing a 33.33% (7/21) of the total TN cases reported. While through PAM50 Basal subtype we had identified a frequency of TN of 12% and 31% (4/13) of younger women (<45 years old).

To overcome the size limitation, commented previously by the reviewers, we integrated our HM sample set along with the samples included in GSE75678 data set that examines the gene expression pattern of 51 Mexican patients with breast cancer from Monterrey, Mexico. We have now clearly stated this data integration in the text. The resulting proportions are as follows: 14% of TN (32/237) from which 34% (11/32) are younger patients (Supplementary Table 2). Additionally, the basal subtype accounted for 15% (24/162) with 33% of younger women (8/24) (Supplementary Table 2).

Regarding HR+/HER2+ (Luminal B by immunohistochemistry) subtype, we observed 36% of younger patients, a higher proportion than those reported in the evaluated women from non-Hispanic ancestry.

We had now accompanied this result section with some relevant epidemiological data from the literature which point out this early on-set in Mexican patients. As well, we reported an independent validation in another of our Mexican sample collection conformed by 94 TN tumors from which 37% (35/94) presented an early onset.

We currently depict these data in a dedicated main figure (Figure 2) and supplementary figure 3.

Figure 2

Figure 2: Age distribution in BC samples from diverse human populations and within IHC and molecular intrinsic subtypes. a) Frequency of younger (≤ 45 years of age) and elderly (> 45 years of age) BC patients among ethnicities. Frequency of BC b) immunochemistry subtypes and c) PAM50 intrinsic molecular subtypes in patients from different ancestry diagnosis at early-age (≤ 45 years of age) or elderly-age (> 45 years of age). Barplots represent proportion of age classes in each population group, while heatmaps represent the BH adjusted p values computed by a Fisher's exact test from multiple comparisons. HM: our profiled tumors integrated with GSE75678 Mexican tumors. Hispanic non-Mexican (H nM): Hispanic tumors from GSE16716, GSE78958, and GSE20271. **adj.p.value <0.05, *adj.p.value <0.1.

The presentation is still muddled and a list of multiple different bioinformatic analysis without a clear hypothesis and it seems like a fishing expedition.

The basis and hypothesis of this study born from different evidences. On one hand, Mexico is a diverse country in terms of culture and ethnicity. Indeed, some relevant studies described how the genomic patterns of the Mexican population can be important for medical traits. As shown by Moreno-Estrada and collaborators¹ ancestral genomic background of Mexico is especially important in pulmonary medicine. That is, depending on ethnic background, the clinical measurements of lung function (value of forced expiratory volume in 1s) can be considered on normal range or as an indicative of pulmonary disease. These results suggest that ancestry could have significant impacts in medical data interpretation in Mexican patients. In other large genomic study of Mexicans and other Latin Americans including type 2 diabetes patients and non-diabetic individuals, the authors identified a novel locus associated with type 2 diabetes risk, a polymorphism in SLC16A11 gene, which is present in about 50% of Native American samples but is rare in other ancestries such as Europeans and Africans².

Moreover, understanding particular molecular patterns in patient from diverse human population is important for the interpretation of medical data. But Mexican healthy and non-healthy individuals, remain largely uncharacterized and, relevantly, Mexico represents an attractive prime focus for genomic analyses, because it presents one of the largest sources of native-Amerindian diversity and, thorough the historic and migratory events, has nowadays different population contributions. On the other hand it has been reported that BC disease burden is greater in developing countries than in developed countries mirrored by a higher mortality rates, early catastrophic events, younger age of women at diagnosis (occurring 10 years younger) and higher prevalence of triple-negative disease³. The reason for this clinical behavior is still unclear, but the combination of demographic, hereditary, environmental, lifestyle risk factors and genomic alterations operating in tumor cells clearly contribute to cancer aetiology and progression.

While recent exiting findings based on cancer genome (at DNA, RNA, proteomic and epigenomic level) have emerged, the genomic contribution of the ethnicity in breast cancer has been more difficult to estimate. All this stress the need to include non-well represented populations such as Mexican Hispanic women in deep genomic characterizations evaluating different molecular alterations. Thus, in this study we questioned how these reported clinical and epidemiological characteristics in BC Mexican patients can be related with unique genomics features?

With the purpose of making our text more readable and to better reflects the objectives and hypothesis we had re-organized our text and rephrase some sections to improve our narrative. In these regards, we would like to be more specific at emphasizing the main conceptual framework that guides our approaches along the report. At the same time, we carefully tried to connect our first results with down-stream analysis and findings.

First, we considered the importance of understanding the frequency of BC subtypes within HM samples and contextualize these particular intrinsic and immunochemistry subtypes within the background of demographic features, mainly age at diagnosis, in our data set and how they compare to frequencies observed in groups of other non-Mexican ancestry component. In accordance with what is reported in the literature, we observed an enrichment of women with early-onset (<45 years) in HM collection (integrating our tumor collection and data from GSE75678), mainly in TN/Basal and HR+ HER+ subtypes.

We then sought to describe the contribution of the somatic mutation profiles and their relation with age at diagnosis through 3 approaches: (1) First, by studying global tumor mutational burden (TMB) and their differences between the evaluated groups, which are still unknown mainly for Mexican patients. HM women analyzed had a higher TMB than Caucasian and AA patients included in the TCGA data set. (2) Next, we questioned if there are differences not only at the total number of non-synonymous somatic mutations but also at the global mutational processes, thus, we explored the prevalence of mutational signatures within the HM group and non-Hispanic patients included in this study. A deep characterization of molecular and demographic characteristics revealed an enrichment of the clock-like signature 1 in the HM evaluated women. Further, Mexican tumors presenting APOBEC-related signatures (Signature 2/13) also presented the highest TMB values across the evaluated samples and were more highly enriched in HR+/HER2- tumors in comparison to non-Hispanic samples. (3) Based on the above results and particular disparities observed, we hypothesized that there are known somatic mutations in BC that are present in HM but at a different frequency. For this purpose, we explored the landscape of somatic mutation in samples from which we had whole exome sequencing. We in fact detected an enriched frequency (8%) in AKT1 E17K mutation in the analyzed HM samples.

As part of the other molecular alterations that can account for the pathological and demographic disparities observed within HM samples are the somatic copy number alterations, so we decided to test the hypothesis that there are some particular SCNA with a different frequency in HM with respect to other ancestry groups. We in fact detected some novel copy number events and variability in the frequency of well-known copy number alterations. Then, we interrogated how mutations and SCNV processes operate in coordination or in an exclusive manner in the evaluated tumors.

Finally, by integrating our previous results describing that HM women are more likely to present higher TMB values and more hypermutated tumors than some other non-Hispanic women and, that the analyzed Mexican HR+/HER2- (Luminal A) malignancies more frequently harbor the signature 2/13 and also presented a higher TMB; we hypothesized that other tumoral components associated with these features, such as the immune infiltrating landscape, might also differ. It has been reported that APOBEC-related signatures significantly contribute to neoantigen loads and the infiltration of certain immune cells in different cancer types⁴. Further, recent evidence has shown that tumors with higher TMB can be more responsive to immune checkpoint inhibitor therapies, which may be due to their increased immunogenicity. We finally detected, a more immunogenic phenotype in luminal A tumors mainly composed by HR+/HER2- malignancies, as shown by a higher cytolytic and tumor inflammation scores, as well as immune score computed by Estimate algorithm.

They conclude that this study provides a conceptual framework to better understand ethnic disparities of breast cancer--but I do not think this is shown in their paper.

We appreciate the reviewer's comment. To attempt to address his/her concern, we have reappraised the intention of our statements and conclusions. In doing so, we have thoroughly reviewed the manuscript in order to tone-down claims that might seem far-reaching and out of the context of our results in HM tumors. Although, we recognize the limitations of our study in these sense, we still consider that a relevant contribution of this study is the multi-omic characterization of a set of HM samples, albeit not being representative of the entire diversity of Hispanic or Mexican population, belongs to a group that remains under-represented among the current body of studies addressing the biological factors affecting BC cancer.

In an effort to reconcile these aspects, we have rephrased the conclusion of our abstract section:

“This study is an initial effort to include patients from Hispanic populations in the research of breast cancer aetiology and biology, and contributes to reduce the knowledge gap in understanding the factors underlying breast cancer disparities among diverse human populations”

And our discussion, to address the concern of the reviewer. The section reads as:

“Cancer health disparities studies are often focused on the differences in measures (e.g., incidence, prevalence, mortality, etc) within the context of race or ethnicity, but might as well refer to groups of populations defined by social, geographical and economic factors, among others. While it is undeniable the impact these factors have on the utility improvements in clinical management of BC among such different groups, refining the molecular characterization of cancer genomes, as well as describing its drawbacks has revealed to be crucial to better contextualize the informed decisions we make regarding clinical care. In this respect, our data extends the knowledge and contributes towards the comprehension of the biological and molecular factors that impact upon the disparities observed in HM patients, as part of the Latin-American population, with respect to other population groups (i.e, age of diagnosis, prevalence of BC subtypes, etc)”

Data presentation of tables should include the total possible participants and missing category for data not available not multiple different N's as this is confusing.

We understand the feeling of the reviewer. Unfortunately, as occur in a large proportion of the multi-platform studies and patient-based collections, clinical information is not fully reported. Even more, not all the samples analyzed in this study, whether from our in-house collection or retrieved from public data, were characterized at DNA and RNA level. Consequently, the different analyses were conducted exclusively on the number of samples for which we had respective information, resulting in different number of samples with partial overlapping for each particular analysis. We have considered presenting tables focusing only on samples used in each analysis, since we found that presenting tables showing the entirety of samples, irrespective of if they were included or not in that analysis render a greater complexity and is confusing for the purpose of locate information about the samples included. Nonetheless, in view of the concern expressed by the reviewer and with the intention of presenting our tables in a clearer way and to avoid confusion, we clearly indicate the total number of samples (N) evaluated in each analysis in the table legend. These modifications have been incorporated along the corresponding section within the text and Supplementary table 2.

Reviewer #2 (Remarks to the Author):

We are very grateful for the reviewer's helpful comments

1. The text still overstates in various places the extendability of this data to the entirety of Mexican-Hispanic women. The authors could tone this claim down in text by strictly referring to the set of n=204 tumors presented here.

We have taken into account this particular concern. We have thoroughly reviewed our text to further adapt the content accordingly to the scope and reaching of our study. This means that a number of modifications have been made along the manuscript, seeking to replace the overstatements and trying as much as possible to refrain from making sweeping claims that could give the false impression about the extendability of our study. In particular, we have worked towards reformulating the conclusions drawn from our analysis for these to be mostly reflective of the findings within our data set and using the comparisons we make against other groups of Mexican-Hispanic or non-Hispanic women as a mean to contextualize the aspects we considered to

be more relevant of our research. We also included a brief sentence in our discussion section describing the limitation of our study.

“Even if, here we report the first relatively large and comprehensive characterization (at genomic and transcriptomic level) of breast tumors in Mexican women our observations can be only extended to the analyzed tumors and may not reflect all population rates and was not conducted in a epidemiologic design.”

2. The authors claim that PCA was ran using ADMIXTURE software whereas STRUCTURE was used to estimate individual ancestry proportions. I suspect a typo/error. ADMIXTURE is a software that estimates individual level ancestry proportions (not PCA); ADMIXTURE and STRUCTURE solve exactly the same problem.

Thank you for this observation. It was indeed a writing error. We have now fixed this description in the Methods section. This paragraph now reads as:

“The ancestry estimation calculation of the Mexicans breast cancer patients of our data set were performed using 299,411 SNPs from Affymetrix SNP6.0 microarray to inferred ancestry proportions based on the four main ancestries populations in the American Continent as reference. The first three ancestries populations were retrieved from the HapMap International Project: 27 individuals with northern European ancestry (CEU), 27 individuals with African ancestry from Yoruba in Ibadan, Nigeria and 41 East Asian ancestry (merge of Japan and China individuals) were included, together, with 37 additional Native Americans from Mexico (10 Zapotecas from Oaxaca, 13 Tepehuanos from Durango and 14 Mayas from Campeche) from the Mexican Genome Diversity Project (MGDP) 15, 16, finally we include 161 Mexican mestizo (admixed population) of six Mexican federal states to compare the ancestry mean between breast cancer patients and Mexican mestizo population. The ancestry proportions were calculated using a fast sequential quadratic programming algorithm and novel quasi-Newton acceleration method implemented in ADMIXTURE Software v.1.3.0 97, 98. Ancestry of samples from GEO series were retrieved from deposited sample information in series matrix file, TCGA ancestry data were recovered from clinical information in cbiportal and from METABRIC tumors in 99.”

3. Several methodological details to replicate the analysis are still missing; e.g., "Enrichment analysis of GO biological process terms among significantly correlated genes was performed using the package Enrichr v2.1111, 112 implemented in Bioconductor/R." What parameters were provided to the software?

We have reviewed the methodological section in search for any missing details regarding the use of programs and tools in our analyses that might preclude their reproducibility by other researchers. We now indicate to the best of our knowledge and as accurately as possible all parameters employed to run them.

The section to which the reviewer refers, now reads as:

“Functional annotation of GO biological process and KEGG terms (GO_Biological_Process_2018 and KEGG_2019_Human) among significantly correlated genes was performed running the enrichr() function from the package enrichR v2.1118, 119 . The function accepts the list of genes of interest and gene sets within the selected databases and performs a Fisher exact test to search for enrichment. We then selected the top 15 terms, ordered by raw p.value from each collection, considering genes in amplify and deleted regions separately”

4. How many tests were controlled for in the Benjamini&Hochberg procedure? Are p-values reported throughout the main text adjusted or raw?

All statistical analysis and comparisons have been controlled for multiple hypothesis testing, using Benjamini&Hochberg procedure. Therefore, statistical cutoffs for significance that we have used along the study are only based on this FDR p-value correction and not on raw p-values. We have carefully examined our manuscript to clearly state this aspect in corresponding sections.

1. Moreno-Estrada A, *et al.* Human genetics. The genetics of Mexico recapitulates Native American substructure and affects biomedical traits. *Science* **344**, 1280-1285 (2014).
2. Consortium STD, *et al.* Sequence variants in SLC16A11 are a common risk factor for type 2 diabetes in Mexico. *Nature* **506**, 97-101 (2014).
3. Villarreal-Garza C, *et al.* Breast cancer in young women in Latin America: an unmet, growing burden. *Oncologist* **18**, 1298-1306 (2013).
4. Chen Z, *et al.* Integrative genomic analyses of APOBEC-mutational signature, expression and germline deletion of APOBEC3 genes, and immunogenicity in multiple cancer types. *BMC Med Genomics* **12**, 131 (2019).

REVIEWERS' COMMENTS:

Reviewer #1 (Remarks to the Author):

I congratulate the authors for addressing my previous comments. I know they were critical, but I want you to know, this was not to be mean but because I sympathise with the cause and want to see the work clearly presentable to a wide audience. Well done!

Some last comments you may consider:

The discussion section trying to address the complexity of biological and socio-economic access to care issues could be revised as its not clear. I suggest the following below:

While it is

undeniable the impact these factor have on the utility improvements in clinical management of BC among such different groups, having diverse populations studied in the molecular characterization of cancer genomes has important value to assure inclusion in cancer research studies. In this respect, our data extends the knowledge and contributes towards the characterization of the biological and molecular factors in HM patients.

you might look at this commentary: <https://www.nature.com/articles/s41568-019-0158-0>

Reviewer #3 (Remarks to the Author):

The authors have satisfactorily addressed my previous comments.